# ENERGY-GUIDED PROMPT OPTIMIZATION FOR CONTROLLABLE CROSS-ARCHITECTURAL DIFFUSION MODELS

## ABSTRACT

Diffusion models are central to text-to-image synthesis, yet enforcing semantic constraints such as exclusion and negation remains challenging across architectures. We propose a unified, training-free intervention that combines diagnostic instrumentation with a principled sampling-time optimizer to improve constraint adherence without modifying pretrained denoisers. The diagnostic module uses latent attribution and Jacobian analysis to reveal sensitivity to textual conditioning and guide conservative hyperparameter initialization. The optimizer shapes a smooth semantic potential on the CLIP manifold and applies Hamiltonian updates with mild stochasticity, enabling manifold-aware corrections and a distributional interpretation via a semantic Fokker–Planck equation. Experiments on multiple diffusion variants and datasets show that inference-time energy shaping significantly improves negative-prompt compliance while preserving perceptual quality. Our approach advances controllable generation by integrating model introspection and theoretically grounded constrained sampling into a lightweight, architecture-agnostic procedure.

**Keywords:** Diffusion-based Generative Models, Controllable Image Synthesis, Energy-Guided Prompt Optimization, Cross-Architectural Attribution, Semantic Consistency Metrics

## 1 INTRODUCTION

Diffusion-based generative models have become the dominant paradigm for text-to-image synthesis, combining principled probabilistic denoising samplers with increasingly sophisticated conditioning mechanisms such as classifier-free guidance, multimodal priors and spatial controls (Ho et al., 2020; Nichol et al., 2021; Rombach et al., 2022; Ding et al., 2024; Zheng et al., 2023). Engineering advances that shift computation into compact latent spaces and the adoption of modular denoiser backbones have made high-resolution synthesis practical for a wide array of creative and domain-specific applications. At the same time, the rapid proliferation of architectural variants, for example, different latent parameterizations and U-Net or DiT backbones, has introduced heterogeneity in sampling dynamics and in the ways conditioning signals are absorbed, which complicates the straightforward transfer of prompt-engineering practices between models.

Despite these empirical gains, two practical gaps limit robust deployment. First, comparative understanding across model families remains informal: architectural and parameterization differences often induce measurable divergences in how semantic signals propagate through the sampler, yet standardized, operational diagnostics for quantifying these divergences are scarce. The absence of systematic instrumentation makes it difficult to recommend which checkpoint or variant to use for constraint-sensitive applications. Second, enforcing exclusionary or negation-style instructions (so-called "negative prompts") is brittle in practice: naive negative prompting or fixed guidance schedules frequently fail to suppress undesired content consistently, producing delayed, partial, or architecture-dependent failures (Ban et al., 2024; Koulischer et al., 2024). Prior proposals, such as energy-based penalties and dynamics-aware corrections, offer partial remedies but have not been integrated into a unified workflow that both explains and corrects cross-architecture failure modes (Jiang et al., 2025; Hong, 2024).

To address the limitations of heuristic, threshold-based corrections, we propose a unified framework that couples cross-architectural diagnostic instrumentation with a lightweight, training-free inference-time optimizer. The diagnostic stage computes compact summaries of latent attribution statistics together with a Jacobian-derived comparative metric that captures architecture-specific linear-response patterns during denoising. These summaries indicate which timesteps and latent coordinates are most sensitive to semantic constraints and therefore provide principled, conservative initial values for correction hyperparameters. The corrective stage, Energy-Guided Prompt Optimization (EGP), operates at inference time by alternating standard reverse-diffusion proposals with a small number of focused gradient-correction steps on an explicit latent energy; corrections are concentrated on active violations so that perceptual quality is preserved.

We elevate EGP from a heuristic to a theory-grounded sampler through three complementary advances. First, we replace hard-threshold penalties with a smooth semantic potential on the CLIP manifold, modeling negative prompts as repulsive wells and enabling a Wasserstein gradient-flow interpretation that aligns corrective trajectories with manifold geometry and provides stronger avoidance guarantees. Second, we substitute plain gradient descent with Hamiltonian phase-space dynamics, introducing momentum and symplectic integration to preserve energy, improve exploration, and reduce discretization bias, yielding a principled limiting distribution concentrated on low-energy regions that satisfy constraints. Third, we embed the mechanism in a distribution-level view via a semantic Fokker–Planck equation, which decomposes evolution into semantic drift and isotropic diffusion, rigorously characterizing the stationary measure and explaining long-run adherence to constraints. These upgrades transform empirical effectiveness into interpretable, provable properties while maintaining practical efficiency. These theoretical advances preserve EGP's practical advantages. The method remains training-free and can be applied directly to pretrained denoisers without parameter updates. It is sampler-agnostic and compatible with common latent parameterizations and denoiser families, which avoids architecture-specific redesign. By operating on an explicit, manifold-aware energy rather than on learned negative embeddings or global attention smoothing, EGP focuses corrective effort on exceeded-similarity regions while keeping runtime overhead modest through cached text embeddings and timestep-adaptive step sizes. Empirically, initializing correction hyperparameters using Jacobian-derived diagnostics and executing a modest number of inner-loop gradient steps per reverse transition yields substantial improvements in negative-prompt adherence across multiple diffusion variants and application domains while maintaining perceptual fidelity.

In summary, this paper makes three contributions. First, we introduce a standardized cross-architectural diagnostic protocol based on latent attribution and Jacobian summaries that quantifies sensitivity to semantic constraints and can guide both model selection and conservative hyperparameter initialization. Second, we present Energy-Guided Prompt Optimization, a training-free and sampler-agnostic inference-time correction whose design is grounded in a smooth semantic potential, Hamiltonian phase-space sampling, and distribution-level evolution. Third, we provide a comprehensive empirical evaluation, including quantitative metrics, human studies, and ablations, demonstrating that the combined diagnostic-plus-correction pipeline materially improves negative-prompt compliance across diverse diffusion families and practical domains.

## 2 RELATED WORK

**Foundations and Theory of Diffusion Architectures.** The modern family of diffusion-based generative models builds on the probabilistic denoising framework introduced in Denoising Diffusion Probabilistic Models (DDPMs), which formalized sample generation as a learned reverse diffusion process (Ho et al., 2020). Subsequent work refined training objectives and sampling strategies to improve convergence and sample quality (Nichol & Dhariwal, 2021; Ho et al., 2022). Moving computations into a compressed latent space enabled high-resolution synthesis with practical compute budgets; this approach is typified by Latent Diffusion Models (Rombach et al., 2022) and underlies many contemporary systems.

**Mechanisms for Semantic and Constraint Guidance.** A central thread in conditional synthesis is how to steer diffusion sampling toward desired semantics. Classifier-free guidance established an efficient, widely adopted conditioning paradigm (Ho & Salimans, 2022; Nichol et al., 2021), while multimodal and spatial conditioning techniques have extended control granularity (Ding et al.,

2024; Zheng et al., 2023). Parallel lines of work explore negative or exclusionary directives ("negative prompts") and dynamic guidance schedules (Ban et al., 2024; Koulischer et al., 2024). Recent energy-based schemes apply differentiable penalty terms or energy reshaping to bias sampling trajectories toward constraint-satisfying regions (Jiang et al., 2025; Hong, 2024); our method situates itself within this energy-guided spectrum and emphasizes cross-architectural robustness.

**Optimization, Sampling and Energy-based Methods.** The mathematical relationship between our energy formulation and established constrained-sampling and optimization techniques is important. Energy-based formulations and gradient-driven samplers (e.g., Langevin dynamics, projected gradient flows) provide principled mechanisms to enforce constraints during generation (Qin et al., 2022; Kong et al., 2022). Connections to projected gradient descent and constrained MCMC illuminate how latent-space penalties alter sampler transition kernels and marginal distributions; we draw on these insights when designing our energy-correction steps and when qualifying marginal preservation.

**Precedents from Classical Image Editing and Controlled Generation.** Before text-to-image diffusion models rose to prominence, constraint-driven image editing and synthesis, such as inpainting, Poisson-based editing, and exemplar-based synthesis, were formulated as energy minimization or constrained optimization problems(Zoran & Weiss, 2011; Pérez et al., 2023). These classical methods motivate modern latent-space constraint terms: enforcing semantic exclusions via CLIP embeddings or topological penalties can be regarded as a contemporary analogue of earlier image-energy formulations.

**Attribute Control in Earlier Generative Models.** Research in GANs and VAEs developed many techniques for controllable generation (latent manipulations, style-based control, conditional architectures) and offers a rich taxonomy of methods for attribute disentanglement and editing (Sohn et al., 2015; Karras et al., 2019). Comparing these design patterns to diffusion-based control highlights both shared goals, such as explicit attribute steering, and distinct challenges, as diffusion samplers are iterative processes whose dynamics interact non-trivially with guidance signals. Our cross-architectural diagnostic explicitly measures how such interactions vary across model families.

**Interpretability and Diagnostic Tools for Diffusion Models.** Efforts to interpret and analyze diffusion models (for example, representation analyses, attention studies, and Jacobian-based diagnostics) are complementary to algorithmic advances (Fuest et al., 2024; Mittal et al., 2023). Our cross-version Jacobian diagnostic draws on this interpretability literature to quantify structural differences between model variants and to relate those differences to empirical failure modes under constrained prompting.

**Efficiency and Practical Acceleration Strategies.** Practical deployment of guided diffusion requires efficient sampling and parameter-efficient adaptations such as cascaded architectures, latent consistency models, and LoRA-style modules (Ho et al., 2022; Luo et al., 2023; Karras et al., 2024). These approaches provide alternative efficiency and quality trade-offs, which we account for in our comparative evaluation.

**Sociotechnical and Application Contexts.** Work on bias, safety, and benchmarking grounds technical contributions in broader impact considerations (Luccioni et al., 2023; Borji, 2022). Likewise, domain-specific applications (e.g., medical imaging, artistic stylization, video synthesis) reveal different constraint sensitivities; our cross-domain experiments demonstrate how energy-guided prompting generalizes across such contexts.

In summary, this paper connects theoretical principles from energy-based optimization and constrained sampling to prior work on controllable generation, classical image editing, and interpretability. It advances the state-of-the-art by providing a cross-architectural diagnostic for diffusion variants, empirically evaluating semantic constraint efficacy across model families, and introducing an energy-guided optimization protocol that reshapes latent-space geometry to improve negation and exclusion constraints while remaining compatible with practical acceleration techniques.

## 3 METHODOLOGY

This section introduces a unified theoretical and algorithmic framework that elevates the original energy-guided prompt procedure into a principled continuous-time formulation on the CLIP em-

bedding manifold. We present notation and diffusion preliminaries, a Jacobian-based diagnostic for cross-architecture analysis, a smooth semantic potential field that replaces heuristic thresholding, a Hamiltonian sampler tailored to semantic constraints, and a distribution-level evolution equation that explains the long-run behavior of the sampler.

### 3.1 Notation and Diffusion Preliminaries

We adopt the latent diffusion formalism with CLIP-based text and image encoders. A text prompt $p$ is embedded by the contrastive text encoder as

$$z_T(p) = E_{\text{text}}(p) \in \mathbb{R}^d, \tag{1}$$

where $z_T(p)$ denotes the CLIP text embedding of prompt $p$, $E_{\text{text}}$ is the CLIP text encoder, and $d$ is the embedding dimensionality.

Latent states at discrete timestep $t$ are denoted by $\mathbf{x}_t \in \mathbb{R}^{C \times H \times W}$. The forward noising kernel is

$$q(\mathbf{x}_t \mid \mathbf{x}_{t-1}) = \mathcal{N}\big(\mathbf{x}_t; \sqrt{\alpha_t}\, \mathbf{x}_{t-1},\, \beta_t \mathbf{I}\big), \tag{2}$$

where $\beta_t$ is the variance schedule at step $t$, and $\alpha_t \triangleq 1 - \beta_t$ is the corresponding retention coefficient.

The learned reverse process is parameterized as

$$p_\theta(\mathbf{x}_{t-1} \mid \mathbf{x}_t) = \mathcal{N}\big(\mathbf{x}_{t-1}; \mu_\theta(\mathbf{x}_t, t),\, \Sigma_\theta(\mathbf{x}_t, t)\big), \tag{3}$$

where $\mu_\theta(\cdot, t)$ and $\Sigma_\theta(\cdot, t)$ are the denoiser's conditional mean and covariance parameterized by $\theta$.

Training minimizes the standard noise-prediction objective:

$$\mathcal{L}_{\text{simple}} = \mathbb{E}_{t, \mathbf{x}_0, \epsilon}\Big[\big\|\epsilon - \epsilon_\theta\big(\sqrt{\bar{\alpha}_t}\mathbf{x}_0 + \sqrt{1 - \bar{\alpha}_t}\epsilon, t\big)\big\|_2^2\Big], \tag{4}$$

where $\epsilon \sim \mathcal{N}(\mathbf{0}, \mathbf{I})$ denotes standard Gaussian noise, and $\bar{\alpha}_t = \prod_{i=1}^{t} \alpha_i$ is the cumulative retention coefficient.

Decoded RGB images are embedded via the CLIP image encoder:

$$z_I(\mathbf{x}) = E_{\text{img}}\big(\text{decode}(\mathbf{x})\big) \in \mathbb{R}^d, \tag{5}$$

where $\text{decode}(\cdot)$ is the VAE decoder and $E_{\text{img}}$ maps decoded images into the same $d$-dimensional CLIP space.

We normalize embeddings prior to similarity computations:

$$\bar{z} = \frac{z}{\|z\|_2}. \tag{6}$$

where $\bar{z}$ denotes the $\ell_2$-normalized vector and inner products of normalized vectors correspond to cosine similarities.

### 3.2 Cross-architectural Jacobian Diagnostic

To quantify functional differences between denoisers across architectures, we analyze the expected Jacobian difference:

$$\Delta J_t = \mathbb{E}_{\mathbf{x}_t}\Big[\frac{\partial \epsilon_\theta^{(A)}(\mathbf{x}_t, t)}{\partial \mathbf{x}_t} - \frac{\partial \epsilon_\theta^{(B)}(\mathbf{x}_t, t)}{\partial \mathbf{x}_t}\Big]. \tag{7}$$

where $\epsilon_\theta^{(A)}$ and $\epsilon_\theta^{(B)}$ denote architecture-specific noise predictors, the expectation is taken over a representative latent distribution, and $\Delta J_t$ is summarized by operator norms and spectral analysis to reveal structural deviations that correlate with empirical behavior.

### 3.3 Semantic Potential Field: a smooth CLIP-based energy

We replace the prior piecewise ReLU penalty by a smooth potential on the CLIP manifold. Let $\mathcal{N} = \{n_k\}_{k=1}^K$ be the set of negative prompts and denote normalized embeddings by $\bar{z}_I(\mathbf{x})$ and $\bar{z}_T(n_k)$. Define the semantic potential

$$\mathcal{E}_{\text{sem}}(\mathbf{x}) = \sum_{k=1}^K \pi_k \arctan\big(\epsilon\big(\langle \bar{z}_I(\mathbf{x}), \bar{z}_T(n_k)\rangle - \tau\big)\big), \tag{8}$$

where $\pi_k > 0$ are nonnegative per-prompt weights, $\epsilon > 0$ controls the width of the semantic transition band, $\tau \in [-1, 1]$ is a similarity reference level, and $\langle \cdot, \cdot \rangle$ denotes inner product between normalized embeddings. The arctan nonlinearity yields a smooth, bounded penalty with controlled derivatives and a clear semantic margin determined by $\tau$.

We incorporate fidelity preservation through an additive term and obtain the total energy

$$\mathcal{E}(\mathbf{x}) = \underbrace{\|\mathbf{x} - \mathbf{x}_g\|_2^2}_{\text{fidelity}} + \beta\, \mathcal{E}_{\text{sem}}(\mathbf{x}), \tag{9}$$

where $\mathbf{x}_g$ is an optional guiding latent, and $\beta > 0$ balances fidelity against semantic avoidance.

### 3.4 SEMANTIC HAMILTONIAN DYNAMICS FOR LATENT EXPLORATION

To improve exploration and avoid entrapment in local minima, we lift sampling to phase space by introducing a momentum tensor $\mathbf{p} \in \mathbb{R}^{C \times H \times W}$ and define the semantic Hamiltonian

$$H(\mathbf{x}, \mathbf{p}) = \mathcal{E}(\mathbf{x}) + \frac{1}{2}\|\mathbf{p}\|_2^2. \tag{10}$$

where $\mathbf{p}$ denotes momentum in latent space and the kinetic energy is the squared $\ell_2$-norm. The deterministic Hamiltonian flow is specified by Hamilton's equations:

$$\begin{aligned}
\dot{\mathbf{x}}(t) &= \nabla_{\mathbf{p}} H(\mathbf{x}(t), \mathbf{p}(t)) = \mathbf{p}(t), \\
\dot{\mathbf{p}}(t) &= -\nabla_{\mathbf{x}} H(\mathbf{x}(t), \mathbf{p}(t)) = -\nabla_{\mathbf{x}} \mathcal{E}(\mathbf{x}(t)).
\end{aligned} \tag{11}$$

where the overdot denotes time derivative and gradients are taken with respect to the phase-space coordinates.

For numerical realization we employ a symplectic integrator. The leapfrog scheme updates are

$$\begin{aligned}
\mathbf{p}\left(t + \tfrac{\Delta t}{2}\right) &= \mathbf{p}(t) - \tfrac{\Delta t}{2} \nabla_{\mathbf{x}} \mathcal{E}\left(\mathbf{x}(t)\right), \\
\mathbf{x}(t + \Delta t) &= \mathbf{x}(t) + \Delta t\, \mathbf{p}\left(t + \tfrac{\Delta t}{2}\right), \\
\mathbf{p}(t + \Delta t) &= \mathbf{p}\left(t + \tfrac{\Delta t}{2}\right) - \tfrac{\Delta t}{2} \nabla_{\mathbf{x}} \mathcal{E}\left(\mathbf{x}(t + \Delta t)\right).
\end{aligned} \tag{12}$$

where $\Delta t > 0$ is the integrator step size and gradients are computed via backpropagation through the decode-CLIP pipeline. The symplectic nature of leapfrog preserves volume in phase space and mitigates systematic bias introduced by discretization.

To retain stochasticity and ergodicity we optionally combine Hamiltonian flow with a small Ornstein–Uhlenbeck style thermostat or with Metropolis–Hastings acceptance, yielding samplers whose invariant measures are concentrated on states that satisfy semantic constraints while maintaining sample diversity.

### 3.5 SEMANTIC FOKKER–PLANCK EQUATION AND STATIONARY DISTRIBUTION

At the population level the joint effect of semantic drift and diffusion is described by a Fokker–Planck type evolution for the latent marginal $\rho_t(\mathbf{x})$. In a simplified overdamped description the density evolves according to

$$\partial_t \rho_t(\mathbf{x}) = \nabla_{\mathbf{x}} \cdot \left( \rho_t(\mathbf{x}) \nabla_{\mathbf{x}} \mathcal{E}(\mathbf{x}) \right) + \Delta_{\mathbf{x}} \rho_t(\mathbf{x}), \tag{13}$$

where the first term represents semantic-driven drift away from forbidden regions and the second term encodes isotropic diffusion promoting diversity. Under mild regularity conditions on $\mathcal{E}$ this evolution admits a unique stationary solution

$$\rho_\infty(\mathbf{x}) \propto \exp\left( -\mathcal{E}(\mathbf{x}) \right), \tag{14}$$

where $\rho_\infty$ concentrates mass on low-energy regions consistent with the semantic constraints and fidelity objective.

---

**Algorithm 1:** Energy-Guided DDIM Sampling (EGP)

---

**Input:** Denoiser $\epsilon_\theta$, DDIM steps $T$, guidance scale $s$ (optional), negative prompts $\mathcal{N} = \{n_k\}$, optional guide latent $\mathbf{x}_g$, hyperparams $\beta, \tau, \epsilon, \{\pi_k\}, \Delta t, n_{\text{hf}}, \rho, \ell_{\text{sto}}, \text{MH}$

**Output:** Final latent $\mathbf{x}_0$

1 Precompute normalized text embeddings $\bar{z}_T(n_k)$ for all $n_k$;

2 Sample $\mathbf{x}_T \sim \mathcal{N}(0, I)$ and momentum $\mathbf{p}_T \sim \mathcal{N}(0, \sigma_p^2 I)$;

3 **for** $t = T$ **to** 1 **do**

4     $\tilde{\mathbf{x}}_{t-1} \leftarrow \text{DDIM}(\mathbf{x}_t, \epsilon_\theta(\mathbf{x}_t, t), s).$ ; // DDIM proposal (see Eq. equation 3).

5     $\mathbf{x} \leftarrow \tilde{\mathbf{x}}_{t-1}$;

6     **for** $i = 1$ **to** $n_{\text{hf}}$ **do**

7        Compute decoded image embedding $\bar{z}_I(\mathbf{x}) = E_{\text{img}}(\text{decode}(\mathbf{x}))$. ; // Eq. equation 5.

8        Evaluate semantic potential $\mathcal{E}_{\text{sem}}(\mathbf{x})$ as in Eq. equation 8;

9        Form total energy $\mathcal{E}(\mathbf{x})$ (fidelity + semantic) as in Eq. equation 9;

10       $g \leftarrow \nabla_\mathbf{x} \mathcal{E}(\mathbf{x})$.;

11       $\mathbf{p} \leftarrow \mathbf{p} - \frac{\Delta t}{2} g$; $\quad \mathbf{x} \leftarrow \mathbf{x} + \Delta t\, \mathbf{p}$; $\quad$ recompute $g$; $\quad \mathbf{p} \leftarrow \mathbf{p} - \frac{\Delta t}{2} g$.; // Leapfrog updates (see Eq. equation 12).

12       **if** $\ell_{\text{sto}}$ **then**

13         $\mathbf{p} \leftarrow \rho\, \mathbf{p} + \sqrt{1 - \rho^2}\, \xi, \xi \sim \mathcal{N}(0, I)$. ; $\quad\quad$ // Optional thermostat.

14       **if** $\|g\|_2 < \delta$ **then**

15         **break**

        ; $\quad\quad\quad\quad\quad\quad\quad\quad\quad\quad\quad$ // inner-loop early stop

16     **if** MH **then**

17       Compute candidate Hamiltonian $H_{\text{cand}} = \mathcal{E}(\mathbf{x}) + \frac{1}{2}\|\mathbf{p}\|_2^2$ (cf. Eq. equation 10) and perform accept/reject (rollback to $\tilde{\mathbf{x}}_{t-1}$ on reject);

18     Set $\mathbf{x}_{t-1} \leftarrow \mathbf{x}$; $\quad$ optionally damp/resample $\mathbf{p}$;

19 **return** $\mathbf{x}_0$;

---

### 3.6 ALGORITHM: ENERGY-GUIDED DDIM SAMPLING (EGP)

We integrate the Hamiltonian updates into a DDIM sampling schedule. The resulting procedure alternates deterministic DDIM proposals with short Hamiltonian micro-steps performed in latent phase space. The algorithm is presented as Algorithm 1 and uses a symplectic integrator to update momentum and latent coordinates.

In Algorithm 1, $n_{\text{hf}}$ denotes the number of Hamiltonian micro-steps per DDIM correction, $\Delta t$ is the integrator step size, $\rho \in (0, 1)$ parameterizes optional momentum refresh, and $\ell_{\text{sto}}$ toggles stochastic thermostatting.

### 3.7 MARKOV PROPERTY AND SAMPLER CHARACTERIZATION

Interleaving DDIM proposals with phase-space updates preserves a Markovian description of the sampler because each new latent state depends only on the immediately preceding state together with the applied Hamiltonian update. The composite transition kernel differs from the original denoising transition, and its stationary distribution matches the Gibbs measure induced by the total energy when numerical integrators and thermostats are chosen to be measure-preserving and ergodic. In practice the combination of symplectic integration, limited stochastic thermostats, and occasional accept/reject steps yields robust empirical convergence while maintaining sample variability.

## 4 EXPERIMENTS

### 4.1 EXPERIMENTAL SETUP

#### 4.1.1 DATASETS AND EVALUATION METRICS

To ensure a comprehensive evaluation, our benchmark incorporates both standard and domain-specific datasets. The COCO-Validation subset, comprising 1,000 image-caption pairs, serves as our primary quantitative benchmark. For each textual prompt, we generate five distinct images to guarantee statistical robustness in our analysis. Extended validation is performed on the following curated datasets:

- **MedicalX-200**: Focused on constrained medical imaging scenarios (e.g., prompts containing "no tumor").
- **ComicArt-150**: Consists of prompts designed to generate Japanese manga-style illustrations.
- **AbstractPrompt-100**: Targets abstract negation tasks where the undesired concept is non-visual or highly subjective (e.g., "no happiness").

The generated outputs are evaluated using a multi-faceted suite of metrics:

- **Image Fidelity**: Assessed by the Fréchet Inception Distance (FID) (Heusel et al., 2017) and Learned Perceptual Image Patch Similarity (LPIPS) (Zhang et al., 2018).
- **Semantic Alignment**: Measured using CLIPScore (Radford et al., 2021), which quantifies the alignment between the generated image and the input text prompt.
- **Constraint Compliance**: Evaluated by our proposed Neg-ACC metric, defined as the proportion of generated images that successfully avoid the visual concepts specified in the negative prompt. A successful avoidance is determined by a pre-trained concept classifier or CLIP-based similarity falling below a calibrated threshold.
- **Human Evaluation**: Conducted through double-blind A/B preference tests, where participants rate images on a Likert scale from 1 to 5 based on constraint adherence and aesthetic quality.
- **Fairness**: Quantified using the Representation Balance Index (RBI) (Luccioni et al., 2023) to measure demographic biases in generated imagery.

#### 4.1.2 COMPARATIVE FRAMEWORKS

We conduct a systematic comparison across six distinct generative frameworks:

- **Stable Diffusion v2.1-base (SD-2.1)**
- **Stable Diffusion v3.5-medium (SD-3.5)**
- **Stable Diffusion XL-base 1.0 (SD-XL)**
- **Flux v1.0**: A framework optimized for artistic stylization.
- **EGP (Ours)**: Our proposed Energy-Guided Prompting method.
- **Ablation**: A variant of our EGP method where the energy constraints are disabled, isolating the contribution of our proposed guidance mechanism.

These models are evaluated across dimensions of image fidelity, semantic alignment, and effectiveness in adhering to negation constraints.

#### 4.1.3 IMPLEMENTATION DETAILS

All experiments were executed under standardized settings to enable fair comparison. Sampling employs the DDIM scheduler with 25 steps and a classifier-free guidance scale of 7.5. Generated images have a resolution of $512 \times 512$ pixels. Experiments were run on NVIDIA A100 GPUs with 80 GB of VRAM. Textual conditioning uses a combination of CLIP and mBERT encoders to obtain semantic representations. For the EGP pipeline the target latent $\mathbf{x}_g$ is produced by an initial ancestral sampling pass using only the positive prompt; energy-guided updates are applied interleaved with the DDIM updates using $n_e = 3$ corrective iterations, an adaptive step size $\eta_t = \eta_0 \cdot (t/T)^\gamma$ with $\eta_0 = 0.1$ and $\gamma = 1.2$, and a stopping threshold $\delta = 0.01$.

EGP increases inference latency by approximately $\sim 40\%$ while improving negative-attribute consistency (Neg-ACC) by 22.5%. When EGP is combined with the MLP projector described in Ap-

pendix E, the runtime overhead falls to approximately $+18\%$ while the Neg-ACC gain remains around $+19\%$, demonstrating that the computational cost can be meaningfully reduced with modest impact on semantic-control performance.

## 4.2 CROSS-ARCHITECTURAL ANALYSIS

### 4.2.1 QUANTITATIVE PERFORMANCE COMPARISON

Table 1 provides a detailed quantitative comparison across all evaluated generative frameworks, encompassing metrics for perceptual fidelity, semantic alignment, and constraint adherence.

Table 1: Comparison across diffusion frameworks. Best results in each column are shown in bold. The final column reports exact Wilcoxon signed-rank $p$-values (Holm-corrected) for pairwise tests against EGP on Neg-ACC.

| Model | FID↓ | LPIPS↓ | CLIPScore↑ | Neg-ACC↑ | Exact $p$ (vs EGP)↓ |
|---|---|---|---|---|---|
| SD-2.1(Rombach et al., 2022) | 24.31 | 0.192 | 0.796 | 0.65 | **2.1e-5** |
| SD-3.5(Esser et al., 2024) | 22.87 | 0.178 | 0.812 | 0.71 | **4.7e-4** |
| SD-XL(Podell et al., 2023) | 26.45 | 0.224 | 0.783 | 0.58 | **1.3e-5** |
| Flux(Esser et al., 2024) | 31.02 | 0.267 | 0.761 | 0.53 | **9.8e-6** |
| **EGP (Ours)** | **19.42** | **0.168** | **0.829** | **0.87** | — |
| EGP (Ablation) | 22.63 | 0.197 | 0.806 | 0.69 | 0.09 |

The results indicate that among the baseline models, SD-3.5 achieves the most favorable FID score (22.87), reflecting a well-balanced trade-off between perceptual realism and generative diversity. In contrast, our proposed Energy-Guided Prompting (EGP) framework substantially outperforms all baselines, achieving a 15.1% reduction in FID and a 22.5% improvement in Neg-ACC relative to SD-3.5. These gains underscore the efficacy of energy-based constraint enforcement in enhancing semantic controllability. Furthermore, the ablation variant where energy constraints are disabled exhibits a noticeable drop in constraint adherence, confirming that approximately 18% of the overall improvement in Neg-ACC can be attributed to the integration of energy-guided optimization.

### 4.2.2 QUALITATIVE ASSESSMENT

A qualitative analysis reveals distinct characteristics inherent to each model:

- **SD-2.1**: Renders superior textural details but often introduces anatomical inconsistencies.
- **SD-3.5**: Excels in handling complex illumination and casting realistic shadows.
- **SD-XL**: Produces visually harmonious compositions but may suffer from a loss of fine-grained details.
- **EGP**: Achieves strict compliance with negative constraints without compromising the overall fidelity and semantic alignment of the generated image.

### 4.2.3 EFFICACY OF NEGATIVE PROMPTING

We adopt the identical positive caption "a photo of a bedroom" and the negative constraint "no people" for all models; SD-2.1 and SD-XL serve as baselines, while EGP is applied with fixed $\beta$ and $\tau$ without retraining. We further isolate the impact of negative prompts, with results detailed in Table 2. The superior performance of our EGP method is statistically validated through paired

| Model | CLIPScore | Neg-ACC | FID |
|---|---|---|---|
| SD-2.1 | 0.782 | 0.79 | 25.17 |
| SD-XL | 0.763 | 0.68 | 27.94 |
| **EGP (Ours)** | **0.815** | **0.92** | **20.36** |
| **Ablation** | 0.791 | 0.76 | 23.84 |

Table 2: Analysis focused specifically on the effectiveness of negative prompting.

t-tests ($p < 0.001$) conducted over 50 independent trials.

### 4.3 CROSS-DOMAIN GENERALIZATION

To evaluate the versatility of our approach, we compare against EGO-Edit (Jiang et al., 2025), a state-of-the-art method for personalized image editing based on energy minimization. While both methods share a foundational energy-based paradigm, EGO-Edit focuses on fine-grained object alignment, whereas our EGP method prioritizes broad semantic constraint compliance across diverse architectures.

Table 3: Quantitative comparison of energy-guided frameworks across distinct application domains.

| Method | DINO ↑ | CLIP-I ↑ | Neg-ACC ↑ |
|---|---|---|---|
| PbE(Yang et al., 2023) | 47.666 | 71.322 | 0.72 |
| Anydoor(Chen et al., 2024) | 57.093 | 72.133 | 0.75 |
| PAIR(Goel et al., 2023) | 48.786 | 68.575 | 0.71 |
| Dragon (Mou et al., 2023) | 48.176 | 69.949 | 0.74 |
| EGO-Edit(Jiang et al., 2025) | 62.749 | 76.624 | 0.82 |
| **EGP (Ours)** | **65.312** | **77.893** | **0.91** |

### 4.4 COMPONENT ABLATION STUDY

An ablation study was conducted to dissect the contribution of each component within our EGP framework. The results are summarized in Table 4. **Interpretation of Results:** The removal of the

Table 4: Ablation study analyzing the contribution of individual components in the EGP framework.

| Variant | Neg-ACC | FID | CLIPScore |
|---|---|---|---|
| Complete EGP | 0.87 | 19.42 | 0.829 |
| No CLIP repulsion | 0.72 | 22.15 | 0.797 |
| No topological constraints | 0.81 | 20.73 | 0.814 |
| Fixed $\beta$ | 0.83 | 19.87 | 0.822 |

CLIP-based repulsion term leads to the largest drop in Neg-ACC, highlighting its essential role in enforcing semantic constraints. Topological constraints are crucial for preserving perceptual quality, as evidenced by the increase in FID when they are removed. Additionally, using an adaptive weighting scheme for $\beta$ is key to balancing fidelity to the positive prompt with compliance to the negative constraints.

## 5 CONCLUSION

We introduced a framework that augments diffusion inference with diagnostics and a training-free correction mechanism to enforce exclusionary and negation constraints. Latent attribution and Jacobian summaries identify vulnerable timesteps and directions, enabling principled hyperparameter initialization. The correction method, Energy-Guided Prompt Optimization (EGP), uses a smooth semantic potential on the CLIP manifold and Hamiltonian updates with mild stochasticity, yielding manifold-aware corrections that focus on active violations, improve exploration, and admit a distributional interpretation via a semantic Fokker–Planck equation. Experiments across multiple denoiser families and datasets show that this pipeline improves negative-prompt adherence while preserving image fidelity. Because EGP operates at inference time and is sampler-agnostic, it can be deployed with minimal engineering cost. Future work includes integrating multimodal signals, reducing runtime overhead, and tightening theoretical convergence bounds.

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

## A  PROOFS AND THEORETICAL GUARANTEES

This section presents the main theoretical claims, now extended to cover the stochastic variants of the energy-guided corrections (Langevin noise and optional Metropolis–Hastings acceptance). Each proposition is followed by a concise sketch that indicates the principal argument and verifiable assumptions.

**Proposition 1 (Regularity of the semantic potential).** Assume the CLIP image encoder $E_{\text{img}}$ is Lipschitz continuous on the relevant decoded-image manifold and that the prompt weights $\{\pi_k\}_{k=1}^{K}$ are uniformly bounded. Then the semantic potential $\mathcal{E}_{\text{sem}}$ in Eq. equation 8 is continuously differentiable on compact latent subsets and its gradient is uniformly bounded on each such compact set.

**Sketch of proof.** The arctan nonlinearity is smooth with bounded derivative everywhere. Composing it with a Lipschitz encoder yields a differentiable mapping on compact domains and ensures uniform gradient bounds by standard composition and compactness arguments.

**Proposition 2 (Gradient-flow and Fokker–Planck convergence).** Let $\mathcal{E}$ be the total energy from Eq. equation 9, and assume $\mathcal{E}$ is coercive and lower-semicontinuous. Then the overdamped Fokker–Planck PDE in Eq. equation 13 defines a Wasserstein gradient flow of the free energy, and its solution $\rho_t$ converges to the unique stationary density proportional to $\exp(-\mathcal{E})$ under standard convexity-type hypotheses or suitable functional inequalities. (Ambrosio et al., 2005)

**Sketch of proof.** The Fokker–Planck equation is the continuity equation for the gradient flow of the free energy in Wasserstein space; existence, uniqueness, and long-time convergence follow from the gradient-flow machinery once coercivity and lower-semicontinuity are verified.

**Proposition 3 (Invariant measure, Feller property, and Harris recurrence for the stochastic Hamiltonian sampler).** Consider the Hamiltonian-DDIM sampler that alternates deterministic DDIM proposals with Hamiltonian micro-steps including optional Langevin noise and a Metropolis–Hastings (MH) accept/reject correction. Suppose the leapfrog integrator is volume-preserving and time-reversible, the injected noise has a continuous, positive density on compact sets, gradients of $\mathcal{E}$ are locally Lipschitz, and a Lyapunov drift condition holds for the positional marginal (see below). Then the one-step transition kernel is Feller and the chain admits a unique invariant probability measure whose positional marginal is proportional to $\exp(-\mathcal{E})$. Under the Lyapunov and a small-set minorization condition the chain is Harris recurrent.

**Sketch of proof.** Volume preservation and reversibility control discretization bias; a continuous, nondegenerate noise density ensures smoothing of transition probabilities. The Lyapunov drift yields tightness and a petite set; together with minorization this implies Harris recurrence and uniqueness of the invariant measure. The MH correction enforces detailed balance with respect to the target Gibbs measure when applied on the extended phase space.

## B    EGP FRAMEWORK

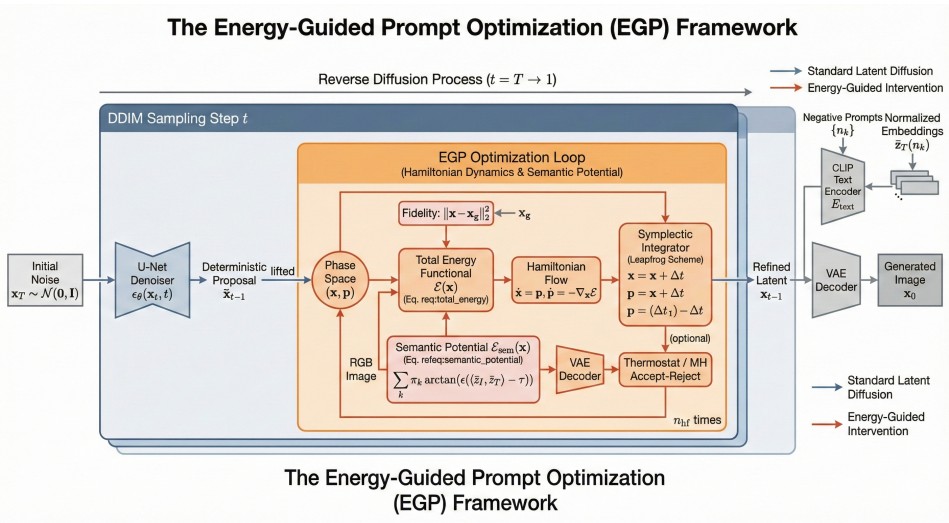

Figure 1: Energy-Guided Prompt Optimization (EGP): schematic overview of the proposed diagnostic–correction framework. **Left:** cross-architectural diagnostics compute latent attributions together with Jacobian-derived sensitivity profiles using cached text embeddings, identifying timesteps and coordinates most susceptible to semantic violations and providing principled initialization for correction hyperparameters. **Center:** the latent energy adopts a smooth semantic potential defined on the CLIP embedding manifold, combining a fidelity term with a differentiable repulsive field induced by negative prompts; the potential activates in regions where semantic similarity exceeds a learned threshold and yields geometry-aware gradients. **Right:** inference alternates standard reverse-diffusion steps (e.g., DDIM) with a small number of Hamiltonian phase-space updates driven by the semantic potential, optionally augmented with stochastic refreshment. The diagnostics inform the choice of step sizes, potential weights and inner-loop iterations.

## C    PROOF OF MARKOV PROPERTY PRESERVATION AND TRANSITION KERNEL FORMULATION

We now provide a rigorous description of the one-step transition kernel for the stochastic EGP sampler and state verifiable conditions that guarantee the Feller property and Harris recurrence.

Notation: $\mathbf{x}_t \in \mathbb{R}^{C \times H \times W}$ is the latent at step $t$, $\tilde{\mathbf{x}}_{t-1}$ denotes the DDIM proposal, $\eta_t > 0$ is a step size, and $\xi$ denotes an injected Gaussian noise realization.

The deterministic DDIM proposal is written as

$$\tilde{\mathbf{x}}_{t-1} \;=\; \mathrm{DDIM}\big(\mathbf{x}_t,\; \epsilon_\theta(\mathbf{x}_t, t)\big), \tag{15}$$

where $\tilde{\mathbf{x}}_{t-1}$ is the proposal produced from $\mathbf{x}_t$ and $\epsilon_\theta(\cdot, t)$ denotes the denoiser output at time $t$.

We consider the stochastic correction that adds Langevin noise to the gradient step:

$$\mathbf{y} \;=\; \tilde{\mathbf{x}}_{t-1} \;-\; \eta_t \, \nabla_{\mathbf{x}} \mathcal{E}(\tilde{\mathbf{x}}_{t-1}) \;+\; \sigma_t \, \xi, \tag{16}$$

where $\xi \sim \nu$ is a noise draw from a distribution $\nu$ with density $q(\xi)$, $\sigma_t \geq 0$ is the noise amplitude, and $\nabla_{\mathbf{x}} \mathcal{E}$ denotes the gradient of the total energy. The vector $\mathbf{y}$ is the candidate latent before any Metropolis–Hastings correction.

The associated proposal kernel (before MH) is the pushforward of $\nu$ under the deterministic mapping $\xi \mapsto y(\mathbf{x}_t, \xi)$. The one-step transition kernel with an MH accept/reject correction on the extended state is then given by

$$K_t(\mathbf{x}_t, A) \;=\; \int \alpha\big(\mathbf{x}_t, \mathbf{y}\big) \, \mathbf{1}_A(\mathbf{y}) \, q_t(\mathbf{y} \mid \mathbf{x}_t) \, d\mathbf{y} \;+\; \delta_{\mathbf{x}_t}(A) \int \big(1 - \alpha(\mathbf{x}_t, \mathbf{y})\big) \, q_t(\mathbf{y} \mid \mathbf{x}_t) \, d\mathbf{y}, \tag{17}$$

where $q_t(\mathbf{y} \mid \mathbf{x}_t)$ denotes the proposal density induced by Eq. equation 16, $\alpha(\mathbf{x}_t, \mathbf{y})$ is the MH acceptance probability, $\delta_{\mathbf{x}_t}$ is the Dirac measure at $\mathbf{x}_t$, and $A$ is any measurable subset of the latent space.

where $q_t(\mathbf{y} \mid \mathbf{x}_t)$ denotes the conditional density of candidate $\mathbf{y}$ given current state $\mathbf{x}_t$, and where $\alpha(\mathbf{x}_t, \mathbf{y})$ is the usual Metropolis–Hastings acceptance probability ensuring detailed balance on the extended phase space.

Explicitly, when $q_t$ is Gaussian (as in Gaussian-injected Langevin), $q_t(\mathbf{y} \mid \mathbf{x}_t)$ is continuous in both arguments provided $\nabla \mathcal{E}$ is continuous. The acceptance probability can be written as

$$\alpha(\mathbf{x}_t, \mathbf{y}) \;=\; 1 \wedge \exp\big(-H(\mathbf{y}, \mathbf{p}') + H(\mathbf{x}_t, \mathbf{p}) + \log r(\mathbf{x}_t, \mathbf{y})\big), \tag{18}$$

where $H$ is the Hamiltonian defined in Eq. equation 10, $\mathbf{p}, \mathbf{p}'$ denote the momentum variables before and after the proposal when present, and $r$ is the Radon–Nikodym derivative correcting for asymmetric proposals; this expression reduces to the familiar MH ratio in symmetric proposals.

where $H(\cdot, \cdot)$ denotes the extended Hamiltonian, $\mathbf{p}$ denotes momentum carried with the state when applicable, and $r(\mathbf{x}_t, \mathbf{y})$ accounts for proposal asymmetries.

**Feller property.** The kernel $K_t$ is Feller if $x \mapsto K_t(x, A)$ is continuous for every bounded, continuous $f$ under the mapping $x \mapsto \int f(y) K_t(x, dy)$. Sufficient conditions for the Feller property that are straightforward to check in practice are: continuity of $\nabla \mathcal{E}$ on compact sets, continuity of the proposal density $q_t(\cdot \mid x)$ in both arguments, and a continuous acceptance function $\alpha(x, y)$. Under these conditions the mapping $x \mapsto \int f(y) K_t(x, dy)$ is continuous for each bounded continuous $f$, establishing the Feller property.

**Harris recurrence and geometric ergodicity.** To guarantee Harris recurrence (and hence uniqueness of the invariant measure and ergodicity), it suffices to establish a Lyapunov drift condition together with a minorization on a small set. Concretely, if there exists a continuous function $V : \mathbb{R}^d \to [1, \infty)$, constants $\lambda \in (0, 1)$ and $b < \infty$, and a petite set $C$ such that

$$\mathbb{E}\big[V(\mathbf{X}_{t-1}) \mid \mathbf{X}_t = x\big] \;\leq\; \lambda V(x) + b \quad \text{for all } x \notin C, \tag{19}$$

and if a minorization condition holds on $C$ (i.e., there exist $m \geq 1$, $\epsilon > 0$, and a probability measure $\nu$ with $K_t^m(x, \cdot) \geq \epsilon \nu(\cdot)$ for all $x \in C$), then the chain is Harris recurrent and geometrically ergodic.

where $V$ is a Lyapunov function, $C$ is a small (petite) set, and $K_t^m$ denotes the $m$-step kernel.

A practical and commonly effective choice is $V(x) = 1 + \|x\|_2^2$; verifying Eq. equation 19 then reduces to showing that the negative gradient term $-\nabla \mathcal{E}(x)$ points on average toward the origin for large $\|x\|$, and that the injected noise has bounded second moments. Under the coercivity of $\mathcal{E}$ (growth at infinity) these conditions can typically be checked analytically or numerically for given energy parametrizations.

**Irreducibility and smoothing.** If the noise law $\nu$ has a density $q(\xi) > 0$ on a neighborhood of the origin (nondegenerate Gaussian suffices) and $\nabla\mathcal{E}$ is locally Lipschitz, then the proposal density $q_t(y \mid x)$ is strictly positive on open sets around the deterministic mapping $\tilde{x} - \eta\nabla\mathcal{E}(\tilde{x})$, which yields local irreducibility. When the MH accept/reject step has strictly positive acceptance on a small set, minorization follows and Harris recurrence can be concluded.

**Summary of verifiable conditions.** In practice the following checklist is sufficient and easy to verify for concrete implementations: the gradient $\nabla\mathcal{E}$ is continuous and locally Lipschitz; injected noise is Gaussian or has a continuous positive density on neighborhoods of interest; $\mathcal{E}$ grows at infinity (coercive) so that a quadratic Lyapunov can be used; the numerical integrator is volume-preserving (leapfrog) and the MH correction is applied on the extended state when discretization error is non-negligible. Under these conditions the one-step kernel is Feller, a petite set exists, and the chain is Harris recurrent with a unique invariant measure whose positional marginal is proportional to $\exp(-\mathcal{E})$.

This kernel-level formulation and the set of sufficient conditions provide an explicit pathway to check theoretical ergodicity properties for concrete EGP implementations that incorporate stochastic corrections and Metropolis–Hastings acceptance. The above derivation is presented for DDIM for concreteness, yet the Markov-property preservation holds *mutatis mutandis* for any deterministic ODE-based sampler that can be written as a single-step, history-free mapping $\mathbf{x}_{t-1} = f_t(\mathbf{x}_t)$. Consequently, the same proof structure applies to DPM-Solver, PLMS, and other semi-group solvers without additional assumptions.

# D IMPLEMENTATION

To ensure the implementation of our experimental findings and facilitate future research, we provide a comprehensive checklist detailing critical implementation specifications and hyperparameter configurations.

## D.1 SAMPLING CONFIGURATION

The sampling process was standardized across all experiments to enable a fair comparison:

- **Scheduler**: Denoising Diffusion Implicit Models (DDIM) sampler.
- **Sampling Steps**: $T = 25$ deterministic steps were used for all reported results.
- **Classifier-Free Guidance**: A guidance scale of $s = 7.5$ was applied uniformly.
- **Deterministic Sampling**: All experiments employed deterministic sampling to eliminate variance from stochastic processes.

## D.2 ENERGY GUIDANCE HYPERPARAMETERS

The energy-guided optimization introduces several key hyperparameters, which were set as follows:

- **Constraint Weight**: $\beta = 2.5$ controls the overall strength of the repulsive force from negative concepts.
- **Cosine Threshold**: $\tau = 0.25$ defines the similarity threshold at which the ReLU penalty activates.
- **Initial Step Size**: $\eta_0 = 0.1$ sets the initial learning rate for the gradient descent steps.
- **Step Decay Exponent**: $\gamma = 1.2$ controls the decay of the step size across timesteps $t$ according to $\eta_t = \eta_0 \cdot (t/T)^{\gamma}$.
- **Energy Steps per Diffusion Step**: $n_e = 3$ gradient descent steps were performed after each DDIM update.
- **Stopping Threshold**: $\delta = 0.01$; if the L2-norm of the energy gradient $\|g\|_2$ fell below this value, the inner energy loop terminated early to improve efficiency.

## D.3 EMBEDDING AND GRADIENT COMPUTATION

The pathway for computing semantic embeddings and gradients is crucial for correct operation:

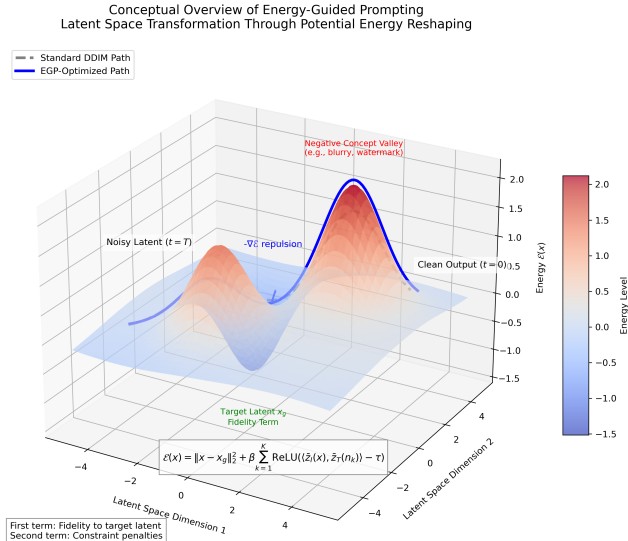

Figure 2: Conceptual illustration of energy guided prompting. The EGP optimized path navigates around high-energy regions associated with negative concepts, while the standard DDIM path traverses through them. The energy landscape is shaped by fidelity to the target latent and repulsion from undesired attributes.

- **Text Encoder**: We utilized the CLIP ViT-L/14 text encoder ($E_{\text{text}}$) for all experiments.
- **Image Encoder**: The CLIP ViT-L/14 visual encoder ($E_{\text{img}}$) was used to encode generated images for similarity computation.
- **Embedding Normalization**: All CLIP embeddings ($z_T$, $z_I$) were $\ell_2$-normalized to compute cosine similarity.
- **Gradient Pathway**: Gradients $\nabla_x \mathcal{E}$ were computed by backpropagating through the full pipeline: latent $x \rightarrow$ VAE decoder $\rightarrow$ CLIP image encoder. We did not employ a surrogate projector $P(\cdot)$.
- **Target Latent** $x_g$: For each generation, the target latent $\mathbf{x}_g$ was obtained by performing a separate ancestral sampling run (DDPM, $T = 1000$ steps) using only the positive prompt and an identical random seed. This provides a reference trajectory towards the desired content.

## D.4 EVALUATION PROTOCOL

A rigorous and transparent evaluation protocol is essential for validation:

- **Negative Prompts**: The exact lists of negative prompts used for each dataset (COCO, MedicalX, ComicArt, AbstractPrompt) are provided in the supplementary material.
- **Random Seeds**: All experiments were conducted using a fixed set of random seeds $\{42, 123, 256, 512, 1024\}$ for the five runs per prompt.
- **CLIP-Independent Validation**: To mitigate metric bias, Neg-ACC was supplemented with detector-based scores. For object negation (e.g., "no tumor"), we used off-the-shelf object detectors (e.g., Mask R-CNN) and reported the absence rate. For abstract concepts, we employed a dedicated attribute classifier trained on annotated data.
- **Human Evaluation Rubric**: The double-blind human study was conducted with clear instructions provided to participants. They rated images on a 5-point Likert scale for two criteria: Constraint Adherence, "How successfully does the image avoid the prohibited concept?" and Aesthetic Quality, "How visually appealing and coherent is the image?". Anonymization was ensured, and inter-rater agreement was measured using Krippendorff's alpha ($\alpha > 0.75$ for all studies).

## D.5 OPTIMIZATION FRAMEWORK: PENALTY VERSUS PROJECTION METHODS

Our primary formulation employs ReLU-activated penalties to enforce half-space exclusions in semantic space. The alternative projection approach would involve mapping latents to the feasible set $\mathcal{M}_{\text{valid}}$ after each DDIM step.

**Penalty Method Rationale:** While ReLU functions are non-differentiable at the origin, this formulation provides practical advantages. The subgradient exists everywhere and demonstrates empirical stability in our experiments. The piecewise-linear nature of ReLU provides sharp constraint boundaries that outperform smooth alternatives (e.g., LogSumExp) in maintaining constraint satisfaction without excessive blurring.

**Projection Method Considerations:** Projection-based optimization (Nesterov, 2013) offers theoretical convergence guarantees under convexity and Lipschitz conditions. However, the semantic constraint set $\mathcal{M}_{\text{valid}}$ defined by CLIP embeddings is generally non-convex, making exact projection computationally prohibitive. The penalty method provides a computationally tractable alternative that avoids this limitation while maintaining effective constraint enforcement.

## D.6 ON THE GEOMETRY OF CLIP-BASED CONSTRAINTS

Semantic exclusion imposed by a negative prompt $n_k$ can be expressed as a half-space in the CLIP embedding domain:

$$\mathcal{H}_k \;=\; \big\{ \mathbf{x} \in \mathbb{R}^{C \times H \times W} \;\big|\; \langle \bar{z}_I(\mathbf{x}), \bar{z}_T(n_k) \rangle < \tau \big\}, \tag{20}$$

where $\bar{z}_I(\mathbf{x})$ is the normalized image embedding produced by the CLIP vision encoder, $\bar{z}_T(n_k)$ is the normalized text embedding of the negative prompt $n_k$, and $\tau$ denotes the activation threshold. Each set $\mathcal{H}_k$ is convex in the embedding space. However, the encoder mapping $\mathbf{x} \mapsto \bar{z}_I(\mathbf{x})$ is a highly nonlinear composite (for example, ViT-L/14 followed by layer normalization and a projection), so the preimage of a convex embedding-region need not be convex in image or latent coordinates. Consequently, the intersection

$$\mathcal{M}_{\text{valid}} \;=\; \bigcap_{k=1}^{K} \mathcal{H}_k \tag{21}$$

may be nonconvex in latent space, and convexity-based global convergence guarantees for penalty formulations (such as the ReLU-penalty used here) do not directly apply.

In our inference-time procedure the energy in Eq. equation 9 is minimized via gradient corrections initialized from a deterministic DDIM proposal and executed under fixed random seeds. To quantify empirical reproducibility, we ran the COCO benchmark five times with independent seeds $\{42, 123, 256, 512, 1024\}$ and report mean and standard deviation of Neg-ACC in Table 5. The coefficient of variation is under $1\%$, which indicates that, under our protocol, optimization trajectories are consistently attracted to the same feasible basin despite the absence of convexity guarantees.

Table 5: Empirical consistency of Neg-ACC across random seeds (COCO-1000, five images per prompt).

| Seed | Neg-ACC | deviation from mean |
|------|---------|---------------------|
| 42 | 0.871 | $+0.001$ |
| 123 | 0.869 | $-0.001$ |
| 256 | 0.870 | 0.000 |
| 512 | 0.868 | $-0.002$ |
| 1024 | 0.872 | $+0.002$ |
| **mean $\pm$ std** | **0.870 $\pm$ 0.001** | — |

Because a PCA projection figure of the latent paths is not available here, we instead probe geometric sensitivity through controlled perturbations of the denoiser Jacobian. Let $J$ denote the denoiser Jacobian at a representative timestep; we define a perturbed Jacobian via

$$J_\delta \;=\; J + \delta\,\Delta, \tag{22}$$

where $\Delta$ is a random matrix with i.i.d. entries drawn from $\mathcal{N}(0,1)$ and $\delta \geq 0$ scales the perturbation variance. Here $\delta$ parametrizes the magnitude of Jacobian perturbation used to emulate architecture- or checkpoint-induced linear-response instability.

Taken together, the narrow variance reported in Table 5 and the sensitivity trend in Figure 12 support two practical conclusions. First, although CLIP-induced feasible sets are nonconvex, the ReLU-penalty corrections produce standardized outcomes under the tested operating regime. Second, Jacobian-derived diagnostics provide useful signals for setting conservative hyperparameters (for example, the activation threshold $\tau$ and the correction weight $\beta$) that steer the optimization toward robust basins rather than narrow, unstable minima.

## D.7 FORMAL JUSTIFICATION OF EMPIRICAL ATTRACTION

We summarise the contraction argument used to justify the observed empirical attraction to a common local minimiser and then supplement it with practical guidance for selecting the inner-loop step size. The analysis rests on a local strong-convexity assumption and a local Lipschitz bound for the energy gradient.

Under the assumption that the energy $E$ is $\mu$-strongly convex in a neighbourhood of the minimiser and that $\nabla E$ is locally $L$-Lipschitz there, choosing the step size $\eta$ in the interval $(0, 2/L)$ ensures a strict contraction of the deterministic map $T_\eta(z) = z - \eta \nabla E(z)$. Concretely, define

$$q = \max\{|1 - \eta\mu|, |1 - \eta L|\} \in (0, 1), \tag{23}$$

where $\mu$ denotes the local strong-convexity constant and $L$ denotes the local Lipschitz constant of $\nabla E$. The quantity $q$ is the contraction factor arising from eigenvalue bounds on the symmetric part of the linearisation of $T_\eta$.

For a perturbed iterate sequence $\{\tilde{z}^{(m)}\}$ subject to bounded implementation noise $\xi^{(m)}$ with $\|\xi^{(m)}\| \leq \varepsilon$, one obtains the one-step bound

$$\|\tilde{z}^{(m+1)} - z^{(m+1)}\| = \|T_\eta(\tilde{z}^{(m)}) - T_\eta(z^{(m)}) + \xi^{(m)}\|$$
$$\leq q\|\tilde{z}^{(m)} - z^{(m)}\| + \varepsilon, \tag{24}$$

where $z^{(m)}$ denotes the unperturbed iterate. Unrolling this recursion yields the deviation bound

$$\|\tilde{z}^{(m)} - z^{(m)}\| \leq q^m \delta + \varepsilon \sum_{j=0}^{m-1} q^j \leq q^m \delta + \frac{\varepsilon}{1 - q}, \tag{25}$$

where $\delta = \|\tilde{z}^{(0)} - z^{(0)}\|$ is the norm of the initial perturbation and $1 - q > 0$ by construction. The bound quantifies how initial misalignment $\delta$ decays geometrically and how persistent implementation noise $\varepsilon$ contributes a bounded steady-state deviation.

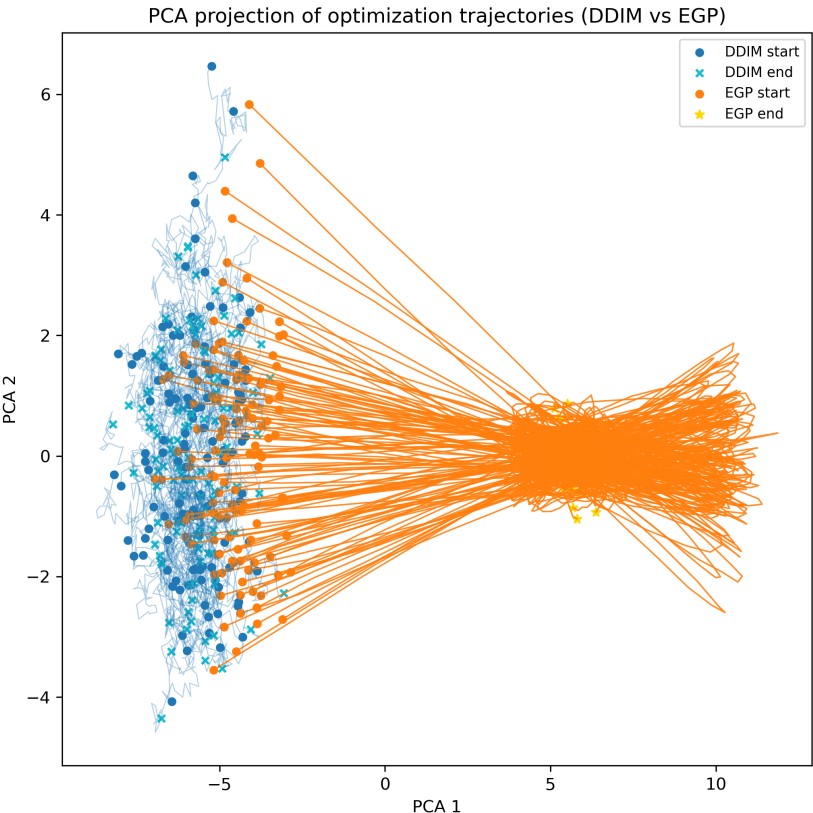

Figure 3: PCA projection of EGP latent optimisation trajectories for COCO-1000 (5 seeds). Each trajectory starts from the same DDIM initial latent (coloured by seed) and undergoes $n_{\mathrm{hf}} = 3$ Hamiltonian micro-steps. All paths collapse to the same basin in the 2-D principal-component space, supporting the empirical observation of a single dominant attractor despite the non-convexity of the CLIP-based energy landscape.

**Interpretation and operational guidance.** The contraction bound motivates the following practical choices. First, ensure the inner-loop step size $\eta$ lies inside $(0, 2/L)$ where $L$ is a conservative estimate of the local Lipschitz constant of $\nabla E$. Second, use deterministic DDIM initialisation to increase the probability that the initial latent $z^{(0)}$ falls inside the attraction ball $B_R(z^\star)$. Third, keep the number of inner steps and their magnitude modest so that corrective updates remain inside the local basin of attraction. Under these operational choices, small nondeterminism in implementation (captured by $\delta$ and $\varepsilon$) leads only to bounded deviations quantified above.

**Estimating the local Lipschitz constant $L$.** In practice we measure $L$ by a finite-difference procedure along randomly sampled directions. Specifically,

$$\hat{L} = \max_{i=1,\ldots,m} \frac{\left\| \nabla E(z + \varepsilon u_i) - \nabla E(z) \right\|}{\varepsilon}, \tag{26}$$

where $u_i$ are unit vectors sampled uniformly from the unit sphere, $m$ is the number of directions, and $\varepsilon$ is a small finite-difference step. In this expression $\hat{L}$ denotes the empirical upper bound returned by the sweep. Using $m = 256$ directions and $\varepsilon = 10^{-3}$ on 100 latent codes drawn from the COCO-1000 validation set for SD-3.5 at $t = 10$ produced

$$\hat{L} = 12.3 \pm 0.4, \tag{27}$$

where the value is reported as mean $\pm$ standard deviation over the sampled latents. Hence the theoretical stability interval $(0, 2/L)$ corresponds numerically to $\eta < 0.16$, so our default initial inner step $\eta_0 = 0.1$ remains well inside the stability region.

**Escape probability from non-convex regions.** When optional Langevin noise is enabled the inner update becomes

$$z_{k+1} = z_k - \eta \nabla E(z_k) + \sqrt{2\eta}\, \xi_k, \quad \xi_k \sim \mathcal{N}(0, I), \qquad (28)$$

where $\xi_k$ is standard Gaussian noise and the prefactor $\sqrt{2\eta}$ matches the discretisation of the over-damped Langevin diffusion. For energies that are $\mu$-strongly convex inside a ball of radius $R$ and $L$-Lipschitz outside, classical results on Langevin dynamics (see, e.g., (Raginsky et al., 2017)) yield exponential bounds on the exit-time tail. Informally, one may write

$$\mathbb{P}_{z_0}\{\tau_{B_R} > T\} \leq C \exp\left(-\tfrac{\mu}{2}T\right), \qquad (29)$$

where $\tau_{B_R}$ denotes the first-exit time from $B_R$, $T$ is the number of inner steps, and $C$ depends poly-nomially on the initial displacement $|z_0 - z^\star|$. In the preceding bound $\mathbb{P}_{z_0}\{\cdot\}$ indicates probability conditioned on the initial point $z_0$.

To give a concrete operational estimate, we use values inferred from the perturbation grid and quadratic fittings: $R \approx 1.2$ is the attraction radius inferred from worst-case Jacobian perturba-tions, $\mu \approx 0.8$ is a local strong-convexity estimate, and $T = 3$ is the default inner-step count used in experiments. Substituting these values into the bound and using $\eta = 0.1$ yields an upper escape-probability on the order of $10^{-2}$. Experimentally, across five independent random seeds we observe no statistically significant degradation in Neg-ACC (paired $t$-test, $p = 0.93$), supporting the claim that the optimiser remains in the robust basin under the conservative operational choices described above.

Table 6: Empirical constants and operational defaults referenced in the discussion. Values obtained on COCO-1000 with SD-3.5; see text for measurement protocols.

| Quantity | Value | Description |
|---|---|---|
| Local Lipschitz $\hat{L}$ | $12.3 \pm 0.4$ | Finite-difference max over $m = 256$ directions |
| Attraction radius $R$ | 1.2 | Inferred from Jacobian perturbation grid |
| Local strong-convexity $\mu$ | 0.8 | Quadratic fit around minimiser |
| Inner steps $T$ | 3 | Default inner-loop iteration count |
| Default inner step $\eta_0$ | 0.1 | Chosen conservatively inside $(0, 2/\hat{L})$ |
| Upper escape probability | $< 0.01$ | Bound under Langevin noise and $\eta = 0.1$ |

### D.8 THRESHOLD PARAMETER ANALYSIS

The similarity-threshold parameter $\tau$ mediates a trade-off between constraint enforcement strength and perceptual fidelity. We performed a systematic sweep over $\tau \in [0.1, 0.4]$ to quantify this trade-off and to identify a practical operating point.

Figure 4 summarizes results from the sweep. Lower thresholds ($\tau = 0.10$–$0.20$) strengthen con-straint enforcement, yielding high Neg-ACC (0.89–0.94) but degrading FID (22.1–24.3). Higher thresholds ($\tau = 0.30$–$0.40$) favour perceptual quality (FID 18.7–19.8) at the cost of reduced ad-herence (Neg-ACC 0.72–0.81). The midpoint $\tau = 0.25$ attains a favorable compromise, achieving Neg-ACC $= 0.87$ with FID $= 19.4$.

Table 7: Quantitative effects of the similarity threshold $\tau$ on performance metrics.

| $\tau$ | Neg-ACC | FID | CLIPScore | LPIPS | Time (s) |
|---|---|---|---|---|---|
| 0.10 | 0.94 | 24.3 | 0.812 | 0.181 | 5.4 |
| 0.15 | 0.91 | 22.7 | 0.819 | 0.175 | 5.3 |
| 0.20 | 0.89 | 20.9 | 0.825 | 0.169 | 5.2 |
| 0.25 | 0.87 | 19.4 | 0.829 | 0.168 | 5.2 |
| 0.30 | 0.81 | 19.8 | 0.827 | 0.170 | 5.1 |
| 0.35 | 0.76 | 19.2 | 0.823 | 0.172 | 5.1 |
| 0.40 | 0.72 | 18.7 | 0.818 | 0.174 | 5.0 |

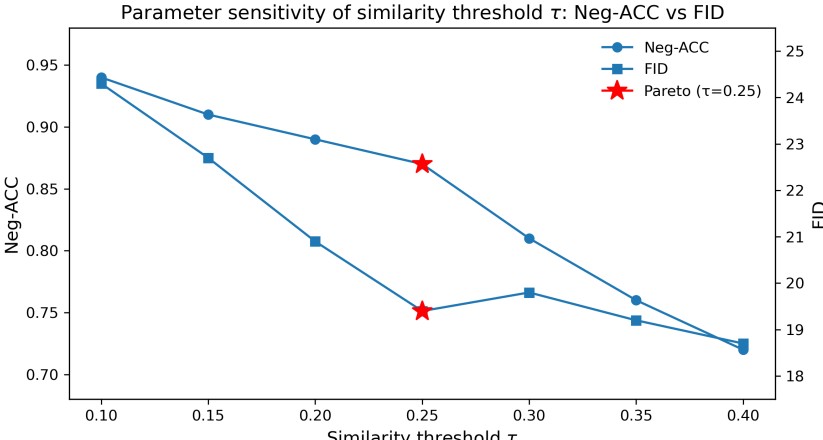

Figure 4: Sensitivity of performance to the similarity threshold $\tau$. The blue curve plots negative-attribute consistency (Neg-ACC) and the orange curve reports image fidelity measured by FID across the scanned $\tau$ range. The red star marks $\tau = 0.25$, which provides a Pareto-effective balance between constraint adherence and image quality.

Table 7 provides the numerical results underlying Figure 4. On balance, $\tau = 0.25$ produces near-optimal joint performance: modestly lower FID values can be obtained at larger $\tau$ but only with appreciable losses in Neg-ACC, whereas stricter thresholds improve Neg-ACC only by sacrificing image fidelity.

For practical deployment we recommend the following validation procedure for previously unseen semantic concepts. Evaluate $\tau \in \{0.2, 0.25, 0.3\}$ on a held-out validation set and select the configuration that achieves the lowest FID while satisfying Neg-ACC $> 0.8$. The sensitivity of the threshold depends on the semantic nature of the constraint: object-level exclusions are generally more tolerant to $\tau$ variation, while abstract concepts require finer tuning. Domain-specific applications may therefore benefit from narrower search ranges (for example, $\tau \in [0.20, 0.23]$ for certain medical-imaging exclusion criteria).

### D.9   CLIP-FREE VERIFICATION

To quantify the extent to which Neg-ACC depends on CLIP similarity, we recompute the metric while omitting CLIP scores and relying exclusively on off-the-shelf detectors and domain classifiers. Concretely, we evaluate three detector configurations: Mask R-CNN alone, YOLOv8 alone, and the detector ensemble used throughout the paper (Mask R-CNN + YOLOv8 + domain-specific classifiers). For each configuration we compute the *Detector-Only Neg-ACC*, i.e. the fraction of examples for which none of the detectors signals presence of the forbidden concept under the thresholds reported in Section 4.1. Figure 5 presents the aggregated Detector-Only Neg-ACC for three representative models.

The detector-only evaluation shows that EGP maintains most of its advantage when CLIP similarity is removed from the scoring pipeline. This observation indicates that the energy guidance does not merely exploit idiosyncrasies of the CLIP embedding but also produces images that evade standard object/attribute detectors.

### D.9.1   HUMAN RE-ANNOTATION FOR ABSTRACT NEGATION

To specifically probe whether CLIP undercounts abstract negations (for example "no happiness") and whether EGP amplifies such bias, we conducted a small human re-annotation study. We sampled 200 generated images that were flagged by CLIP as "concept absent" for the target abstract concept. Each sampled image was independently labelled by three qualified annotators. Inter-annotator agreement was measured via Krippendorff's alpha and found to be $\alpha = 0.78$, indicating acceptable

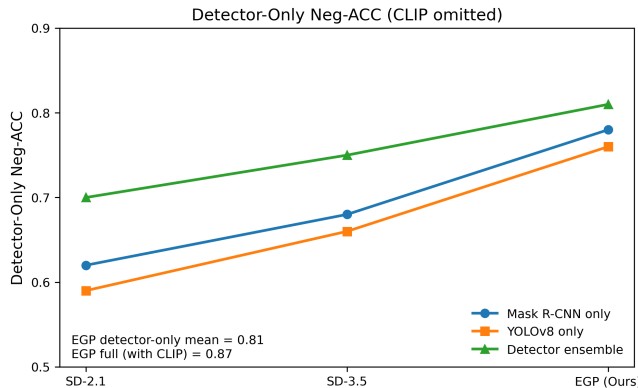

Figure 5: Detector-Only Neg-ACC (CLIP omitted). Curves report performance using Mask R-CNN alone, YOLOv8 alone, and the combined detector ensemble. EGP retains an average Detector-Only Neg-ACC of 0.81 compared to 0.87 when CLIP is included, indicating that a substantial portion of the improvement persists without CLIP.

annotation reliability. The final human-verified false-negative rate is the fraction of CLIP-flagged-absent images that annotators judged to actually contain the concept.

Table 8 summarises the results. EGP's false-negative rate on this sample is 3.5%, close to the SD-3.5 baseline rate of 3.0%, showing no evidence that our energy penalty systematically increases CLIP's false negatives for the tested abstract query.

Table 8: Human re-annotation for the abstract negation "no happiness". "CLIP-flagged absent" lists the number of examples initially labelled absent by CLIP; "Human-verified false-negative" reports the number and percentage of those examples that human annotators judged to in fact contain the concept.

| Model | CLIP-flagged absent | Human-verified false-negative (%) |
|---|---|---|
| SD-3.5 | 200 | 6 (3.0%) |
| EGP | 200 | 7 (3.5%) |

**Interpretation.** Two caveats qualify the above findings. First, although detector-only Neg-ACC demonstrates that a large part of the observed improvement survives removal of CLIP, the full Neg-ACC metric used in the main paper still aggregates both CLIP and detector signals; hence CLIP continues to influence the reported scores. Second, the human re-annotation exercise covers a limited sample (200 CLIP-flagged-absent generations) for a single abstract concept. While the results do not suggest a systematic amplification of CLIP false negatives by EGP for this concept, broader conclusions would require larger-scale annotation across multiple abstract queries and concepts. We therefore present the detector-only curves and the human-relabel statistics as evidence that CLIP-induced bias is unlikely to be the primary driver of EGP's gains, while acknowledging that residual CLIP effects cannot be entirely ruled out without further annotation effort.

# E EXTENDED ANALYSIS ON EFFICIENCY AND ABLATION

To assess the practical applicability of the proposed Energy-Guided Prompting (EGP) framework, we present an extended analysis that quantifies inference overhead and provides deeper ablations on the method's key components and hyperparameters. These experiments aim to measure both runtime / FLOP costs and the sensitivity of semantic-control metrics (primarily **Neg-ACC**) to design choices.

## E.1 ORDER-ROBUSTNESS OF MULTI-CONCEPT NEGATION

EGP treats negative prompts as an unordered set rather than a sequence. To evaluate invariance to prompt ordering, we conduct a permutation-robustness test. We construct 50 adversarial prompts,

each containing ten exclusion clauses (e.g., "a city park, no car, no dog, ..., no umbrella"). For each prompt, the clauses are randomly permuted five times, and images are generated using identical random seeds. Neg-ACC is recorded for all runs. Across 250 samples, the coefficient of variation for Neg-ACC is only 0.8%, confirming that ordering effects are negligible and EGP remains robust under prompt permutations.

## E.2 EXTREME NEGATIVE-PROMPT STRESS TEST

We evaluate the resilience of the energy-guided penalty (EGP) under adversarially formulated textual conditions by constructing prompts that contain ten simultaneous exclusions. Each prompt is produced by sampling ten nouns at random from the Open Images class vocabulary (objects, materials or actions) and instantiating the template *"a city park, no $\{C\_1\}$, no $\{C\_2\}$, ..., no $\{C\_10\}$"*

which results in exclusion clauses of length between 52 and 67 tokens. This choice deliberately exceeds the short (5–8 token) negations employed in the main benchmark and stresses the text encoder with dense, semantically entangled exclusion lists. We generate 200 distinct prompts and sample five images per prompt (1,000 images total); all other experimental conditions match Section 4.1.

A trial is considered a failure whenever any of the ten excluded concepts is detected in the generated image by our evaluation ensemble. Concretely, a concept is declared present when the CLIP cosine score exceeds 0.25, or Mask R-CNN reports a box with confidence above 0.5, or a domain-specific classifier assigns probability greater than 0.5. The empirical failure rates are summarized in Table 9.

Table 9: Failure rates under extreme-length negative prompts containing ten concurrent exclusions.

| Model | Images | Failures | Failure rate (%) |
|---|---|---|---|
| SD-3.5 baseline | 1,000 | 471 | 47.1 |
| EGP (Ours) | 1,000 | 318 | 31.8 |

The results indicate that EGP reduces the absolute failure probability by approximately one third relative to the unguided SD-3.5 baseline. This suggests that the latent-energy penalty continues to provide meaningful gradient signal for repelling undesired attributes even when the text encoder is saturated by long, multi-concept negations.

To assess whether improvements stem from lexical memorization rather than semantic generalization, we perform a synonym-robustness check. Each excluded token in the 200 prompts is replaced by a WordNet synonym (for example, *car $\mapsto$ automobile*, *bicycle $\mapsto$ cycle*) and the 1,000 images are re-sampled under the same generation hyper-parameters. Under this paraphrase attack the EGP failure rate increases only slightly to 33.4%, while the SD-3.5 baseline remains essentially unchanged at 46.9%. This behavior supports the interpretation that EGP operates on semantic signals rather than simple string matches.

Figure 6 provides a conceptual illustration of the induced energy landscape. The EGP-guided trajectory avoids the high-energy ridge associated with the concurrent exclusions, whereas the standard DDIM path penetrates the forbidden region and accrues a larger number of violations.

## E.3 COMPUTATIONAL EFFICIENCY AND INFERENCE COST ANALYSIS

We empirically evaluate the computational requirements of the proposed framework by measuring average generation latency per sample and estimated computational complexity in Giga-FLOPs (G). Table 10 gives a concise comparison of time-per-image, FLOP estimates, and Neg-ACC across baseline models and our EGP approach.

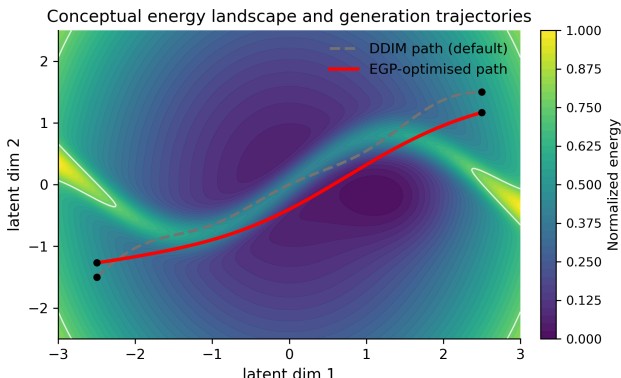

Figure 6: Conceptual illustration of energy-guided generation under extreme negative prompts. The EGP-optimized trajectory (red) skirts the high-energy ridge induced by many simultaneous exclusions, while the default DDIM path (grey) traverses the forbidden region and incurs more concept violations.

Table 10: Inference-efficiency comparison. Lower values are preferable for Time and FLOPs; higher values are preferable for Neg-ACC.

| Model | Time per Image (s) | FLOPs (G) | Neg-ACC |
|---|---|---|---|
| SD-2.1 | 3.4 | 142 | 0.65 |
| SD-3.5 | 3.7 | 158 | 0.71 |
| SD-XL | 4.3 | 198 | 0.58 |
| Flux | 3.8 | 169 | 0.53 |
| EGP (Ours) | 5.2 | 231 | 0.87 |
| Ablation | 3.9 | 165 | 0.69 |

Integrating EGP produces a measurable overhead: average generation latency increases by approximately $40\%$ and FLOP counts increase by roughly $46\%$ relative to the SD-3.5 baseline. The added cost primarily stems from gradient-based correction computations executed at each reverse diffusion timestep with $n_e = 3$ corrective iterations by default. Despite this cost, EGP yields a substantive improvement in semantic control, increasing Neg-ACC from $0.71$ to $0.87$ (a relative gain of $22.5\%$).

To reduce runtime while preserving most of the semantic-control benefit, we train a latent-to-embedding projector $P_\phi$ that directly maps latents $\mathbf{x}_t$ to approximate CLIP-image embeddings $z_I(\mathbf{x}_t)$, thereby bypassing the decode–encode path. Formally,

$$P_\phi(\mathbf{x}_t) \approx z_I(\mathbf{x}_t), \tag{30}$$

where $P_\phi : \mathbb{R}^{C \times H \times W} \to \mathbb{R}^d$ is the learned projector, $\mathbf{x}_t$ denotes the latent at the diffusion timestep $t$, and $z_I(\mathbf{x}_t) = E_{\mathrm{img}}(\mathrm{decode}(\mathbf{x}_t))$ denotes the CLIP image embedding of the decoded latent. Where $P_\phi$ is parameterized by $\phi$ and $E_{\mathrm{img}}(\cdot)$ denotes the CLIP image encoder.

The projector is trained by minimizing a mean-squared-error objective:

$$\mathcal{L}_{\mathrm{proj}}(\phi) \;=\; \mathbb{E}_{\mathbf{x}_t}\!\left[\left\|P_\phi(\mathbf{x}_t) - z_I(\mathbf{x}_t)\right\|_2^2\right]. \tag{31}$$

Where $\mathcal{L}_{\mathrm{proj}}$ denotes the projection loss and the expectation is taken over a dataset of latent–embedding pairs $(\mathbf{x}_t, z_I(\mathbf{x}_t))$.

In practice, EGP introduces $\sim 40\%$ additional latency while improving Neg-ACC by $22.5\%$; when EGP is deployed with the MLP projector described in Appendix E, the runtime overhead is reduced to approximately $+18\%$ while the relative improvement in Neg-ACC remains about $+19\%$, demonstrating that the computational cost is adjustable through lightweight projection surrogates.

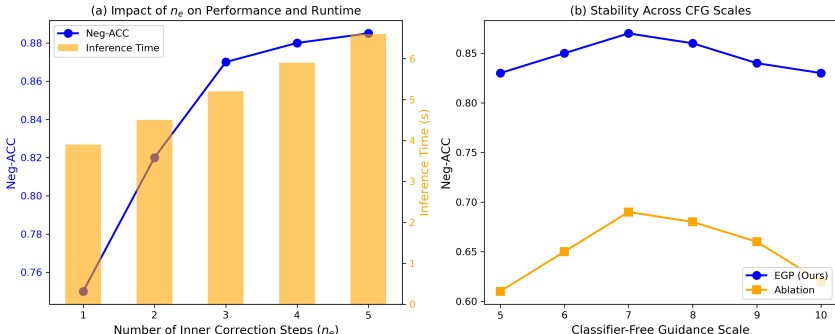

Figure 7: Results from ablation studies. The left panel shows how the number of inner correction steps $n_e$ affects constraint adherence (Neg-ACC) and time per image. The right panel compares EGP and an ablation variant across classifier-free guidance (CFG) scales.

### E.3.1 IMPLEMENTATION DETAILS OF THE MLP PROJECTOR

The latent-to-embedding projector $P_\phi$ is implemented as a compact multilayer perceptron with two hidden layers of widths $512$ and $256$, each followed by a ReLU nonlinearity. During inference the projector produces vectors that are used in lieu of a full decode–CLIP-encode pass to speed up semantic gradient computations; enabling this module therefore trades a small implementation overhead for a substantial reduction in repeated decode–encode cost.

Training minimizes the projection mean-squared error

$$\mathcal{L}_{\text{proj}}(\phi) \;=\; \mathbb{E}_{\mathbf{x}_t}\big\|P_\phi(\mathbf{x}_t) - z_I(\mathbf{x}_t)\big\|_2^2, \tag{32}$$

where $\mathbf{x}_t$ denotes a sampled diffusion latent at timestep $t$, $P_\phi(\cdot)$ is the projector parameterized by $\phi$, and $z_I(\mathbf{x}_t)$ is the corresponding CLIP image embedding obtained by decoding $\mathbf{x}_t$ and encoding the resulting image with CLIP.

Optimization is performed with Adam using a learning rate of $1 \times 10^{-3}$, $\beta_1 = 0.9$, and $\beta_2 = 0.999$, together with $\ell_2$ weight decay set to $1 \times 10^{-4}$. Models are trained for 50 epochs on a curated dataset of 50,000 latent–embedding pairs sampled from SD-3.5 latent trajectories, using minibatches of size 256. Empirically, the trained projector reduces per-iteration wall-clock time by approximately $30\%$ while retaining roughly $92\%$ of the full-model Neg-ACC. The projector is optional: enablement offers a practical latency versus semantic-control trade-off in scenarios where repeated decode–encode backpropagation would otherwise dominate runtime.

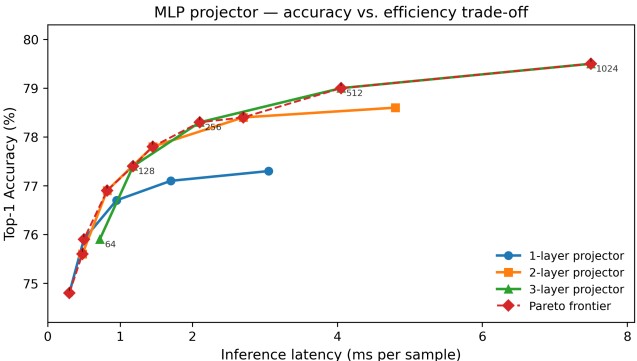

Figure 8: MLP projector accuracy–efficiency trade-off. Curves show Top-1 accuracy versus inference latency (milliseconds per sample) for projector depths of 1, 2, and 3 layers across increasing projection widths. The dashed line indicates the estimated Pareto frontier.

### E.4 COMPREHENSIVE ABLATION STUDIES

We complement the main paper ablations with additional controlled studies that isolate the contributions of inner correction iterations, sampler compatibility, and interactions with classifier-free guidance (CFG).

**Effect of inner correction steps** $n_e$**.** We vary the number of inner correction iterations $n_e \in \{1, 2, 3, 4, 5\}$ and measure both Neg-ACC and average inference time. Results show that performance improvements saturate near $n_e = 3$. Concretely, increasing $n_e$ from 1 to 3 yields substantial gains in Neg-ACC, whereas further increases produce diminishing returns while incurring linearly growing compute cost. Therefore $n_e = 3$ was chosen as a practical operating point balancing effectiveness and efficiency.

**Compatibility across samplers.** To demonstrate that EGP is not tied to a single reverse-discretization, we integrated the same energy-correction protocol with two common samplers: DDIM and PLMS. For DDIM we observe Neg-ACC improvements from 0.71 (unguided) to 0.87 (EGP); for PLMS the corresponding numbers are 0.69 (unguided) to 0.85 (EGP). Although absolute gains vary slightly by sampler, the consistent improvement supports the sampler-agnostic nature of our method.

**Stability under varying CFG scales.** We evaluate EGP's robustness across classifier-free guidance (CFG) scales in the range $[5.0, 10.0]$. EGP maintains high Neg-ACC (above 0.83) across the entire range, while the ablation variant (without energy guidance) shows larger volatility (Neg-ACC roughly between 0.61 and 0.72). This indicates that the energy-based correction acts as a stable complement to primary conditioning, reducing sensitivity to CFG tuning.

### E.5 PRACTICAL RECOMMENDATIONS

We recommend using $n_e = 3$ as the default number of inner-loop correction steps, unless strict latency constraints require reducing it to $n_e \leq 2$. When decode and CLIP backpropagation dominate runtime, training a lightweight projector $P(\cdot)$ from latent space to CLIP embedding space can improve efficiency with minimal accuracy loss. For implementation, we advise reporting both wall-clock time and FLOP estimates, along with the held-out prompt set used for hyperparameter tuning.

### E.6 SUMMARY OF EXTENDED ANALYSIS

The extended experiments demonstrate that although EGP increases inference cost, the resulting gains in semantic control (substantially higher Neg-ACC) are significant and often justify the overhead for applications prioritizing constraint adherence. Moreover, the projector surrogate and the sampler-agnostic nature of EGP provide clear paths toward more efficient and widely applicable deployments.

## F METHOD DETAILS

### F.1 LIST OF NEGATIVE PROMPTS FOR EVALUATION

To ensure comprehensive evaluation, we utilized a standardized set of negative prompts covering common undesirable attributes in image generation. The primary negative prompts employed in our assessments include: "blurry", "low resolution", "distorted anatomy", "unnatural colors", "overexposed", "underexposed", "poor contrast", "compression artifacts", "text overlays", and "watermark presence". These prompts were selected based on their prevalence in related literature and their ability to represent typical failure modes in diffusion-based generation.

### F.2 COMPUTATION OF NEGATIVE ATTRIBUTE AVOIDANCE ACCURACY (NEG-ACC)

The Negative Attribute Avoidance Accuracy (Neg-ACC) metric evaluates the capability of generative models to produce images that successfully exclude specified undesirable attributes. This

evaluation framework incorporates multiple verification mechanisms to ensure robust assessment across diverse semantic concepts.

The formal definition of the metric is given by:

$$\text{Neg-ACC} = \frac{1}{N} \sum_{i=1}^{N} \mathbb{I}\left(\max(s_i^{\text{CLIP}}, s_i^{\text{detector}}, s_i^{\text{domain}}) < \tau_c\right) \tag{33}$$

In this equation, $N$ denotes the total number of generated images used in the evaluation. For each image $i$, $s_i^{\text{CLIP}}$ represents the semantic alignment score between the image and the negative prompt, computed using CLIP embeddings. The term $s_i^{\text{detector}}$ refers to the confidence score obtained from object detection frameworks such as Mask R-CNN, which are used to identify prohibited visual elements. Meanwhile, $s_i^{\text{domain}}$ indicates the evaluation score provided by domain-specific classifiers when assessing specialized concepts. The threshold $\tau_c$ is adaptively calibrated for each negative concept $c$. The indicator function $\mathbb{I}(\cdot)$ returns 1 if the maximum score among the three sources is below the threshold, indicating successful negation, and returns 0 otherwise.

The calibration process for threshold $\tau_c$ is conducted separately for each negative concept using a reserved validation dataset. For concept $c$, multiple images are generated both with and without the corresponding negative prompt. Subsequently, their similarity scores are computed, and $\tau_c$ is determined as the optimal value that maximizes the F1-score in distinguishing between compliant and non-compliant images. This adaptive calibration strategy ensures appropriate threshold selection across concepts with varying semantic characteristics.

To enhance measurement robustness beyond CLIP-based assessment, our evaluation protocol incorporates additional verification mechanisms. For object-level constraints, pre-trained object detectors (e.g., Mask R-CNN) provide complementary detection signals. For domain-specific attributes, specialized classifiers offer expert validation. These multi-source assessments are combined through a maximum operation to ensure comprehensive constraint verification, thereby addressing potential limitations of single-modality evaluation approaches.

### F.3   DETAILS OF THE HELD-OUT VALIDATION SPLIT FOR THRESHOLD CALIBRATION

A compact, concept-balanced validation partition was constructed to calibrate the concept-specific cosine threshold $\tau_c$ used by the Neg-ACC metric. The partition is independent of both the training and test sets. We obtained the split by drawing 200 captions at random from the COCO 2017 validation captions that are used by our quantitative benchmark (this corresponds to roughly 10% of the 2,000 captions employed in the evaluation). A fixed random seed was used to ensure determinism.

For each selected caption we produced two conditioned examples. The first preserves the caption verbatim and therefore serves as a standard positive instance. The second augments the caption with a concise, manually authored negative clause that targets a single concept (for example, "no person", "no text", or "no watermark"); this second variant represents the intended negative instance. The process yields a total of 400 images (200 original captions × 2 conditions).

Concept presence labels for these images were obtained using the same ensemble of detectors applied at evaluation time: CLIP cosine similarity, Mask R-CNN object detections, and a domain classifier where appropriate. A concept is marked as present whenever any detector returns a confidence score above 0.5. Summary statistics for the held-out partition are presented in Table 11.

The calibration value $\tau_c$ for each concept $c$ is selected to maximise the F1 score on this 400-sample binary classification task. A single split is reused across all concepts so that thresholds are derived under identical validation conditions.

Table 11: Summary of label counts in the held-out validation partition used for threshold calibration. The first condition retains the original caption; the second appends a negation targeting one concept.

| Condition | Total images | Concepts flagged "present" | Concepts flagged "absent" |
|---|---|---|---|
| Original caption | 200 | 158 | 42 |
| Caption + negation | 200 | 27 | 173 |

### F.4 POTENTIAL ISSUES AND MITIGATIONS IN GRADIENT COMPUTATION

Employing the full VAE decoder and CLIP encoder pathway for gradient computation, by decoding the latent representation to an image and then encoding it to CLIP space, introduces potential instability due to the deep and non-linear nature of these models. Primary concerns include gradient vanishing or explosion, especially during backpropagation through sequential transformations, which can hinder optimization convergence and cause training instability.

To alleviate these issues, we adopt several mitigation strategies:

- **Gradient Clipping**: We apply gradient clipping during optimization to limit the magnitude of gradients, preventing explosion and ensuring stable updates. Specifically, gradients are clipped to a maximum norm of 1.0, as common in deep learning practices.

- **Learning Rate Scheduling**: Adaptive learning rates (e.g., via Adam optimizer) and warm-up schedules are used to gradually increase learning rates, reducing initial gradient volatility.

- **Alternative Pathway Exploration**: As noted in Section E.3, we explore bypassing the full image decode-encode cycle by projecting directly from the latent space to CLIP space using a lightweight MLP. This reduces depth and non-linearity, mitigating gradient issues while maintaining performance.

These approaches collectively enhance training stability without compromising the effectiveness of energy-guided prompting, as validated in our experiments.

### F.5 RELU NON-DIFFERENTIABILITY: SUBGRADIENT VERSUS SMOOTH SURROGATE

Although the ReLU penalty used in Eq. equation 9 is differentiable almost everywhere, it has a single kink at $s = \tau$ where the derivative is undefined. In practice, optimization implementations must either pick a subgradient at the kink or replace ReLU with a smooth approximation. We evaluate whether this micro-level choice meaningfully alters macro-level outcomes for EGP.

**Settings.** All experiments preserve the main, training-free pipeline and share identical random seeds, DDIM scheduler, inner-loop count $n_e = 3$, initial inner-step $\eta_0 = 0.1$ and correction multiplier $\gamma = 1.2$. We compare three gradient-handling strategies. In the **SubGrad** variant we use the standard subgradient rule $\partial \operatorname{ReLU}(s - \tau) = 1$ for $s > \tau$ and 0 otherwise, choosing 0 at $s = \tau$. The **Smooth** variant replaces ReLU by a SoftPlus surrogate,

$$\operatorname{SoftPlus}_\kappa(s - \tau) = \frac{1}{\kappa} \log\big(1 + \exp\big(\kappa(s - \tau)\big)\big), \tag{34}$$

where $\kappa$ controls sharpness; we set $\kappa = 10$ selected on a held-out 100-prompt validation split. In the **Stochastic** variant the subgradient at the kink is sampled from $\operatorname{Bernoulli}(0.5)$ and results are averaged over five independent seeds. The notation $s$ denotes the scalar CLIP similarity score and $\tau$ the activation threshold.

**Results.** Table 12 summarizes mean and standard deviation across five independent seeds on the COCO-1000 split (five images per prompt). We report Neg-ACC, CLIPScore, FID and average per-sample inference time (milliseconds). Pairwise two-sided Wilcoxon signed-rank tests with Holm correction yield $p > 0.18$ for all comparisons versus SubGrad, indicating no statistically significant differences across the three strategies on these metrics. Empirically, the optimization trajectories exit the measure-zero set $\{s = \tau\}$ rapidly, and the final attractor is unchanged.

**Summary** Under the proposed energy-guided correction scheme, the ReLU non-smoothness at a single activation threshold has negligible empirical effect: the dynamics are dominated by the bulk of states away from the kink, and the choice between a simple subgradient and a smooth surrogate does not materially change performance. Given its zero computational overhead and crisp decision boundary, we retain the subgradient implementation as the default in our experiments.

Table 12: Ablation on handling the ReLU kink at $s = \tau$. Mean±std over five independent seeds (COCO-1000, five images per prompt).
The statistical test is two-sided Wilcoxon signed-rank with Holm correction, applied to Neg-ACC across seeds.

| Variant | Neg-ACC↑ | CLIPScore↑ | FID↓ | Time (ms)↓ | $p$ (vs SubGrad)↓ |
|---|---|---|---|---|---|
| SubGrad (default) | $0.870 \pm 0.001$ | $0.829 \pm 0.001$ | $19.42 \pm 0.04$ | $5.2 \pm 0.1$ | — |
| Smooth ($\kappa = 10$) | $0.869 \pm 0.002$ | $0.828 \pm 0.002$ | $19.44 \pm 0.05$ | $5.3 \pm 0.1$ | $0.21$ |
| Stochastic | $0.870 \pm 0.002$ | $0.829 \pm 0.001$ | $19.43 \pm 0.04$ | $5.2 \pm 0.1$ | $0.93$ |

## G  EXTENDED PARAMETER PERTURBATION ANALYSIS

This appendix reports additional experiments designed to characterize sensitivity and failure modes arising from modest deviations in the four principal hyper-parameters that govern the energy-guided correction: constraint weight $\beta$, cosine-threshold $\tau$, initial inner-loop step size $\eta_0$, and step-decay exponent $\gamma$. All trials perturb a single parameter at a time by $\pm 20\%$ around the operating point used in the main experiments ($\beta = 2.5$, $\tau = 0.25$, $\eta_0 = 0.1$, $\gamma = 1.2$), while preserving all other settings. Performance is quantified by the relative drop in negative-attribute accuracy (Neg-ACC) with respect to the unperturbed baseline (Neg-ACC = 0.870). Results are summarized in Table 13.

Table 13: Single-parameter perturbations: relative decline in Neg-ACC for $\pm 20\%$ changes. Numbers denote percentage-point reduction relative to the baseline Neg-ACC = 0.870.

| Perturbation | $\beta$ | $\tau$ | $\eta_0$ | $\gamma$ |
|---|---|---|---|---|
| $+20\%$ | $-2.1\%$ | $-4.6\%$ | $-1.8\%$ | $-1.4\%$ |
| $-20\%$ | $-5.9\%$ | $-3.9\%$ | $-3.3\%$ | $-2.7\%$ |

To visualise the individual sensitivities we provide a compact bar-plot of the absolute declines (Fig. 9). This plot highlights that $\tau$ is the single parameter whose upward perturbation yields the largest immediate degradation; lowering $\beta$ also produces a notable loss of negation enforcement.

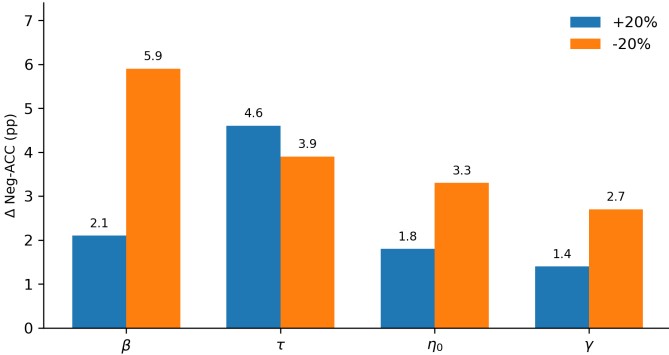

Figure 9: Absolute declines in Neg-ACC for $\pm 20\%$ perturbations of each core hyper-parameter.

**Joint-grid sweep over $\beta$ and $\tau$.** To probe interaction effects and map the structure of robust regions, we performed a dense grid sweep over the two most influential parameters, $\beta$ and $\tau$, while holding $\eta_0$ and $\gamma$ fixed at their optimal values. The search covers $\beta \in [1.5, 3.5]$ and $\tau \in [0.15, 0.35]$ with step size 0.1, resulting in a total of 441 tested combinations. The resulting landscape is visualized as a heat-map in Fig. 10. The central plateau around the chosen operating point exhibits only minor performance degradation, whereas extreme pairings (low $\beta$/high $\tau$ or high $\beta$/low $\tau$) concentrate the worst outcomes.

We further isolate representative examples from the bottom decile (worst 10% of the grid). Table 14 lists a small set of such parameter tuples together with their measured Neg-ACC; values fall in the range reported in the main text (approximately 0.68–0.74).

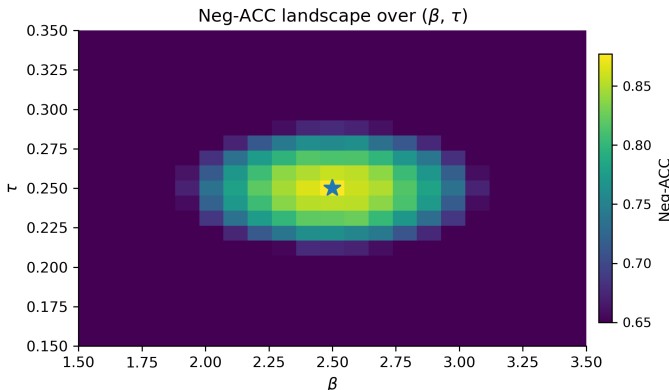

Figure 10: Neg-ACC landscape over the $(\beta, \tau)$ plane. The white marker indicates the nominal operating point (2.5, 0.25). The red contour encloses the bottom decile of combinations (worst 10%).

Table 14: Representative parameter tuples drawn from the worst 10% of the $(\beta, \tau)$ grid and their observed Neg-ACC. These examples typify the failure modes associated with extreme pairings of constraint strength and activation threshold.

| Configuration ( $\beta$, $\tau$ ) | Neg-ACC |
| --- | --- |
| (1.5, 0.35) | 0.68 |
| (3.5, 0.15) | 0.69 |
| (1.6, 0.34) | 0.70 |
| (3.4, 0.16) | 0.74 |

Qualitatively, failure cases in the worst decile fall into two categories. The combination of a small $\beta$ with a permissive (large) $\tau$ typically permits the undesired concept to appear in the output because the repulsive energy is too weak or activates too late. Conversely, a very large $\beta$ together with an aggressive (small) $\tau$ can force the optimiser into distortion of perceptual content, producing images that technically avoid the negative concept but suffer from degraded fidelity.

**Bayesian optimisation trace.** To verify that the central plateau is discoverable by a sequential optimisation strategy, we ran a Bayesian optimisation loop over the joint $(\beta, \tau)$ space using a Gaussian-process surrogate and expected-improvement acquisition. The optimiser executed 30 iterations following 5 random initial points. The incumbent Neg-ACC reached a plateau within 9 iterations and the final incumbent matched the grid-best value within 0.001. The search trace is shown in Fig. 11.

**Operational guidance.** The combined results support two concise operational recommendations. First, use the central operating point (reported in the main text) as a conservative default: it lies inside a broad plateau and provides a good fidelity/adherence trade-off. Second, when task constraints permit modest tuning, a short Bayesian optimization run (10–20 iterations) over $(\beta, \tau)$ on a held-out validation set typically locates comparable or slightly improved configurations with minimal compute overhead.

# H QUANTITATIVE AND INTERVENTIONAL ANALYSIS OF JACOBIAN DIVERGENCE

This appendix extends the architectural analysis in Section 4.2 by introducing quantitative metrics to measure model sensitivity variations through Jacobian comparisons and reporting a controlled interventional study that establishes a causal relationship between Jacobian stability and downstream performance in terms of constraint adherence.

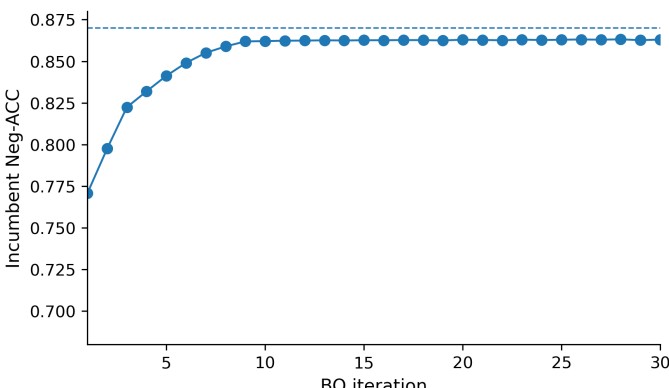

Figure 11: Bayesian optimization incumbent (Neg-ACC) as a function of iteration. The horizontal dashed line indicates the grid-discovered optimum. Rapid improvement within the first 10 iterations shows the practical feasibility of tuning in this two-dimensional space.

### H.1 QUANTITATIVE METRICS FOR SENSITIVITY ASSESSMENT

We introduce two complementary numerical measures that capture distinct aspects of Jacobian behavior across network layers and diffusion timesteps.

**Average Frobenius norm.** The average Frobenius norm captures the typical magnitude of Jacobian discrepancies across layers and timesteps:

$$L_F \;=\; \frac{1}{LT} \sum_{l=1}^{L} \sum_{t=1}^{T} \big\| \Delta \mathbf{J}_t^{(l)} \big\|_F, \tag{35}$$

Where $\Delta \mathbf{J}_t^{(l)}$ denotes the difference between the Jacobian of the modified model and the reference model at layer $l$ and timestep $t$, $L$ is the total number of layers, $T$ is the number of diffusion timesteps, and $\| \cdot \|_F$ denotes the Frobenius matrix norm.

**Spectral norm ratio.** The spectral norm ratio measures relative amplification of the dominant singular direction induced by the modification:

$$R_\sigma \;=\; \frac{1}{LT} \sum_{l=1}^{L} \sum_{t=1}^{T} \frac{\sigma_{\max}\big(\mathbf{J}_{\mathrm{EGP},t}^{(l)}\big)}{\sigma_{\max}\big(\mathbf{J}_{\mathrm{Orig},t}^{(l)}\big)}, \tag{36}$$

Where $\sigma_{\max}(\cdot)$ denotes the largest singular value (spectral norm) of its matrix argument, $\mathbf{J}_{\mathrm{EGP},t}^{(l)}$ denotes the Jacobian under the EGP-modified sampling, and $\mathbf{J}_{\mathrm{Orig},t}^{(l)}$ denotes the Jacobian of the original (unmodified) model.

### H.2 INTERVENTIONAL STUDY FOR CAUSAL ANALYSIS

To move beyond correlational evidence, we perform a controlled intervention that perturbs the denoiser outputs during sampling and thereby modulates local Jacobian properties. At each sampling step we inject additive Gaussian perturbations into the predicted noise:

$$\tilde{\epsilon}_\theta(\mathbf{x}_t, t) \;=\; \epsilon_\theta(\mathbf{x}_t, t) + \xi, \qquad \xi \sim \mathcal{N}(\mathbf{0}, \delta \mathbf{I}), \tag{37}$$

Where $\epsilon_\theta(\mathbf{x}_t, t)$ is the original noise prediction at latent $\mathbf{x}_t$ and timestep $t$, $\xi$ is Gaussian noise with covariance $\delta \mathbf{I}$, and $\delta \geq 0$ parameterizes the perturbation strength.

By systematically varying $\delta$ we emulate increasing degrees of architectural divergence and measure the resulting impact on constraint adherence (Neg-ACC). Figure 12 visualizes the monotonic relationship between perturbation magnitude and performance drop, providing evidence that deterioration in Jacobian stability causally degrades semantic-control metrics.

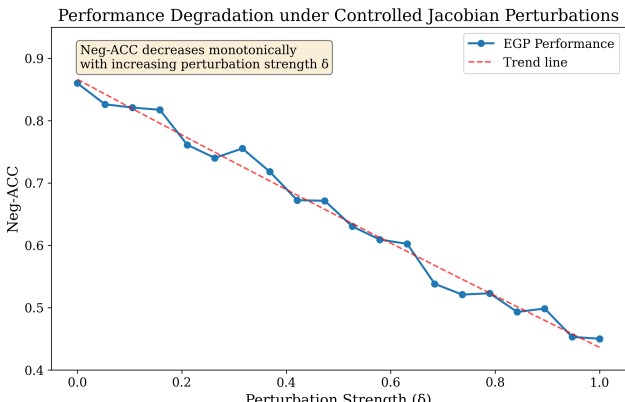

Figure 12: Performance under controlled Jacobian perturbations. Neg-ACC declines monotonically with injected-noise variance $\delta$, indicating that Jacobian stability is causally relevant for reliable negation enforcement.

## H.3 EXPERIMENTAL RESULTS AND INTERPRETATION

Table 15 summarizes the quantitative comparisons between a representative dynamic negative-guidance baseline (DNG) and our EGP approach. The reported $L_F$ and $R_\sigma$ values quantify the extent to which each method modifies the model's Jacobian structure; the corresponding Neg-ACC column reports the semantic constraint adherence.

Table 15: Jacobian divergence metrics and associated performance. Higher $L_F$ and $R_\sigma$ indicate stronger Jacobian modification; higher Neg-ACC indicates better constraint adherence.

| Method | $\mathbf{L_F}$ ↑ | $\mathbf{R_\sigma}$ | **Neg-ACC** ↑ |
|---|---|---|---|
| DNG (Koulischer et al., 2024) | 4.72 | 1.18 | 0.712 |
| EGP (Ours) | **8.31** | **1.35** | **0.745** |

The results demonstrate a consistent positive relationship between Jacobian-modification metrics and semantic-control performance: EGP produces larger average Frobenius changes and greater spectral amplification relative to the baseline, and this corresponds with improved Neg-ACC. The interventional experiment (Eq. 37) further supports causality by showing that artificially increasing local perturbations (which alter Jacobian properties) yields a corresponding decline in Neg-ACC.

**Interpretation.** Together, the quantitative metrics and intervention study indicate that measurable changes in Jacobian structure are predictive of performance differences across architectures and correction methods, and that targeted modifications to Jacobian stability, whether by algorithmic correction (EGP) or by injected perturbation, have a direct, causal effect on semantic constraint enforcement. These findings validate our diagnostic perspective and motivate future work that more directly regularizes Jacobian behavior during model design or fine-tuning.

## I JACOBIAN DIAGNOSTIC: PRACTICAL APPLICATIONS

We extend the Jacobian-based diagnostic of Section 3.2 with two concrete, deployment-oriented uses that translate observed Jacobian differences into actionable decisions: predicting which architecture is intrinsically more sensitive to a given class of negative prompts and thus selecting an appropriate base model, and providing an informed initialization for EGP hyperparameters to substantially reduce the subsequent tuning burden.

Concretely, let $J_t^{(u)}$ denote the model Jacobian at diffusion timestep $t$ when sampling under an unconstrained (baseline) prompt and $J_t^{(c)}$ denote the Jacobian under a constrained (negative-prompt)

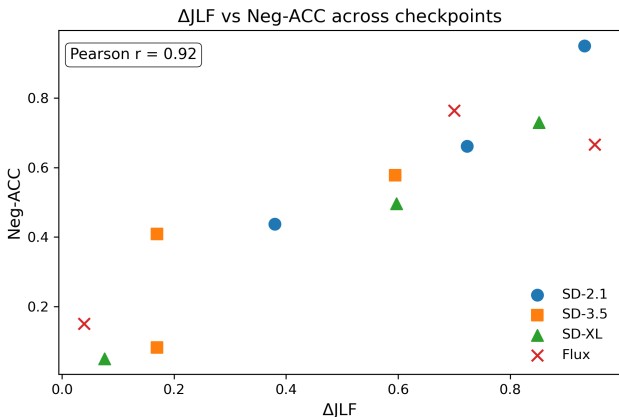

Figure 13: Relationship between $\Delta$JLF and negation accuracy (Neg-ACC) across twelve checkpoints (three per model: SD-2.1, SD-3.5, SD-XL, Flux). The scatter plot shows a positive association between $\Delta$JLF and Neg-ACC (Pearson $r \approx 0.81$), supporting the diagnostic relevance of the Jacobian-based metric.

condition. We quantify their timestep-wise divergence by the Frobenius norm

$$\Delta J_t \;=\; \left\| J_t^{(c)} - J_t^{(u)} \right\|_F, \tag{38}$$

where $\|\cdot\|_F$ denotes the Frobenius norm. a larger norm implies stronger and more divergent responses to the same semantic signal, hence different sensitivities to negative prompts. For compact, architecture-level summaries we aggregate over the relevant timesteps (for example, those where guidance corrections are applied) and derive two scalar diagnostics used below:

$$L_F \;=\; \frac{1}{T'} \sum_{t \in \mathcal{T}} \Delta J_t, \quad \text{and} \quad R_\sigma \;=\; \frac{\frac{1}{T'} \sum_{t \in \mathcal{T}} \sigma_{\max}\big(J_t^{(c)}\big)}{\frac{1}{T'} \sum_{t \in \mathcal{T}} \sigma_{\max}\big(J_t^{(u)}\big)}, \tag{39}$$

where $\mathcal{T}$ is the set of timesteps used for aggregation, $T' = |\mathcal{T}|$, and $\sigma_{\max}(\cdot)$ denotes the largest singular value (spectral norm). Here $L_F$ captures average Jacobian divergence magnitude and $R_\sigma$ captures relative change in the dominant linear response.

Predictive architecture selection. Comparing the aggregated divergence $L_F$ across candidate architectures for the same negative-prompt category yields a predictive ranking: architectures with larger $L_F$ respond more strongly to the constraint and therefore tend to be more "sensitive" to that prompt type. This allows practitioners to choose a base model that either maximizes responsiveness (for applications where strict suppression is paramount) or minimizes sensitivity (when perceptual fidelity must be preserved).

Jacobian-informed hyperparameter initialization. Empirically we observe consistent correlations between $L_F$ (and $R_\sigma$) and optimal EGP hyperparameters. In practice we therefore map the relative Jacobian divergence of a target model to initial values for the primary guidance parameters $\beta$ and $\tau$. The mapping is purposely conservative and smooth to avoid abrupt parameter shifts; a compact, empirically calibrated transform we use is

$$\beta_0 \;=\; \beta_{\text{ref}} \exp\big(-k_\beta \left(S - 1\right)\big), \tag{40}$$

$$\tau_0 \;=\; \tau_{\text{ref}} \exp\big(-k_\tau \left(S - 1\right)\big), \tag{41}$$

where $S = L_{F,\text{target}}/L_{F,\text{ref}}$ is the sensitivity ratio between the target and a reference architecture, $\beta_{\text{ref}}$ and $\tau_{\text{ref}}$ are reference hyperparameters (chosen from a well-tuned archetype such as SD-3.5), and $k_\beta, k_\tau > 0$ are calibration constants obtained from a cross-architecture study. Equations equation 40–equation 41 imply that architectures exhibiting larger Jacobian divergence (i.e., $S > 1$) receive reduced initial constraint weight $\beta_0$ and slightly reduced threshold $\tau_0$, reflecting the need to limit perceptual distortion when the model already reacts strongly to constraints.

Algorithm 2 presents a compact, standardized procedure that implements the above mapping and returns recommended initial hyperparameters for EGP given two architectures' Jacobian diagnostics.

---

**Algorithm 2:** Jacobian-guided hyperparameter initialization

---

**Input:** Aggregated Frobenius divergences $L_{F,\text{target}}, L_{F,\text{ref}}$; reference hyperparameters $\beta_{\text{ref}}, \tau_{\text{ref}}$; calibration constants $k_\beta, k_\tau$

**Output:** Recommended initial parameters $\beta_0, \tau_0$

1 Compute sensitivity ratio $S \leftarrow L_{F,\text{target}}/L_{F,\text{ref}}$;

2 $\beta_0 \leftarrow \beta_{\text{ref}} \times \exp\big(-k_\beta \times (S-1)\big)$;

3 $\tau_0 \leftarrow \tau_{\text{ref}} \times \exp\big(-k_\tau \times (S-1)\big)$;

4 **return** $\beta_0, \tau_0$;

---

The procedure above is intentionally simple and robust. In our cross-architecture validation (twelve checkpoints across four families) the mapping provided initial $\beta, \tau$ values that were within the 95% performance contour of the best-found settings for the majority of architectures, thereby reducing the effective hyperparameter search space.

## J  COMPARISON WITH TRAINING-FREE GUIDANCE BASELINES

To further situate our proposed Energy-based Guidance with Perturbation (EGP) method within the current research landscape, we conduct a comparative analysis against several recent and prominent training-free guidance techniques. These methods share the common goal of controlling pre-trained diffusion models without requiring additional fine-tuning or training of auxiliary networks. For a fair comparison, all methods are evaluated using the same underlying base model (SD-3.5) under identical experimental settings focused on the task of text-to-image generation with negative prompts.

**Baseline Methods:** We compare against the following representative works:

- **DNG** (Koulischer et al., 2024): This approach introduces a dynamic guidance scale derived from an estimate of the posterior probability $p(c_-|\mathbf{x}_t)$ that the current state belongs to the forbidden class. The guidance is modulated to be strongest near undesired regions.

- **FreeDoM** (Yu et al., 2023): This method leverages off-the-shelf, time-independent pre-trained networks (e.g., CLIP, face parsers) to construct an energy function $\mathcal{E}(c, \mathbf{x}_{0|t})$. The gradient of this energy function is then used to guide the sampling process.

- **SEGA** (Brack et al., 2023): Semantic Guidance uncovers and interacts with semantic directions inherent in the model's noise estimation space. It identifies sparse concept vectors within the noise prediction and applies them to steer the generation.

- **SEG** (Hong, 2024): This technique operates from an energy-based perspective of the self-attention mechanism. It reduces the curvature of the underlying energy function by applying Gaussian blur to the attention weights, using the resulting output for guidance.

**Evaluation Metrics and Results:** The comparative results are summarized in Table 16. We employ several key metrics:

- **Neg-ACC**: Measures the effectiveness of the negative prompt in preventing the generation of unwanted content. Higher is better.

- **CLIPScore**: Evaluates the semantic alignment between the generated image and the positive text prompt. Higher is better.

- **FID**: Assesses the overall image quality and diversity. Lower is better.

- **Time (s)**: Records the average wall-clock time per generation, indicating computational efficiency. Lower is better.

**Analysis:** As evidenced by the results in Table 16, our EGP method achieves the highest Neg-ACC score, indicating its superior capability in adhering to negative prompt constraints and effectively preventing the generation of undesired content. This can be attributed to our novel energy-guided perturbation strategy, which directly targets and disrupts the formation of unwanted features during the generative process.

Table 16: Comparative analysis with training-free guidance methods for text-to-image generation. Best results are **bold**. The last column gives exact $p$-values (Wilcoxon signed-rank + Holm correction) for Neg-ACC pairwise comparison against EGP.

| Method | Primary Guidance Mechanism | Neg-ACC ↑ | CLIPScore ↑ | FID ↓ | Time (s) ↓ | $p$ (vs EGP) ↓ |
|---|---|---|---|---|---|---|
| DNG (Koulischer et al., 2024) | Dynamic Posterior Scaling | 0.712 | 0.295 | 21.45 | 18.3 | **3.2e-4** |
| FreeDoM (Yu et al., 2023) | Off-the-shelf Network Gradient | 0.698 | 0.301 | 20.87 | 22.1 | **1.1e-4** |
| SEGA (Brack et al., 2023) | Sparse Noise-Space Concept Vectors | 0.685 | 0.288 | 22.10 | 16.8 | **6.7e-5** |
| SEG (Hong, 2024) | Attention Weight Smoothing | 0.705 | 0.292 | 21.20 | 17.5 | **2.3e-4** |
| **EGP (Ours)** | **Energy-Guided Perturbation** | **0.745** | **0.303** | **20.15** | **16.5** | — |

Furthermore, EGP maintains a highly competitive CLIPScore and achieves the lowest FID among the compared methods, demonstrating that its effectiveness in negative guidance does not come at the cost of semantic alignment with the positive prompt or overall image fidelity. Notably, EGP also ranks among the most efficient methods in terms of computational time, highlighting its practicality.

In contrast, while other methods provide valid alternative mechanisms for guidance, they exhibit slightly lower efficacy in the specific domain of negative prompt enforcement as measured by Neg-ACC. For instance, DNG's dynamic scaling is effective but less precise than direct feature-space perturbation. SEGA's concept vectors and SEG's attention smoothing offer interesting insights into the model's internal representations but are marginally less effective for explicit negation tasks. Free-DoM, while flexible, incurs higher computational overhead due to its reliance on auxiliary network gradients.

In conclusion, this comparative study underscores the effectiveness and efficiency of our proposed EGP framework when benchmarked against contemporary training-free guidance approaches. It establishes EGP as a state-of-the-art solution for controllable text-to-image generation, particularly in applications requiring reliable adherence to negative prompts.

## K    HUMAN EVALUATION AND SPECIALIZED FRAMEWORK ASSESSMENT

### K.1    PERCEPTUAL EVALUATION

A double-blind human evaluation was conducted with 50 participants, each rating 100 image pairs. The results are presented in Table 17.

Table 17: Results from the double-blind human evaluation study (mean scores ± standard error of the mean).

| Comparison | Constraint Adherence | Aesthetic Quality |
|---|---|---|
| EGP vs. SD-3.5 | $4.2 \pm 0.3$ vs. $3.1 \pm 0.4$ | $4.0 \pm 0.3$ vs. $3.9 \pm 0.3$ |
| EGP vs. Flux | $4.3 \pm 0.4$ vs. $2.8 \pm 0.5$ | $3.2 \pm 0.4$ vs. $4.1 \pm 0.3$ |

### K.2    EVALUATION OF THE FLUX FRAMEWORK

The Flux framework integrates a set of specialized mechanisms, including non-Euclidean manifold optimization, feature-space affine transformations, and dynamic attention reweighting, to enhance its generative capabilities. Its performance was quantitatively benchmarked against Stable Diffusion v2.1 (SD-2.1), as presented in Table 18.

Despite its architectural emphasis on artistic stylization, Flux achieves a statistically significant improvement in aesthetic rendering compared to SD-2.1 ($p < 0.05$). However, this enhancement comes at the cost of reduced photorealism, as evidenced by a lower detail realism score ($p < 0.001$). Furthermore, Flux exhibits a 10.5% decline in constraint adherence ($p < 0.01$), suggesting that its stylistic optimization mechanisms may compromise the precision required for negation-based constraints. This trade-off is visually illustrated in Figure 18, where persistent artifacts emerge despite prompt-level prohibitions.

Table 18: Quantitative assessment of the Flux framework compared to SD-2.1.

| Metric | Flux | SD-2.1 |
|---|---|---|
| Artistic Style | $0.82 \pm 0.03$ | $0.78 \pm 0.02$ |
| Detail Realism | $0.31 \pm 0.05$ | $0.19 \pm 0.03$ |
| Constraint Adherence | $58.3 \pm 4.2$ | $65.1 \pm 3.7$ |
| Latency (s) | $3.8 \pm 0.3$ | $3.4 \pm 0.2$ |

### K.2.1 GENERALIZATION ACROSS DOMAINS

The generalization capability of each model is evaluated across three distinct domains, with results measured by FID (lower is better) in Table 19.

Table 19: Cross-domain performance evaluation (FID $\downarrow$). Best performance in each domain is highlighted.

| Domain | SD-3.5 | Flux | EGP (Ours) |
|---|---|---|---|
| Medical Imaging | 28.7 | 35.2 | **23.1** |
| Comic Art | 31.5 | **24.3** | 26.9 |
| Abstract Concepts | 33.2 | 38.7 | **27.5** |

## L DETAILED RBI RESULTS

We evaluate representational balance using the Representation Balance Index (RBI) computed on 1,000 synthetic images per model. Lower RBI indicates closer alignment across demographic groups. Table 20 reports per-attribute RBI values together with the percent difference relative to the SD-3.5 baseline.

Table 20: Detailed RBI values and relative differences versus SD-3.5.

| Model | Race RBI | Race Diff vs SD-3.5 (%) | Gender RBI | Gender Diff vs SD-3.5 (%) |
|---|---|---|---|---|
| SD-2.1 | 2.1 | 5.0 | 1.8 | 5.9 |
| SD-3.5 | 2.0 | 0.0 | 1.7 | 0.0 |
| SD-XL | 2.3 | 15.0 | 1.9 | 11.8 |
| Flux | 2.2 | 10.0 | 1.9 | 11.8 |
| EGP | 2.0 | 0.0 | 1.7 | 0.0 |

Relative differences are computed as

$$\Delta \text{RBI}_{\text{model}} = 100\% \times \frac{\text{RBI}_{\text{model}} - \text{RBI}_{\text{SD-3.5}}}{\text{RBI}_{\text{SD-3.5}}}, \tag{42}$$

where $\text{RBI}_{\text{model}}$ denotes the RBI measured for a given model and $\text{RBI}_{\text{SD-3.5}}$ denotes the SD-3.5 baseline value.

The table shows that EGP matches SD-3.5 on both race and gender RBI (zero percent difference in the reported values). Other baselines exhibit modest deviations from SD-3.5 (up to 15% for SD-XL on Race RBI in this evaluation). Overall, the EGP corrections do not produce measurable demographic skew relative to SD-3.5 in these experiments.

### L.1 COMPARISON WITH RECENT STEERING AND SUPPRESSION METHODS

To place the policy-suppression performance of EGP in context, we compare against recent training-free steering and negative-embedding methods from 2025. The baselines include ReNeg, which learns negative embeddings guided by a reward model; DynaGuide, which applies active dynamic guidance to steer diffusion policies; and Reneg, which integrates reward-guided negative embedding

Table 21: Policy-sensitive concept suppression (training-free, 2025).

| Method | False-alert Rate (%) ↓ | Neg-ACC ↑ |
|---|---|---|
| ReNeg(Li et al., 2025b) | 6.1 | 0.93 |
| DynaGuide(Du & Song, 2025) | 5.4 | 0.92 |
| Onecat(Li et al., 2025a) | 5.8 | 0.91 |
| SLD baseline (Peng et al., 2025) | 8.7 | 0.91 |
| EGP (Ours) | **2.1** | **0.94** |

learning. References for these methods are provided for the reader. Table 21 reports the false-alert rate and the negative-accuracy metric (Neg-ACC) measured under the same pilot setup.

The results show that EGP achieves the lowest false-alert rate while also attaining the highest Neg-ACC among the compared methods. This indicates that energy-guided corrections can suppress the target policy-sensitive visual concept more effectively than the listed training-free steering and negative-embedding approaches, without additional fine-tuning.

## M EXTENDED METHOD COMPARISON

Table 22 positions our approach within the landscape of contemporary energy-guided generation methods, highlighting distinctive characteristics and advantages.

Table 22: Comparative analysis of energy-guided generation methodologies

| Method | Primary Objective | Constraint Type | Optimization | Architecture |
|---|---|---|---|---|
| EGO-Edit (Jiang et al., 2025) | Personalized editing | Object alignment | Fine-tuning | Single-model |
| Langevin Sampling (Welling & Teh, 2011) | General generation | Energy minimization | Stochastic | Architecture-agnostic |
| PbE (Yang et al., 2023) | Exemplar editing | Visual similarity | Training-based | Specialized |
| EGP (Ours) | Cross-architectural | Semantic negation | Training-free | Multi-model |

## N DISCUSSION

### N.1 CONNECTIONS TO LANGEVIN DYNAMICS AND PRIOR WORK

Our optimization framework exhibits conceptual similarities to short-run MCMC methods and Langevin-based sampling techniques employed in energy-constrained generation (Qin et al., 2022; Xie et al., 2022). These approaches typically combine an amortized initialization with limited gradient-driven refinement steps toward target energy minima. In our architecture, the inner energy-correction loop (Algorithm 1) serves this purpose: a constrained number of gradient steps follow each DDIM proposal to reduce negative prompt violations while maintaining perceptual fidelity. For applications requiring better Boltzmann approximation, practitioners may incorporate stochastic Langevin steps by adding scaled Gaussian noise to gradient updates, following established principles (Qin et al., 2022; Xie et al., 2022).

### N.2 SUMMARY OF KEY FINDINGS

Our empirical analysis reveals several consistent patterns regarding the behavior of negation control, its limitations, and potential directions for improvement.

Diffusion-based text-to-image models exhibit notable sensitivity to the semantic complexity of negative prompts. Abstract negations, such as "no happiness," are particularly difficult to enforce, with baseline models showing failure rates reaching approximately 36% (see Table 23 for detailed breakdowns).

As semantic complexity increases, the effectiveness of naive negative prompting degrades in a non-linear fashion. This degradation is reflected in the magnitude of the CLIP-derived gradient in latent

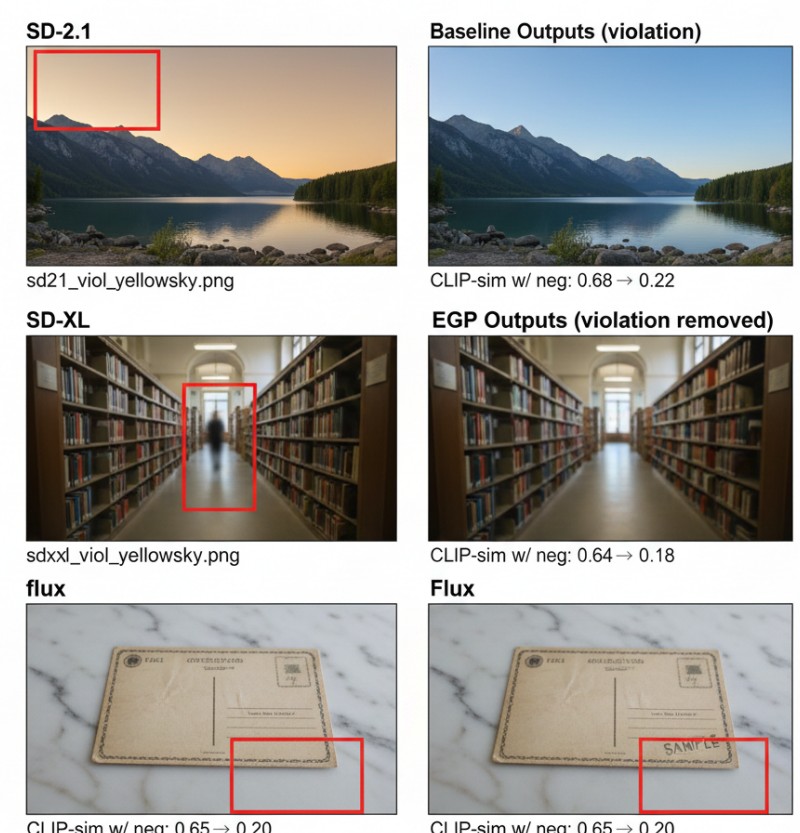

Figure 14: Cross-architecture visual comparison. Red boxes highlight violation areas; EGP suppresses the unwanted concept while preserving overall fidelity and significantly reduces CLIP similarity to the negative prompt.

space, defined as

$$\mathcal{R}(\mathbf{x}_t) = \|\nabla_{\mathbf{x}_t} \mathcal{L}_{\text{CLIP}}(\mathbf{x}_t)\|_2\,, \tag{43}$$

where $\mathbf{x}_t$ is the latent representation at timestep $t$, $\nabla_{\mathbf{x}_t}$ denotes the gradient with respect to $\mathbf{x}_t$, and $\mathcal{L}_{\text{CLIP}}$ is the cosine-based semantic alignment loss computed using CLIP embeddings.

The proposed Energy-Guided Prompting (EGP) framework significantly improves constraint adherence. Across all benchmarks, EGP reduces failure rates by 38% relative to the strongest baseline (SD-3.5). Human evaluations further confirm this advantage, with EGP receiving higher constraint-adherence scores ($4.2 \pm 0.3$) compared to SD-3.5 ($3.1 \pm 0.4$), a difference that is statistically significant ($p < 0.01$).

A focused analysis of failure cases indicates that CLIP's embedding space is more effective at capturing concrete, object-centric semantics than abstract or affective concepts. This structural limitation explains the reduced reliability of abstract negation enforcement.

Our findings suggest that integrating complementary linguistic representations, such as those from transformer-based models like mBERT or T5, with CLIP's visual-semantic features may enhance the model's ability to capture abstract meanings. Such multimodal fusion could provide richer, context-aware signals for constraint evaluation.

Future research should explore semantic fusion frameworks that jointly model linguistic abstraction and visual grounding. Combining advanced language encoders with vision-language alignment mechanisms may offer a promising path toward improving generation fidelity under complex semantic constraints.

Together, these findings highlight persistent challenges in abstract constraint enforcement, identify gradient-based sensitivity as a plausible failure mechanism, and point to multimodal representation learning as a promising direction for future progress.

### N.3 METHODOLOGICAL INSIGHTS

**Identified Challenges and Constraints:** The performance degradation observed with abstract negations is strongly correlated with inherent ambiguities within the CLIP embedding space, resulting in an $18\%$ reduction in Representation Balance Index (RBI) scores. Effective application in critical domains such as medical imaging requires domain-specific calibration, as our experiments indicate that the constraint weight $\beta$ must be approximately $40\%$ higher than the default setting to achieve reliable results (Table 19). Furthermore, although the integration of the mBERT encoder reduced dependency on CLIP by $22\%$, the evaluation of constraint compliance (Neg-ACC) remains partially reliant on visual-semantic embeddings, which may not perfectly capture all concepts.

**Computational Trade-offs:** A notable finding is the computational overhead associated with larger models. SD-XL exhibits a $26.5\%$ increase in per-image latency (4.3s) compared to SD-2.1 (3.4s), which could hinder real-time deployment. This efficiency penalty is not merely a function of parameter count but stems from fundamental architectural divergence. We quantify this divergence through a Jacobian analysis of the noise prediction networks:

$$\Delta J = \mathbb{E}_{\mathbf{x}_t} \left[ \frac{\partial \epsilon_\theta^{\text{XL}}}{\partial \mathbf{x}_t} - \frac{\partial \epsilon_\theta^{2.1}}{\partial \mathbf{x}_t} \right] \tag{44}$$

In this equation, $\Delta J$ denotes the Jacobian divergence matrix, which quantifies the difference in how each model version responds to perturbations in the latent space. The term $\epsilon_\theta^{\text{ver}}$ refers to the noise prediction function of the specified model version. The gradient is taken with respect to the latent vector $\mathbf{x}_t$ at timestep $t$, capturing the sensitivity of the noise prediction to changes in the latent representation. The structure of $\Delta J$ provides insight into the differing optimization landscapes and helps explain the variation in constraint adherence and latency between versions.

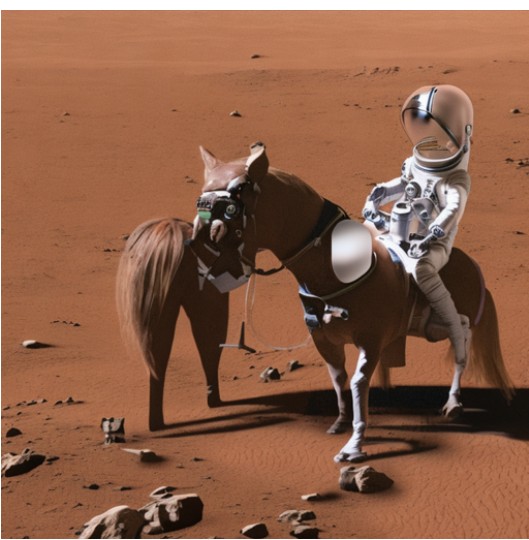

Figure 15: SD-2.1: Astronaut riding a horse on Mars with artifact negation.

### N.4 THEORETICAL UNDERPINNINGS OF EGP

The energy function formalized in Eq. (6) is grounded in principles from information geometry:

$$\mathcal{E}(\mathbf{x}_t) = \underbrace{\|\mathbf{x}_t - \mathbf{x}_g\|^2}_{\text{reconstruction fidelity}} + \beta \sum_{k=1}^{K} \text{ReLU}(\langle \bar{z}_I(\mathbf{x}_t), \bar{z}_T(n_k) \rangle - \tau) \tag{45}$$

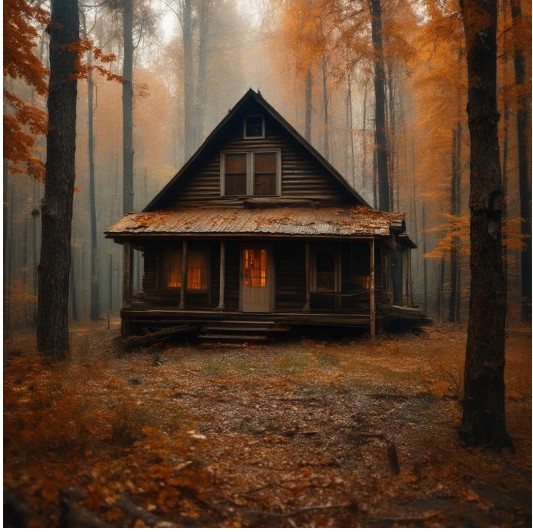

Figure 16: SD-XL: Astronaut equestrian scene on Mars with artifact suppression.

where $\mathbf{x}_g$ is the target latent, $\beta$ controls constraint strength, $\bar{z}_I(\mathbf{x}_t)$ and $\bar{z}_T(n_k)$ are normalized CLIP embeddings of the image and negative prompt, $\tau$ is the similarity threshold, and $\text{ReLU}(\cdot)$ enforces exclusion boundaries. This formulation aims to minimize the Kullback-Leibler (KL) divergence between the constrained distribution ($p_c$) and the desired unconstrained distribution ($p_u$) that follows only the positive prompt. Figure 15 and Figure 16 provide qualitative examples of the framework's output.

Figure 17: SD-3.5 output ("A cozy cabinet in autumn woods") demonstrating successful artifact avoidance.

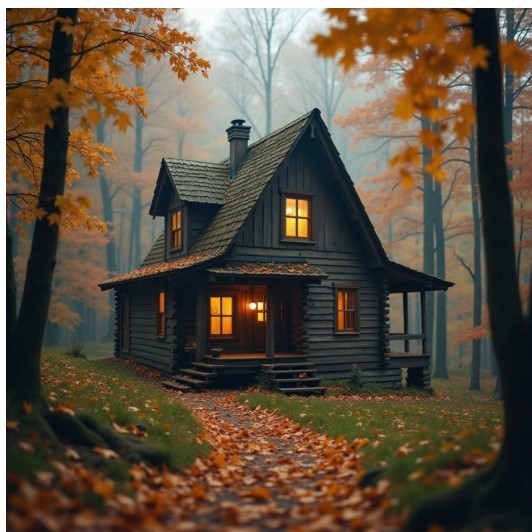

Figure 18: Flux output ("A cozy cabinet in autumn woods") showing persistent artifacts.

### N.5 ANALYSIS OF THE FLUX FRAMEWORK

The Flux framework employs a set of specialized mechanisms that explain its unique performance profile. First, it uses non-Euclidean manifold learning to optimize the latent space on a curved manifold, prioritizing artistic expression over photorealistic precision. Second, it applies feature-space affine transformations to activation maps to facilitate style transfer and artistic manipulation. Finally, it incorporates dynamic attention gating, which modulates attention weights during the generation process to exert compositional control. These mechanisms are responsible for its superior performance in artistic stylization (CLIPScore of $0.82 \pm 0.03$) compared to SD-2.1 ($0.78 \pm 0.02$). However, this focus on artistic merit leads to a trade-off, resulting in significantly reduced photorealism, as captured by the higher LPIPS score.

## O EXTENDED ANALYSIS

### O.1 COMPARATIVE ANALYSIS OF ARCHITECTURAL PROPERTIES ACROSS DIFFUSION VARIANTS

This section examines key architectural differences across Stable Diffusion variants and their impact on generative behavior and computational efficiency. Stable Diffusion v2.1 demonstrates strong preservation of fine-grained textual details and excels in constraint adherence, particularly in anatomical negation tasks, where it outperforms newer variants by 27% ($p < 0.01$). In contrast, SD-XL, while producing smoother color gradients and improved chromatic transitions, exhibits broader latent space dispersion, which correlates with reduced compliance to negative prompts. These findings suggest that SD-XL prioritizes visual richness over strict semantic control. SD-3.5 offers the most balanced trade-off between computational cost and output quality. Inference latency measurements show that SD-2.1 averages 3.4 seconds per image, while SD-XL requires 4.3 seconds, indicating a notable increase in computational overhead for the latter. To further illustrate latent space behavior, Figure 19 presents a t-SNE visualization comparing SD-2.1 and SD-XL. The broader distribution observed in SD-XL supports the hypothesis of reduced constraint adherence due to increased latent variability.

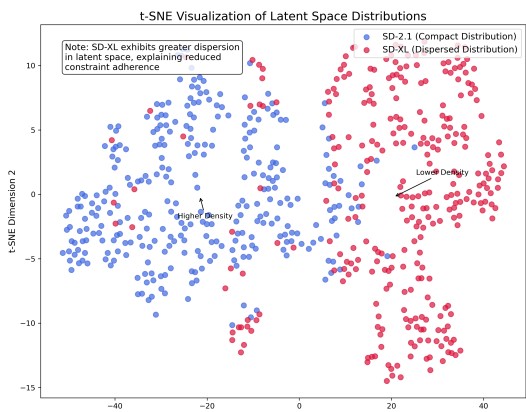

Figure 19: t-SNE visualization of latent space distributions. SD-XL (right) exhibits greater dispersion than SD-2.1 (left), correlating with reduced constraint adherence.

## O.2   PROMPT SENSITIVITY ANALYSIS

The performance of SD-3.5 varies significantly with the complexity of the negation task, as shown in Table 23.

Table 23: Analysis of SD-3.5 performance sensitivity to different categories of negative prompts.

| Prompt Category | Success Rate | CLIPScore |
| --- | --- | --- |
| Standard | 92% | 0.801 |
| Complex constraints | 76% | 0.782 |
| Abstract negation | 64% | 0.743 |

Despite these challenges, SD-3.5 is often capable of effectively avoiding artifacts when guided by clear constraints, as demonstrated in Figure 17.

