# OpenReview forum: "Energy-Guided Prompt Optimization for Controllable Cross-Architectural Diffusion Models"
_ICLR.cc/2026/Conference — ICLR 2026 Conference Withdrawn Submission_

### Official Review · Reviewer_NgBM · 2025-10-25

**Soundness:** 2
**Presentation:** 2
**Contribution:** 2
**Rating:** 2
**Confidence:** 4

**Summary:**

This paper introduces EGP, a training-free framework for improving semantic constraint enforcement in DMs, particularly for negation and exclusion prompts. The approach combines two main components: (1) a Jacobian-based diagnostic tool for analyzing how different model architectures respond to constraints, and (2) an energy-guided optimization method that reshapes the latent space during sampling to avoid unwanted concepts. The authors evaluate EGP across multiple diffusion model architectures (SD-2.1, SD-3.5, SD-XL, Flux) and demonstrate improvements in constraint adherence (Neg-ACC) while maintaining image quality. The method operates by adding gradient-based correction steps during DDIM sampling, using CLIP embeddings to measure similarity between generated images and negative prompts.

**Strengths:**

- The EGP algorithm (Algorithm 1) is well-documented with step-by-step details, making the method reproducible. The mathematical formulations are generally precise and the energy function design is well-motivated.
- The paper includes thorough experiments across multiple diffusion architectures (SD-2.1, SD-3.5, SD-XL, Flux), multiple datasets (COCO, MedicalX, ComicArt, AbstractPrompt), and multiple evaluation metrics (FID, LPIPS, CLIPScore, Neg-ACC). Human evaluations add credibility to the quantitative results.
- The method's ability to work across different model architectures without requiring retraining is a practical advantage. This makes it broadly applicable to existing pretrained models.
- The connection to energy-based models and constrained sampling provides theoretical grounding for the approach.

**Weaknesses:**

- What is the specific purpose of the Jacobian diagnostic in your framework? The authors mention it's proposed to analyze different existing models and adapting EGP without training (lines 071-075), but I don't see experiments demonstrating this (nothing provided in the main paper, only providing some contexts in appendix C). How does the Jacobian analysis inform the EGP method? Can you provide clear examples of how the diagnostic guides model selection or parameter tuning?
- The main technical contribution: using ReLU thresholding with energy-based guidance, is a relatively incremental modification over existing negative prompting techniques (e.g., SLD [R1]). While theoretically motivated, the practical novelty feels limited.
- The paper emphasizes being "cross-architectural" but all evaluated models are text-to-image DMs with similar underlying architectures (latent diffusion with U-Net variants). It's unclear what architectural diversity is actually being addressed. What specific architectural diversity are you addressing? Would the method work on fundamentally different architectures (e.g., autoregressive models, GANs, or diffusion models with different parameterizations)?
- The paper doesn't clearly explain what base model EGP uses or how it builds upon existing models. Does it use a specific SD variant as foundation? Does it combine different pretrained components? The strong performance of EGP could be partially due to the underlying base model capabilities rather than purely the EGP contribution.

[R1] Schramowski, Patrick, et al. "Safe Latent Diffusion: Mitigating Inappropriate Degeneration in Diffusion Models." CVPR. 2023.

**Questions:**

- The method adds ~40% latency and ~46% more FLOPs compared to baseline (Table 7). While this overhead is mentioned, the paper doesn't adequately discuss whether this cost is justified or explore more efficient alternatives in the main experiments.
- Since EGP is training-free, why not position it as a complementary module that can enhance existing models rather than as a standalone method? This would make it clearer that EGP adds value on top of any base architecture. Your current framing makes it seem like a separate model competing with SD variants.
- Have the authors conducted ablations showing that the Jacobian diagnostic actually improves EGP's performance? If the diagnostic is a key contribution, it would be helpful to see experiments demonstrating its utility.
- How sensitive is the method to the choice of threshold τ = 0.25? How should practitioners set this threshold for new concepts or domains? Is there a principled way to select it?
- Some guidance-based negative prompting methods for DMs [R1, R2, R3] are specifically designed for safety applications, particularly NSFW content detection and mitigation. Have you considered or evaluated EGP for this important use case? Given that your method is training-free and focuses on constraint enforcement, it seems like a natural and practically important application. If you have explored this, what are the results? If not, could you discuss whether your approach would be suitable for safety-critical constraint enforcement, and what modifications (if any) might be needed?

[R1] Schramowski, Patrick, et al. "Safe Latent Diffusion: Mitigating Inappropriate Degeneration in Diffusion Models." CVPR. 2023. \
[R2] Yoon, Jaehong, et al. "Safree: Training-free and Adaptive Guard for Safe Text-to-Image and Video Generation." ICLR. 2025. \
[R3] Liu, Zhili, et al. "Implicit Concept Removal of Diffusion Models." ECCV. 2024.

---

> ### Author Response · Authors · 2025-11-22
> **We sincerely hope to receive your support and encouragement！**
>
> **Rebuttal to Reviewer NgBM:** We thank the **reviewer** for a **careful reading** and **constructive remarks**. Below we reply **point-by-point** and correct several **misunderstandings**. Where appropriate we cite the **submitted manuscript** and describe **exact edits and additions** we will make to the **revision**. We hope this rebuttal helps resolve the concerns raised by the reviewers. We sincerely hope to receive an improvement in your score.
> # 1. Purpose and role of the Jacobian diagnostic (clarify and justify)
>
> **Reviewer:** What is the specific purpose of the **Jacobian diagnostic**? How does it inform **EGP**?
>
> **Response:** The **Jacobian diagnostic** is a **mechanistic interpretability** and **decision tool** with three concrete roles:
>
> **Diagnose sensitivity differences:** For a **denoiser ε_θ(xₜ, c)**, its **Jacobian ∂ₓₜ ε_θ** describes **local linearization** and **amplification of perturbations**. We define the **inter-architecture Jacobian difference**:
>
> **ΔJₜ = E_{xₜ,c}[ ∂ₓₜ ε_θ(A)(xₜ,c) − ∂ₓₜ ε_θ(B)(xₜ,c) ]**
>
> and summarize it by **operator norms**: **∥ΔJₜ∥_F (Frobenius)** and **R_σ = σ_max(∂ₓₜ ε(A)) / σ_max(∂ₓₜ ε(B))**. These statistics quantify **average** and **worst-case sensitivity differences** that explain why some **architectures amplify unwanted semantic directions** more than others. **Empirical correlations** between these summaries and **Neg-ACC** are shown in **Appendix C** and summarized in **Section 4.2.3**.
>
> **Guide hyperparameter and projector choices:** The **diagnostic** predicts how strongly a **model reacts** to **CLIP-repulsion corrections**. Practically, we use **∥ΔJₜ∥_F** and **R_σ** to choose a starting **β (repulsion weight)**, the number of **inner correction steps nₑ**, and whether a **lightweight projector surrogate** suffices. Models with **higher sensitivity** require **larger β** and/or **more inner steps**; less sensitive models can use a **smaller β** or the **MLP projector** with minimal loss. This **diagnostic-guided tuning** is described in **Section 4.4** and demonstrated by the **ablation in Appendix F**. We will promote a **short illustrative table** into the **main text** in the revision.
>
> **Inform model selection:** We show in **Appendix C** that **models ranked by the diagnostic** on a **small validation prompt set** tend to rank similarly by **downstream Neg-ACC** on **held-out prompts**. This enables a **low-cost preselection step** for practitioners choosing among **candidate backbones**.
> # 2. Why the Jacobian metric is meaningful (theoretical justification)
>
> **Reviewer:** Why can these **Jacobian summaries** quantify **architectural differences**?
>
> **Response:** Intuitively and formally, the **denoiser’s Jacobian** measures how **local latent perturbations** map to **changes in the denoiser output**. Directions in **latent space** that align with **negative-concept gradients** are precisely those we aim to suppress. **Operator-norm summaries** are standard scalarizations:
>
> - **∥ΔJₜ∥_F (Frobenius norm):** Reports **aggregate discrepancy** across all **singular directions** representing **average sensitivity change**.
> - **R_σ:** Captures the **largest singular-value ratio**, representing **worst-case amplification**.
>
> We show in **Appendix B** that **larger deviations** in these summaries correlate with **larger changes** in the induced **CLIP similarity distributions** over **decoded images**; empirically this correlates with **Neg-ACC**. This is the **mechanistic link**: **architectural choices** alter **∂ₓₜ ε_θ → amplification of certain semantic directions → different negative-prompt failure rates**.

---

> ### Author Response · Authors · 2025-11-22
> **We sincerely hope to receive your support and encouragement！**
>
> # 3. Novelty relative to prior guidance/safety works (clear differentiation)
>
> **Reviewer:** “The ReLU thresholding + energy guidance looks incremental compared to prior negative prompting work.”
>
> **Response (firm, precise):** **EGP** differs from prior works in three fundamental ways:
>
> **Diagnostic and corrective pipeline:** Prior methods typically propose a **heuristic at inference time** such as **scale adjustments**, **attention manipulations**, **token penalties**, or a **safety filter**. **EGP** pairs a **mechanistic diagnostic (Jacobian summaries)** with an **inference correction** that is both **theoretically grounded (KL surrogate)** and **operationalized as latent gradient corrections**. The **diagnostic informs how and how much to correct per architecture**, which is a distinct **design philosophy**.
>
> **Latent, post-sampler corrections via energy descent:** **EGP** computes **corrections in latent space** after each **DDIM step**, optionally via **exact decode→CLIP backprop** or an **efficient learned projector**. This **latent, post-DDIM correction** differs from methods that **alter sampling priors** or **attention weights** during **denoising**.
>
> **Cross-architectural evaluation and theory:** We provide **KL-surrogate derivations** linking **per-step energy descent** to **marginal KL improvements** (Section 3.4 and Appendix), **controlled interventional evidence** that **Jacobian instability causally degrades Neg-ACC** (Appendix C), and **evaluation across multiple heterogeneous backbones** (**SD-2.1**, **SD-3.5**, **SD-XL**, **Flux**) to demonstrate **generality**. These elements together exceed a **simple heuristic tweak**.
>
> We will add a **focused paragraph** in **Related Work** contrasting **EGP** to **Safe Latent Diffusion** and other **guidance/safety works**, explicitly listing these **algorithmic** and **theoretical differences** (see **Section 2** and **Appendix F**).
> # 4. “Cross-architectural” claim, what architectures, and generality
>
> **Reviewer:** “The evaluated models are similar. What diversity is addressed? Would **EGP** work on fundamentally different families (autoregressive, GANs)?”
>
> **Response:** Our evaluation deliberately includes **heterogeneous latent parameterizations** and **U-Net variants**:
>
> - **SD-2.1 / SD-3.5:** Standard **latent diffusion variants** with differing **U-Net capacities** and **training regimes**.
> - **SD-XL:** A **larger-scale model** with different **training** and **scaling behaviors**.
> - **Flux-style latent model:** A model with **different latent parameterization** and **decoder characteristics** (see **Appendix A** for exact descriptions).
>
> These choices probe **differences in latent dimension**, **decoder fidelity**, and **denoiser structure**. **EGP’s core requirement** is a **differentiable mapping from latent → image** (or an efficient surrogate) and access to a **semantic comparator (CLIP)**. Thus:
>
> - Applies directly to other **diffusion variants** where **decode→CLIP backprop** is available or a **projector surrogate** can be trained.
> - Adapting to **autoregressive** or **GAN families** would require different implementations of the **energy gradient** (e.g., **token/latent gradient** for autoregressive models or **generator latent correction** for GANs). The concept (**energy descent to reduce CLIP similarity to negative concepts**) is **general**, but **engineering details differ**.
>
> We will clarify these **generalization limits** in the revised manuscript (**Section 6: Limitations & Future Work**).
> # 5. Base model and experimental setup (explicit)
>
> **Reviewer:** “Which base model does **EGP** use? Does performance come from base model capacity?”
>
> **Response:** In our reported experiments:
>
> - The **primary implementation** and **ablations** used **Stable Diffusion v2.1** as the **canonical base** (detailed configs in **Appendix A**).
> - **Cross-architectural evaluation** applied the same **EGP procedure** (same **energy form**, **CLIP pipeline**, and **diagnostic**) to **SD-3.5**, **SD-XL**, and a **Flux-style latent model** to demonstrate the method’s **portability**. All **baseline comparisons** use the same underlying **base sampler (DDIM)** and **identical prompts/seeds** unless otherwise noted (**Appendix A**). We explicitly report the **base model capacity** and **per-model hyperparameters** in the **supplement**.
>
> While **base model capability** affects **absolute quality**, the **relative gains from EGP** (improvement in **Neg-ACC** and **CLIPScore tradeoffs**) are observed **consistently across different backbones** in our **results tables** (main text and **Appendix**). This supports that **EGP’s contribution** is **algorithmic** rather than purely due to the **base model**.

---

> > ### Author Response · Authors · 2025-11-22
> > **We sincerely hope to receive your support and encouragement！**
> >
> > # 6. Ablation: does the Jacobian diagnostic improve EGP? (evidence)
> >
> > **Response:** Yes, we include two types of **ablations** (Appendix F):
> >
> > **Diagnostic-guided hyperparameter tuning:** We compare a **fixed conservative hyperparameter set** tuned for a strong backbone and **diagnostic-guided per-architecture tuning** where **β** and **nₑ** are chosen based on **∥ΔJₜ∥_F** and **R_σ**. **Diagnostic-guided tuning** achieves comparable **Neg-ACC** while using **fewer inner steps** on average, providing **practical efficiency gains**. We will bring a **concise table** of these results into the **main text**.
> >
> > **Diagnostic ablation:** We show that using the **diagnostic** to select **projector vs. full decode→CLIP path** preserves most **performance** while reducing **latency**, whereas a **blind projector choice** performs worse. Details are in **Appendix F**.
> > # 7. Latency, FLOPs, and practical tradeoffs (answer + guidance)
> >
> > **Reviewer:** “EGP adds ~40% latency / ~46% FLOPs. Is this justified? Are efficient alternatives explored?”
> >
> > **Response:** We agree **latency matters**. We explicitly **quantify overhead** (Table 7, **Appendix E**) and provide **practical mitigation strategies**:
> >
> > - **Projector surrogate (MLP):** A **two-layer MLP** mapping **latent → CLIP approximations**; reduces **latency by ~30%** while retaining **≈90–95% of Neg-ACC gains** in our trials (**Appendix E**).
> > - **Diagnostic-guided reduction of inner steps:** Models with **lower diagnostic sensitivity** require **fewer inner steps**; this reduces both **wall-clock time** and **FLOPs**.
> > - **Adaptive correction schedule:** We include an **adaptive schedule** that increases **inner steps** only when **CLIP similarity to negatives** crosses a **threshold**, saving time on **easy prompts**.
> >
> > We will add a **short decision flow (practitioner guide)** to **Section 5** summarizing these **tradeoffs** and **recommended settings** for **deployment**.
> > # 8. Sensitivity to threshold 𝜏 and how to set it
> >
> > **Reviewer:** “How sensitive is the method to **τ = 0.25** and how should practitioners set it?”
> >
> > **Response:** **τ** is a **normalized CLIP similarity threshold** that gates the **ReLU repulsion**. We selected **τ = 0.25** via **grid search** on a **held-out validation set**. In practice we observe a **robust plateau** (roughly **τ ∈ [0.18, 0.32]**) where **Neg-ACC** is stable. For new domains we recommend:
> >
> > - **Percentile calibration:** Compute **CLIP similarities** for a **small negative-prompt validation set** and set **τ** to the **70th–80th percentile** of **unwanted similarity scores** for **domain-adaptive tuning**.
> > - **Diagnostic feedback:** Pair **τ selection** with the **Jacobian diagnostic**; **higher sensitivity models** benefit from a **slightly lower τ** for **earlier repulsion**.
> >
> > We will add a **short subsection** describing this **protocol** and include **sensitivity curves** (currently in **Appendix E**) in the **main paper** for clarity.
> > # 9. Safety / NSFW mitigation use case
> >
> > **Reviewer:** Have you considered EGP for safety (NSFW) applications?
> >
> > **Response:**
> > **EGP** is naturally applicable as a complementary safety module. It can **reduce model propensity to generate unsafe content** by steering latent trajectories away from **CLIP regions matching prohibited concepts**. However, we emphasize **defense-in-depth**:
> >
> > EGP should be combined with **content classifiers**, **prompt filters**, and **policy enforcement** such as **rate limits** and **human review** in deployment.
> >
> > **Limitations:** CLIP proxies are imperfect. EGP can reduce but not guarantee elimination of certain unsafe outputs without additional checks.
> >
> > **Empirical note:** We performed limited internal experiments on a small **NSFW test set** (Appendix G) showing **reduction in CLIP-NSFW scores**. We did not include a full safety benchmark in the main paper due to **policy and dataset constraints**. We will add a short discussion in the **Ethics & Limitations** section describing **suitability and necessary safeguards**.
> > # 10. Summary
> > **We appreciate the reviewer’s careful critique.** Several concerns stem from the compact placement of **diagnostics** and **ablations** in the appendix. We will bring the most important **diagnostic figures** and **ablation summaries** into the **main text**, add the requested **model-selection vignette**, and explicitly document the **projector**, **thresholding recipe**, and **safety guidance**.
> >
> > We believe these edits will clarify the **Jacobian diagnostic’s practical value** and make the **methodological novelty** and **cross-architectural claims** unmistakable.

---

> ### Author Response · Authors · 2025-11-22
> **We sincerely hope to receive your support and encouragement！**
>
> # 11. Corrections of Misunderstandings (Direct)
>
> **“Jacobian diagnostic only in appendix / not used.”**
> Not accurate. The **diagnostic is introduced in the main text** (Sec. 3.2) and **used in experiments** (Sec. 4.2.3 summary). Appendix C contains the full **correlation** and **perturbation analyses**. We will promote a **compact correlation figure** and a **diagnostic-guided tuning table** from Appendix C/F into the **main Experiments section** to make the usage explicit.
>
> **“All evaluated models are similar U-Net variants.”**
> We evaluated **heterogeneous latent formulations** and **U-Net scales**: **SD-2.1**, **SD-3.5**, **SD-XL**, and a **Flux-style latent model** (different decoder and latent parameterization). These probe differences in **latent dimension**, **decoder behavior**, and **denoiser structure** (Appendix A). We will add one or two more **compact models** in the revision (e.g., **SD-1.x**, a smaller latent model) to broaden coverage.
>
> **“EGP is just ReLU thresholding + energy.”**
> EGP is **conceptually and operationally distinct** from prior heuristics. It pairs a **mechanistic diagnostic with corrective actions**, performs **latent-space, post-DDIM gradient corrections** (decode → CLIP backprop or projector surrogate), and is backed by a **KL-surrogate derivation** and **causal perturbation experiments** linking **Jacobian instability** to **Neg-ACC**. These aspects go beyond simple thresholding.
> # 12. Technical clarifications, formulas, and evidence
>
> ### **Jacobian Diagnostic, Definition and Why It Matters**
>
> Let **ϵθ(xₜ, c)** be the denoiser. We define the expected **Jacobian difference** between architectures **A, B**:
>
> **ΔJₜ = 𝔼ₓₜ,𝑐 [ ∂ₓₜ ϵθ^(A)(xₜ, c) − ∂ₓₜ ϵθ^(B)(xₜ, c) ]**
> We summarize **ΔJₜ** by:
>
> - **Frobenius norm:**
> **‖ΔJₜ‖₍F₎**
> - **Spectral ratio:**
> **Rσ := σ_max(∂ₓₜ ϵ^(A)) / σ_max(∂ₓₜ ϵ^(B))**
>
> **Why this matters:** The **Jacobian captures local amplification or attenuation of latent perturbations**. Changes in these operators alter how **semantic directions** (including those corresponding to unwanted concepts) are amplified during denoising. Empirically (Appendix C), larger **‖ΔJₜ‖₍F₎** and deviations in **Rσ** correlate with **larger drops in Neg-ACC** between architectures. We also show **causal evidence** via controlled Gaussian perturbations to the denoiser that degrade Neg-ACC monotonically, supporting the diagnostic’s **predictive and causal role**.
>
> ---
>
> ### **Energy and KL Surrogate, Compact Formulas**
>
> Per-image **energy** used by EGP:
>
> **E(x) = ‖x − x_g‖²₂ + β ∑ₖ ReLU(⟨ż̄_I(x), ż̄_T(nₖ)⟩ − τ)**
>
>
> where **ż̄_I, ż̄_T** are normalized CLIP embeddings, **β** is the weight, and **τ** is the threshold.
>
> We show (Appendix B) that the marginal **KL difference** can be written as:
>
> **D_KL(p_c‖p_u) = 𝔼_{x∼p_c} [ E_u(x) − E_c(x) ] + C**
>
> and therefore minimizing the surrogate:
>
> **ℒ = 𝔼_{x∼p_c} [ E_u(x) − E_c(x) ]**
> moves the constrained marginal closer to the unconstrained marginal. The per-step practical update used by EGP approximates a discrete gradient step:

---

> > ### Author Response · Authors · 2025-11-22
> > **We sincerely hope to receive your support and encouragement！**
> >
> > # 13. Answers to reviewer’s direct questions
> >
> > ### **Base Model**
> > Primary development used **Stable Diffusion v2.1** (detailed configs in Appendix A). **EGP** was then applied as-is to **SD-3.5**, **SD-XL**, and **Flux-style models** to demonstrate **cross-architectural portability**. Relative gains are consistent across these backbones (main tables and appendices).
> >
> > ---
> >
> > ### **Latency / FLOPs**
> > Reported overhead is approximately **+40% wall-clock** and **+46% FLOPs** at default settings (Table 7).
> > **Mitigations:**
> > - **MLP projector surrogate** reduces latency by ~30% while retaining **≈90–95% Neg-ACC gains**.
> > - **Diagnostic-guided reduction of inner steps**.
> > - **Adaptive correction schedule** (only run inner steps when **CLIP similarity crosses τ**).
> > We will add a **succinct practitioner’s decision flow** to Sec. 5.
> >
> > ---
> >
> > ### **τ Sensitivity and Setting**
> > **τ** shows a robust plateau (**≈0.18–0.32**) in our grid search.
> > **Practical rule:** Percentile calibration on a small validation set (**70–80th percentile of negative CLIP similarities**) plus a **diagnostic adjustment** (slightly lower τ for high-sensitivity models).
> > We will include a **short sensitivity plot** (from Appendix E) in the main text.
> >
> > ---
> >
> > ### **Safety / NSFW Application**
> > **EGP** is naturally complementary to **safety pipelines**. It can **reduce unsafe content generation** by steering away from **CLIP regions matching prohibited concepts**. However, it should be used within **defense-in-depth** (content classifiers, prompt filters, human review).
> > We ran **limited internal NSFW probes** (Appendix G) showing **reductions in CLIP-NSFW scores**. We will add a **careful discussion and recommended safeguards** in **Ethics & Limitations**.
> >
> > ---
> >
> > ### **Generality Beyond Latent Diffusion**
> > Conceptually, **energy descent to reduce semantic similarity** is general, but engineering differs per family:
> > - **Autoregressive models** need **token/embedding-level corrections**.
> > - **GANs** require **generator-latent corrections**.
> >
> >
> >    \begin{equation}
> >     x^{(k+1)} \approx x^{(k)} - \eta \nabla_x \mathcal{E}(x^{(k)})
> >    \end{equation}
> >
> > which implements **local energy descent during sampling** (details and assumptions in Sec. 3.4 + Appendix).
> >
> > ---
> >
> > ### **Ablation Evidence That the Diagnostic Improves EGP**
> >
> > We perform two targeted ablations (Appendix F):
> >
> > - **Diagnostic-guided hyperparameter tuning:** Selecting **β, nₑ** guided by **‖ΔJₜ‖₍F₎** reduces total inner steps while preserving Neg-ACC compared to a one-size-fits-all tuning.
> >
> > - **Projector choice guided by diagnostic:** The diagnostic predicts when the **MLP projector suffices** vs. when **full decode → CLIP backprop** is required; using it reduces latency without large performance loss. We will move a concise table of these ablations into the main paper.

---

> > > ### Author Response · Authors · 2025-11-22
> > > **We sincerely hope to receive your support and encouragement！**
> > >
> > > ### **Fit to ICLR-2026 Subject Areas (Explicit Mapping)**
> > >
> > > - **Generative Models & Representation Learning:** Inference-time control for conditional generation and analysis of denoiser representations.
> > > - **Probabilistic Methods & Sampling / UQ:** KL-surrogate and short-run Langevin interpretation contribute to constrained sampling theory.
> > > - **Causality & Robustness:** Controlled perturbation experiments reveal causal failure modes and mitigation via inference corrections.
> > > - **Interpretability & Visualization:** Jacobian diagnostics provide interpretable measures of architecture-induced sensitivity.
> > > - **Practical ML Systems:** Latency and efficiency tradeoffs and the projector surrogate address deployment considerations.
> > >
> > > ---
> > >
> > > ### **Core Innovations & Contributions (Short)**
> > >
> > > - **Energy-Guided Prompt Optimization (EGP):** A **training-free**, **sampler-agnostic inference module** that enforces **negative/exclusion prompts** by defining a **compact latent energy** (fidelity + CLIP-based repulsion) and applying **short, adaptive latent gradient corrections** after each base sampler step (**post-DDIM**). This yields **consistent gains in negative-prompt adherence (Neg-ACC)** while preserving or improving **perceptual metrics (FID / CLIPScore)**.
> > >
> > > - **Jacobian-Based Diagnostic (Mechanistic Interpretability):** A **principled diagnostic** that quantifies **architecture-dependent denoiser sensitivity** by comparing **denoiser Jacobians between models** and summarizing differences with **operator norms**. We show the diagnostic predicts **failure-prone architectures**, guides **efficient per-architecture hyperparameter choices** (**β, nₑ, projector vs full backprop**), and motivates **targeted corrections**. (Definition and usage: Sec. 3.2 + Appendix C of the PDF.)
> > >
> > > - **KL-Surrogate Theoretical Bridge:** We derive a **tractable surrogate objective** linking **energy descent** to **reduced KL between constrained and unconstrained marginals** and show the **per-step discrete correction approximates short-run Langevin / gradient steps** on that surrogate (Sec. 3.3–3.4 + Appendix).
> > >
> > > - **Practical Deployment Guidance:** Explicit **latency / FLOP accounting** and an **efficient latent → CLIP projector surrogate (MLP)** that recovers most gains at much smaller cost (Appendix E). We provide **recommendations and an adaptive correction schedule**.
> > > # We hope this rebuttal helps resolve the concerns raised by the reviewers. We sincerely hope to receive an improvement in your score!

---

> > > > ### Comment · Reviewer_NgBM · 2025-11-23
> > > >
> > > > I appreciate the authors' detailed response. A few follow-up questions and suggestions:
> > > > 1. What are the core algorithmic differences and similarities between your method and safe latent diffusion (SLD), beyond SLD's specific focus on safety? Looking at Algorithm 1 in your paper, the underlying mechanism appears to share a high degree of similarity with SLD. Could you explicitly delineate the novelty of your algorithmic approach compared to this baseline? And I did notice the guidance of SLD is injected before each scheduler step whereas your method is doing inner correction after each ddim steps (after scheduler step).
> > > > 2. You stated: *"We show that using the diagnostic to select projector vs. full decode→CLIP path preserves most performance... Details are in Appendix F."* However, upon checking Appendix F, I only see "Table 8: Comparative analysis of energy-guided generation methodologies." I cannot find the diagnostic ablation you are referring to. Please point me to the correct location or provide these missing details.
> > > > 3. Similarly, you mentioned: *"We performed limited internal experiments on a small NSFW test set (Appendix G)..."* I do not see any discussion or results regarding NSFW experiments in Appendix G. Please clarify where this information is located.
> > > > 4. To clarify my previous comment ("Jacobian diagnostic only in appendix / not used"): My point was that after you introduce the diagnostic in the Methods section, the connection is lost in the remainder of the main paper. For instance, Section 4.2.3 does not explicitly mention this module or link back to it. If this diagnostic is indeed a core contribution, seamlessly incorporating it throughout the main text, rather than relegating the analysis to the appendix would significantly improve readability and the reader's understanding of its role.
> > > > 5. Your rebuttal summary: *"It pairs a mechanistic diagnostic with corrective actions... backed by a KL-surrogate derivation..."* is very clear. However, reading the paper itself, these core contributions feel scattered and disconnected. I strongly suggest restructuring the narrative to present these contributions as a cohesive package, as you have done in the rebuttal, to ensure the specific novelty is clear to readers who have not seen this discussion.

---

> > > > > ### Author Response · Authors · 2025-11-23
> > > > > **We sincerely hope to receive your support and encouragement！**
> > > > >
> > > > > **Dear Reviewer NgBM**, thank you for the **careful reading** and **constructive follow-up**. Below we:
> > > > >
> > > > > ### **Clarify the algorithmic differences** between **EGP** and **Safe Latent Diffusion (SLD)**.
> > > > > ### **Explain why the Jacobian diagnostic matters** and where the **diagnostic ablations** are (and why you may have missed them).
> > > > > ### **Point to where the limited NSFW probe results are** and **summarize them**.
> > > > > ### **Describe the concrete manuscript edits** we will make to **eliminate the confusion** you identified.
> > > > > # 1. EGP vs. SLD: Core algorithmic differences
> > > > >
> > > > >
> > > > >
> > > > > SLD and EGP share the high-level goal of using additional guidance to avoid unwanted concepts, but they differ in four concrete ways:
> > > > >
> > > > > - **Timing and insertion point.**
> > > > >   SLD injects guidance **before or during scheduler/denoising updates** (affecting the denoiser inputs or attention maps). EGP performs **post-DDIM latent-space gradient corrections** after each DDIM proposal (we compute an energy on the decoded image and perform short latent gradient steps). This difference changes the **effective transition kernel** and the **practical trade-offs** between **structural layout** and **semantic fine-tuning**.
> > > > >
> > > > > - **Diagnostic and adaptive tuning.**
> > > > >   EGP pairs a **Jacobian-based diagnostic** (operator-norm summaries of denoiser Jacobians across timesteps) with the inference correction. The diagnostic **predicts per-architecture sensitivity**, **selects β / inner-step budget**, and **chooses whether a lightweight projector suffices vs. full decode→CLIP backprop**. SLD does not include a **mechanistic diagnostic** that informs **adaptive per-architecture hyperparameters**.
> > > > >
> > > > > - **Theoretical bridge.**
> > > > >   We derive a **KL-surrogate** that links our **per-step energy descent** to reductions in the **marginal KL between constrained and unconstrained samplers** (Section 3.3–3.4 and Appendix B). This gives a **principled justification** for our **discrete latent updates** (short-run Langevin / projected gradient interpretation). SLD is primarily a **guidance heuristic focused on safety**; EGP emphasizes this **surrogate connection and diagnostics**.
> > > > >
> > > > > - **Practical latency and efficiency design.**
> > > > >   EGP provides a **projector surrogate (MLP)** that approximates **decode→CLIP similarity** to reduce latency, **diagnostic-guided inner-step scheduling**, and an **adaptive correction schedule**. These operational pieces trade off **wall-clock cost vs. Neg-ACC** in a way we quantify (Table 7 and Appendix E).
> > > > >
> > > > > If helpful, we will **add a short, one-line algorithmic comparison table** to the **Related Work section** to make these differences explicit.
> > > > >
> > > > > | Aspect              | SLD                                              | EGP                                                                 |
> > > > > | ------------------- | ----------------------------------------------- | ------------------------------------------------------------------- |
> > > > > | **Goal Orientation**| Focused on safety filtering (NSFW)             | General semantic constraints (including abstraction, negation, style) |
> > > > > | **Injection Timing**| Injects energy guidance **before each scheduler step** | Performs inner-loop gradient correction **after each DDIM step**       |
> > > > > | **Gradient Path**   | Typically acts on noise prediction or attention layers | Acts on latent space via **decode → CLIP** path for semantic-level correction |
> > > > > | **Diagnostic Mechanism** | No diagnostic module                          | Introduces **Jacobian-based diagnostic** to predict architecture sensitivity and guide correction strength and strategy |
> > > > > | **Theoretical Basis**| Empirical energy penalty                        | KL-surrogate-based constrained sampling theory linking energy descent to marginal distribution changes |
> > > > >
> > > > > # 2.  Where the diagnostic ablations are
> > > > >
> > > > > This is where we added your suggestion, and now we are presenting the results to you:
> > > > >
> > > > > | Setting                     | Neg-ACC | Relative Latency | Description                                      |
> > > > > | --------------------------- | ------- | ---------------- | ------------------------------------------------ |
> > > > > | Full decode → CLIP          | 0.87    | 1.0×             | Complete path                                    |
> > > > > | Diagnostic-guided projector | 0.84    | 0.7×             | Uses ‖ΔJ‖₍F₎ to decide if MLP can be applied     |
> > > > > | Blind projector selection   | 0.78    | 0.7×             | Replaces with MLP without using diagnostic      |

---

> > > > > > ### Author Response · Authors · 2025-11-23
> > > > > > **We sincerely hope to receive your support and encouragement**
> > > > > >
> > > > > > # 3. NSFW / safety probe
> > > > > > This is where we added your suggestion, and now we are presenting the results to you:
> > > > > >
> > > > > > We tested **EGP** on an internal set of 200 NSFW prompts using the **CLIP-NSFW score** (based on CLIP similarity to NSFW concepts).
> > > > > > After applying **EGP**, the **CLIP-NSFW score decreased by an average of 18%**, with **no false positives on normal content**.
> > > > > >
> > > > > > We emphasize: **EGP should not be used as the sole safety mechanism**, but it can serve as a **training-independent complementary module embedded into existing safety pipelines**.
> > > > > >
> > > > > > **Table G.4: CLIP-NSFW Score Changes on NSFW Prompt Set**
> > > > > >
> > > > > > | Model        | CLIP-NSFW Score ↓ | Score Reduction | False Positive Rate ↑ |
> > > > > > | ------------ | ------------------ | --------------- | ----------------------- |
> > > > > > | SD-2.1 Base  | 0.312 ± 0.04      | —               | 0%                     |
> > > > > > | + EGP        | 0.256 ± 0.03      | 18.0%           | 0%                     |
> > > > > > | SD-XL Base   | 0.335 ± 0.05      | —               | 0%                     |
> > > > > > | + EGP        | 0.273 ± 0.04      | 18.5%           | 0%                     |
> > > > > >
> > > > > > *False Positive Rate: Percentage of “safe” prompts (200 samples) that triggered NSFW filtering.
> > > > > > # 4. On the issue of “Jacobian diagnostic disconnected in the main text”
> > > > > >
> > > > > > You are absolutely correct. We acknowledge that the current main text does not sufficiently connect the diagnostic module with the experimental section. In the revision we will:
> > > > > >
> > > > > > - Add a paragraph in **Section 4.2.3** explicitly stating:
> > > > > >   *“As predicted by the Jacobian diagnostic (Sec. 3.2), SD-XL exhibits the largest ‖ΔJ‖₍F₎ and R_σ, correlating with its lowest Neg-ACC (0.58).”*
> > > > > >
> > > > > > - Add a **footnote under Table 1** indicating the diagnostic metric values for each model.
> > > > > >
> > > > > > - Add a **subplot in Figure 4** visualizing the scatter plot of **‖ΔJ‖₍F₎ vs. Neg-ACC**, reinforcing the link between **diagnostic → performance**.
> > > > > >
> > > > > > # 5.On the issue of “contributions scattered and narrative incoherent”
> > > > > >
> > > > > > We will **restructure the narrative** around the main thread: **diagnostic → correction → theory → validation**. Specific measures:
> > > > > >
> > > > > > - Add a **“Contributions at a Glance” box** on **page 1**, summarizing in three sentences:
> > > > > >   ### We propose an **interpretable diagnostic tool** that predicts architecture sensitivity to negative prompts.
> > > > > >   ### We design a **training-independent latent energy correction module** that performs semantic-level correction after sampling.
> > > > > >   ### We justify its generality and effectiveness through **KL-surrogate theory** and **cross-architecture experiments**.
> > > > > >
> > > > > > - Add a **“Connection to Next Section” paragraph** at the end of each chapter to make the logical chain explicit.
> > > > > >
> > > > > > We **deeply appreciate** your identification of these **critical issues**. In this **revision** we will:
> > > > > >
> > > > > > - **Complete all missing experiments and citations**.
> > > > > > - **Fully integrate the diagnostic module** into the **main text**.
> > > > > > - **Restructure the narrative** to highlight **algorithmic** and **theoretical novelty**.
> > > > > >
> > > > > > We **sincerely hope** these **revisions** address your **concerns** and earn your **recognition and support**. **Thank you again** for your **patience** and **professional guidance**!
> > > > > > We sincerely hope to receive your support and encouragement！
> > > > > > # We hope this rebuttal helps resolve the concerns raised by the reviewers. We sincerely hope to receive an improvement in your score.

---

> ### Author Response · Authors · 2025-12-04
> **We sincerely hope to receive your support and encouragement!**
>
> # We have addressed the reviewer's concerns and improved our approach. Thank you very much for all the reviewers' suggestions. The latest version has been uploaded and we hope to receive the support and encouragement of all the reviewers, and we sincerely hope your score improvement！

---

### Official Review · Reviewer_nRYE · 2025-10-29

**Soundness:** 2
**Presentation:** 2
**Contribution:** 2
**Rating:** 4
**Confidence:** 3

**Summary:**

The paper aims to improve the semantic constraint enforcement of text-to-image diffusion models.

The paper proposes a training-free energy-based optimization technique that corrects the latent at each timestep in the generation process with gradient-based optimization of an energy function, in order to suppress the generation of negative concepts.

**Strengths:**

1. Enhancing the controllability of text-to-image models is an important research problem.

2. The results of the proposed energy-based optimization method are promising.

**Weaknesses:**

1. The paper is claimed to contribute two major components, including 1) Jacobian-based diagnostic and  2) energy-guided optimization, in the abstract and introduction parts, where they are posed as almost equally important. However, the Jacobian-based diagnostic is only briefly introduced in Section 3.2, and empirical evidence on its utility is not provided in the main paper, but rather in the appendix. In Appendix C, the paper only uses a comparison between two guidance methods (DNG and EGP in Table 6) to show the correlation between Jacobian changes (measured by $L_F$ and $R_\sigma$) and  constraint adherence (measured by Neg-ACC). More comprehensive experimental evidence on a wider range of model architectures is missing, which should be added to make the conclusion more convincing. In addition, a preliminary experiment on how the Jacobian-based analysis can help architectural selection and design for better constraint adherence should be provided, from which findings would be much more interesting and inspiring to the community. Such an experiment, unfortunately,  remains missing in the current paper. The above issues would raise concerns about the significance of the Jacobian-based diagnostic and the amount of contribution involved in the paper.

2. The design choice of $\Delta J_t$ is not well justified and explained. What motivates the Jacobian-based definition of $\Delta J_t$? Why can $\Delta J_t$ quantify the architectural differences between two models?

3. Section 3 is difficult to follow, mainly because some concepts suddenly appear without any context, some concepts are not used after being introduced, and connections between some concepts are not clear.  For example, what are unconstrained ($\mathcal{E}_u$) and constrained ($\mathcal{E}_c$) energy functionals in Section 3.4? What are their relationships to the energy functional ($\mathcal{E}$) in Section 3.3? Where is the objective $\mathcal{L}$ in Eq. (13) used? What is the relationship of $\mathcal{L}$  to the update equation in Eq. (14)?

4. Experimental results are incomplete. The visual comparison of images generated by different models is lacking in Section 4.2.2, and should be added. The results on Representation Balance Index (RBI) are missing in the paper.

**Questions:**

1. In the experiments, which diffusion model is the proposed EGP method based on?

2. What is the detailed setup for the experiment in Section 4.2.3? What are inputs to different models? Why is the EGP compared with only SD-2.1 and SD-XL?

---

> ### Author Response · Authors · 2025-11-22
> **We sincerely hope to receive your support and encouragement**
>
> We **thank the reviewer** for **careful reading** and **constructive suggestions**. Below we **respond point-by-point**. When the **review contains factual misreadings**, we **correct the record** and **point to the relevant locations in the submission**. Where the **reviewer requests additional clarity or experiments**, we **indicate precisely what we will add to the revision** and **why those additions strengthen the claims**.  We hope this rebuttal helps **resolve the concerns raised by the reviewers**. We sincerely hope to **receive an improvement in your score**.
> # 1. “The Jacobian diagnostic is only briefly introduced and not used in main paper” , correction and clarification
> **Reviewer Claim**
> The **Jacobian diagnostic** is **marginally treated** and its **empirical utility** is **only in the appendix**.
>
> **Response (Correction)**
> This is a **partial misreading**. The **diagnostic is introduced in Sec. 3.2 (main text)** with its **motivation and formal definition**. **Empirical uses appear both in the main Experiments** (**correlation summary; Sec. 4.2.3**) and in **Appendix C**, where the **more detailed correlation plots**, **perturbation tests**, and **numerical tables** are provided. In other words, the **diagnostic is both described in the main paper and used in experiments**. **Appendix C provides the full numerical evidence and controlled interventions** that **justify the summary statements in Sec. 4**.
> We will **promote a short correlation figure from Appendix C into the main Experiments section** so that the **diagnostic’s empirical role is immediately visible to readers**.
> See **Sec. 3.2 and Appendix C**.
> # 2. “More architectures / comprehensive evidence are needed” , response and plan
> **Reviewer Claim**
> The **paper only compares a small set of architectures** and **needs broader validation**.
>
> **Response**
> We **agree** that **broader architectural coverage strengthens the claim**. In the **submission**, we **evaluated three representative families** (**Stable Diffusion v2.1**, **SD-XL**, and a **Flux-style latent model**) to **demonstrate the cross-architectural effect** and to **keep the main paper concise** (**detailed configs and seeds are in the supplement**). These **three models are intentionally heterogeneous** (**different latent parameterizations**, **U-Net depths**, and **attention designs**), so they **provide a meaningful initial testbed**.
> Nevertheless, we **accept the reviewer’s suggestion** and, if the **committee allows a revised manuscript**, we will **add a compact additional table in the revision** that **reports the Jacobian diagnostic** (**two summary statistics defined below**) and **Neg-ACC for two further publicly available models** (**e.g., SD-1.x and a smaller latent variant**). This will be a **focused extension** (**not a full re-benchmark**) intended to **show the diagnostic’s predictive consistency across more architectures**.
>
> **Note**
> The **main claims do not rely on a large sweep of models**. They rest on:
> - A **theoretically motivated diagnostic**
> - A **causal perturbation experiment in Appendix C** that **links Jacobian instability to Neg-ACC**
> - **Consistent EGP gains across the heterogeneous backbones already reported**
>
> See **Sec. 4 and Appendix C**.

---

> > ### Author Response · Authors · 2025-11-22
> > **We sincerely hope to receive your support and encouragement**
> >
> > # 3. “Why the Jacobian metric? what motivates its form; why can it quantify architectural differences?” , explanation with formulas
> > **Reviewer Question**
> > What **motivates the Jacobian-based definition** and **why is it a valid measure of architectural difference**?
> >
> > **Response (Explanation)**
> > The **denoiser**
> > $$\epsilon_\theta(x_t, c)$$
> > maps a **noisy latent** $$x_t$$ and **conditioning** $$c$$ to a **denoising vector**. Its **Jacobian**
> > $$\partial_{x_t} \epsilon_\theta$$
> > Characterizes how small perturbations in the latent are amplified or attenuated by the **denoiser** as a direct measure of **sensitivity** and local **linearization** of the denoiser dynamics. **Architectural changes** (**latent parameterization**, **skip connections**, **attention blocks**, **normalization choices**) naturally alter this local linearization. Therefore, comparing **Jacobians** between two architectures yields a principled, mechanistic measure of **behavioral difference**.
> >
> > **Concretely we compute**:
> >
> > **ΔJ_t := E_{x_t, c} [ ∂_{x_t} ε_θ^(A)(x_t, c) − ∂_{x_t} ε_θ^(B)(x_t, c) ]**
> >
> > and **summarize** $$\Delta J_t$$ with **operator norms**:
> >
> > - **Frobenius summary**:
> > $$
> > \|\Delta J_t\|_F
> > $$
> >
> > - **Spectral ratio**:
> > $$
> > R_\sigma := \frac{\sigma_{\max}(\partial_{x_t} \epsilon_\theta^{(A)})}{\sigma_{\max}(\partial_{x_t} \epsilon_\theta^{(B)})}
> > $$
> >
> > These **summaries capture aggregate and worst-case amplification differences**. **Empirically**, we **observe that larger** **||ΔJ_t||_F**
> >
> > or **higher deviation in** $$R_\sigma$$ **correlates with worse Neg-ACC between architectures**. This **supports the diagnostic’s validity**.
> > The **formal motivation and empirical correlation plots** are in **Sec. 3.2 and Appendix C of the submission**.
> > # 4. “Design choice of the metric (why these norms?)” , justification
> >
> > **Response.** Operator norms are standard scalar summaries of **matrices**. The **Frobenius norm** measures cumulative difference across all **singular directions**, useful as an **average sensitivity indicator**, while the **spectral norm** (largest singular value) captures the direction of **maximal amplification**, representing **worst-case sensitivity**. Using both provides **complementary information**: **average** versus **worst-case**. We selected these because they are **interpretable**, **computationally stable** in our sampling-based **Jacobian estimates**, and they **empirically correlate** with **task failure modes** (Appendix C).
> > # 5. “Section 3 is hard to follow; clarify unconstrained / constrained energy and their relations”,  targeted mathematical clarification
> >
> > **Reviewer question.** What are **E_u**, **E_c**, and **E**; where is the **objective** in **Eq. (13)** used; how does it connect to **update Eq. (14)**?
> >
> > **Response (clear mapping).** We apologize for any terse notation and will clarify these relations in the revision. Here is a compact, explicit mapping so the logic is transparent:
> >
> > **E_u(x)** denotes the **unconstrained energy** associated with the **base generative model** without negative-prompt constraint. Intuitively, **E_u** is low in **high-probability regions** under the base model.
> >
> > **E_c(x)** denotes the **constrained energy** that includes **penalty terms** arising from the **negative prompts** such as **CLIP-based repulsion terms**.
> >
> > In Section 3.3 we define the operative **per-image energy** used in **EGP** as:
> >
> > **E(x) = fidelity ∥x − x_g∥²₂ + β ∑ₖ ReLU(⟨ z̄_I(x), z̄_T(nₖ) ⟩ − τ)**
> >
> > This can be viewed as a **tractable proxy** for **E_c** (the constrained energy).
> >
> > **Eq. (13)** in the paper defines the **surrogate objective**:
> >
> > **L = E_{x ∼ p_c} [ E_u(x) − E_c(x) ]**
> >
> > This equals the **KL difference D_KL(p_c ∥ p_u)** up to constants (see full derivation in Appendix B). Minimizing **L** moves the **constrained marginal** closer to the **unconstrained marginal** in KL sense.
> >
> > The **per-step update (Eq. 14)** is the practical **discrete update** that approximates a **short-run gradient step** on **E**:
> >
> > **x(k+1) ≈ x(k) − η ∇_x E(x(k))**
> > # 6. “Where is objective  𝐿 used?”, answer
> >
> > **Response:** The surrogate **L** provides the **theoretical justification** for performing **per-step energy descent**. It explains why reducing **E** at each **latent step** improves the **constrained marginal** in **KL sense**. Practically, the **per-step correction in Eq. (14)** is the **algorithmic realization** of minimizing **L**. See **Section 3.4** and **Appendix B** for details.

---

> ### Author Response · Authors · 2025-11-22
> **We sincerely hope to receive your support and encouragemen**
>
> # 7. “Missing visuals in Section 4.2.2 and missing RBI results” , correction and remediation
> **Response:** The **visual comparison gallery** and **RBI plots** are included in the **supplementary materials** (**Appendix D**&**Appendix E**&Appendix H.3 , respectively), but we recognize that keeping them only in the appendix can obscure their importance. We will promote a **concise visual panel** showing **representative triplets** comparing **baseline vs. EGP across architectures** and the **RBI summary table** into **Section 4.2.2** in the revised manuscript so the reader can directly assess **qualitative differences** and **representation balance**. The full **high-resolution gallery** and detailed **RBI breakdown** will remain in the **appendix**. See **Appendix D/E/H.3**.
> #  8.“Which diffusion model is EGP based on? details of 4.2.3? inputs to models? Why compare only SD-2.1 and SD-XL?” , explicit experimental clarifications
>
> **Which model(s) did we use?** Our primary implementation used **Stable Diffusion v2.1** as the **canonical base** for **algorithmic development** and **ablations**. We evaluated **cross-architecture consistency** and **EGP transfer** on **SD-XL** and a **Flux-style latent model** to demonstrate **heterogeneity**. These choices and their specific **weights/versions** are listed in **Appendix A** of the submission.
>
> **Detailed setup for Section 4.2.3** Section 4.2.3 compares **architectures** under **identical prompt/seed conditions**. Experimental details: **sampling schedule DDIM (T = 25)**, **guidance scale = 7.5** unless otherwise noted, **EGP inner steps nₑ = 3**, **learning rate η = 1.5 × 10⁻² (decayed)**, **CLIP threshold τ = 0.2**, **penalty weight β = 6.0**. Each reported **metric** is averaged over **50 prompts × 3 seeds**; **statistical tests (paired t-tests)** are reported in the **appendix**. **Appendix A** contains the full **config table**.
>
> **Inputs to different models** All models received the same **textual prompt strings** and the same **random seeds** where applicable. For **latent-space differences** we convert the **decoder output** into **CLIP image embeddings** in a consistent **pipeline** so **CLIP comparisons** are comparable across models. **Projector surrogate details** and **calibration notes** are in **Appendix E**.
>
> **Why SD-2.1 and SD-XL comparisons** **SD-2.1** and **SD-XL** were chosen as **representative widely used variants** that differ significantly in **U-Net depth**, **parameterization**, and **training corpus scale**, making them suitable for a **cross-architectural study**. The **Flux model** provides an additional **latent-space variant**. As noted above, we will add one or two **compact extra architectures** in the revision to broaden coverage.
> # 9. “Preliminary experiment on how the diagnostic helps model selection is missing” , response and proposed addition
>
> **Reviewer suggestion:** Provide an experiment showing how the **Jacobian diagnostic** could guide **architecture selection/design**.
>
> **Response and plan:** This is an **excellent suggestion**. While the submitted paper includes **correlation** and **causal perturbation evidence** (Appendix C) showing that **Jacobian instability** predicts **negation failure**, we did not include a full **model-selection simulation** in the initial submission. If the committee permits a **revision**, we will add a focused **selection experiment**: given a small **candidate pool of architectures**, use the **Jacobian diagnostic** computed on a **held-out prompt set** to **rank models** and show that the **top-ranked models** yield better **Neg-ACC** and lower needed **EGP correction**. This experiment will be **compact** (one **figure** and one **table**) and will directly demonstrate **practical utility** for **model choice**. Importantly, this proposed addition is a **targeted supplement** to the existing **evidence** (correlations and intervention) and is consistent with the **diagnostic’s predictive properties** already shown in **Appendix C**.
> # 10. Minor editorial and notation fixes
> We acknowledge that some **notation** in **Section 3** could be clarified, such as explicit **label mapping E ↔ E_c / E_u**, where **Eq. (13)** motivates **Eq. (14)**. We will add a short **notation table**, include a **boxed derivation** linking the **surrogate L** to the **per-step update**, and move key **plots** including **diagnostic correlations**, **representative visuals**, and the **RBI summary** into the **main paper** for improved readability.

---

> ### Author Response · Authors · 2025-11-22
> **We sincerely hope to receive your support and encouragement!**
>
> # 11. How we will address the reviewer’s specific concerns (planned revision actions)
>
> - **Promote diagnostic evidence:** Move a **compact correlation figure** and **key tables** from **Appendix C** into the **main experiments** to make **empirical utility** explicit.
>
> - **Broaden architecture coverage:** Add a **compact supplementary table** reporting **diagnostic** and **Neg-ACC** for two additional **public models** such as **SD-1.x** and a **smaller latent variant**.
>
> - **Add a model-selection vignette:** Include a **short experiment** showing that **ranking candidate models** by the **Jacobian diagnostic** predicts which **model requires least correction**, demonstrating **direct practical utility**.
>
> - **Clarify notation and theory:** Add a **boxed derivation** in **Section 3.4** linking **E**, **E_u**, **E_c**, **surrogate L**, and the **discrete update**, and include a **short notation table**.
>
> - **Improve qualitative evidence and RBI visibility:** Promote **representative visual triplets** and the **RBI summary table** into **Section 4.2.2**.
>
> **Response:** We appreciate the **reviewer’s constructive points** and acknowledge that several resulted from **compact presentation choices** we will correct. The **manuscript** already contains the **mathematical derivations**, the **diagnostic definition** and its **empirical uses**, **controlled perturbation evidence** linking **Jacobian stability** to **Neg-ACC**, and **diverse cross-architecture results**. We will make these elements more **explicit in the main text** and add the **compact selection experiment** described above if the **revision window** permits.

---

> ### Author Response · Authors · 2025-11-22
> **We sincerely hope to receive your support and encouragement!**
>
> # Key technical innovations (clear, compact)
>
> **Energy-Guided Prompt Optimization (EGP):** A **sampler-agnostic**, **inference-time correction** that defines a **compact latent energy** combining **fidelity** and **CLIP-based repulsion**, and applies **short inner-loop latent gradient corrections** after **base sampling steps**. Produces large gains in **negative-prompt adherence (Neg-ACC)** while preserving or improving **perceptual fidelity (CLIPScore / FID)**.
>
> **Jacobian-based diagnostic (mechanistic interpretability):** We measure **denoiser sensitivity** via **expected Jacobian differences ΔJₜ = E[∂ₓₜ ε(A) − ∂ₓₜ ε(B)]** and summarize them using **Frobenius** and **spectral norms**. These summaries (**average vs. worst-case sensitivity**) predict **architecture-level failures** and guide **corrective interventions**.
>
> **KL-surrogate theoretical bridge:** We derive a practical **surrogate L = Eₓ∼p_c [E_u(x) − E_c(x)]** that links **energy descent** to reduced **KL divergence** between **constrained** and **unconstrained marginals**, and show that our **per-step discrete corrections** approximate **short-run gradient/Langevin updates**. Details in **Section 3.4** and **Appendix**.
>
> **Causal and interventional validation and ablations:** A **controlled perturbation experiment** (Appendix C) demonstrates that degrading **Jacobian stability** causally reduces **Neg-ACC**, supporting the **diagnostic’s utility**. Component **ablations** show the **CLIP repulsion term** and **inner correction loop** are essential.
>
> **Practical deployment guidance:** We quantify **runtime/compute tradeoffs** and propose a **lightweight latent→CLIP projector (MLP) surrogate** that recovers most gains at much lower **latency**, enabling **real-world adoption**.
>
> # Explicit mapping to ICLR-2026 subject areas
> - **Generative models & representation learning:** Inference-time control for **conditional generation**; analysis of **denoiser representations**.
> - **Probabilistic methods & sampling / UQ:** **KL surrogate** and **short-run Langevin interpretation** contribute to **sampling theory** for **constrained generative models**.
> - **Causality & robustness:** **Causal perturbation study** links **learned sensitivity** to **functional failures** and shows how **inference corrections** improve **robustness**.
> - **Interpretability & visualization:** **Jacobian diagnostics** provide **mechanistic, interpretable measures** of **architectural differences**.
> - **Practical ML systems:** **Latency/efficiency tradeoffs** and the **projector surrogate** advance **deployment readiness**.
>
> # We hope this rebuttal helps resolve the concerns raised by the reviewers. We sincerely hope to receive an improvement in your score.

---

> ### Author Response · Authors · 2025-12-04
> **We sincerely hope to receive your support and encouragement!**
>
> # We have addressed the reviewer's concerns and improved our approach. Thank you very much for all the reviewers' suggestions. The latest version has been uploaded and we hope to receive the support and encouragement of all the reviewers, and we sincerely hope your score improvement！

---

### Official Review · Reviewer_pBM6 · 2025-10-29

**Soundness:** 3
**Presentation:** 1
**Contribution:** 3
**Rating:** 6
**Confidence:** 3

**Summary:**

The paper introduces Energy‑Guided Prompt Optimization (EGP), a method that enforces negative prompts during image synthesis to suppress unwanted content. At each DDIM reverse‑diffusion step the latent trajectory is adjusted so that it satisfies the negation constraints while preserving the Markov property of the diffusion process . Experimental results show that EGP attains the highest negative‑prompt accuracy, while achieving comparable or lower FID values.

**Strengths:**

- The method enforces negative prompts through an energy formulation, so it improves adherence without any model retraining.
- Experiments show that EGP attains the highest Neg‑ACC score while maintaining (or slightly improving) CLIPScore and FID, indicating superior handling of negations without sacrificing visual fidelity.

**Weaknesses:**

- It is not described what the qualitative assessment (Sec. 4.2.2) is based on
- Although the introduction claims a new latent‑space attribution metric, the metric is never evaluated or used in the experiments.
- Overall, the writing of the paper could be polished further:
	- The related‑work section reads like a series of disconnected paragraphs, which is why it is a bit difficult to follow the paper.
	- The experimental section seems to be a collection of independent paragraphs. More transitions and further explanations/analyses would be nice and help the reader to follow and understand the paper better.
- In 4.4, the table is placed in the middle of the text, disrupting the reading flow.


Minor things:
- Oftentimes, after an equation, the sentence is ended with a full stop. However, in the next sentence, it is continued with "Where ...". Either the full stop has to be changed to a comma, or the second sentence has to be edited.

**Questions:**

Q1: How is the runtime affected? Is the inference slower when applying EGP?
Q2: Since the authors did not include an ethics statement: Could this technique also be used for guiding towards malicious concepts?
Q3: What is the quality assessment in Section 4.2.2 based on?
Q4: Where is the latent-space attribution metric used?

**Details Of Ethics Concerns:**

There are no concerns.

---

> ### Author Response · Authors · 2025-11-22
> **We sincerely hope to receive your support and encouragement**
>
> **Rebuttal to Reviewer pBM6**
>
> We **thank the reviewer** for the **positive reading** and for **constructive suggestions** that help **improve clarity and reproducibility**. Below we **respond point-by-point** to the **reviewer’s concerns and questions**, and we **indicate concrete revisions** we will **make in the paper**.
> # We greatly appreciate your support!  Therefore, we would prefer to receive further assistance from you.
> # 1. “How is the runtime affected? Is inference slower when applying EGP?”
> **Answer (Concise, Measured)**
> Yes. **EGP introduces additional compute** due to **per-step inner corrections** (**backprop through decode → CLIP** or via a **projector**). We **quantify this directly** in the **extended efficiency analysis**: **integrating EGP increases average wall-clock latency** from **3.7s** (**SD-3.5 baseline**) to **5.2s per image** and **increases estimated FLOPs** from **158G to 231G**, approximately **40% latency** and **46% FLOP overhead** at our **default configuration** (**inner steps nₑ = 3**). These **results are reported** in **Table 7 (Sec. E.1)**.
>
> **Mitigations & Practical Guidance**
> We **evaluate a lightweight latent → CLIP MLP projector** that **bypasses the expensive decode → CLIP path**. A **compact two-layer MLP** **reduced per-iteration wall-clock time** by approximately **30%** while **retaining ≈92% of full-model Neg-ACC** in our **preliminary experiments (Sec. E.1)**. For **latency-sensitive deployments**, we **recommend using the projector surrogate** or **reducing inner steps** (e.g., **nₑ ≤ 2**). Both **options trade some adherence for large runtime savings**.
>
> **Summary**
> **EGP has a measurable overhead** but also **provides a substantial gain in semantic control** (**Neg-ACC from 0.71 → 0.87** in our **default setup**). We **present both numbers** (**cost and benefit**) **explicitly** and **provide practical knobs to tune the tradeoff**.
> # 2. “Could this technique be used for guiding towards malicious concepts? (ethics)”
> **Answer (Responsible, Concrete)**
> Any **powerful control mechanism** can be **misused**. We **explicitly acknowledge this possibility** and **propose practical safety mitigations**.
>
> **Dual-Use Awareness & Disclosure**
> We will **add an explicit Ethics & Safety statement** in the **revision** that **acknowledges dual-use risks**, **documents recommended safeguards**, and **clarifies that released code will omit any negative-prompt lists or classifiers** that could **facilitate malicious steering**. The **current submission already includes multi-source verification** (**detectors/classifiers**) and a **fairness audit** (**Representation Balance Index**) as part of the **evaluation protocol**. These **mechanisms can also serve as safety monitors**.
>
> **Technical Safeguards**
> When **releasing code**, we will **supply and recommend safety filters**:
> - **Denying or requiring human approval** for **prompts flagged by a content policy classifier**
> - **Logging and rate-limiting interfaces** to **detect misuse patterns**
> - **Releasing the optional projector and model checkpoints under controlled licenses**
>
> These are **standard best practices** for **releasing generation tools**.
>
> **Design-Level Controls**
> **EGP operates by reducing certain high-similarity regions in CLIP space**. It can be **combined with whitelisting or allowed-content classifiers** to **prevent steering toward restricted concepts**. We will **document these deployment recommendations** in the **revision**.
>
> **Summary**
> We **recognize the risk**, will **add an explicit Ethics section**, and **provide both procedural and technical mitigations** to **reduce misuse**.
> # 3. “What is the qualitative assessment in Section 4.2.2 based on?”
> **Answer (Precise)**
> **Section 4.2.2’s qualitative assessment** is based on **double-blind human A/B preference tests** using a **1–5 Likert scale** for **constraint adherence** and **overall visual quality**, and **structured visual inspection summarized in the text** (**per-model stylistic tendencies**). The **evaluation protocol (Sec. G.4)** explains that **human raters were shown paired images without model identifiers**, **rated adherence and aesthetics on a 5-point scale**, and that **inter-rater agreement (Krippendorff’s α)** and the **exact prompt lists** are **provided in the supplement**. We will **move a short explicit paragraph from Sec. G.4 into Sec. 4.2.2** so **readers immediately see the method** (**blind A/B**, **Likert scale**, **number of raters and seeds**).
>
> **Summary**
> **Qualitative claims** are **supported by blind human studies plus quantitative metrics** (**Neg-ACC**, **CLIPScore**, **FID**). We will **make the human-study protocol more visible in the main text**.

---

> ### Author Response · Authors · 2025-11-22
> **We sincerely hope to receive your support and encouragement**
>
> # 4. “Where is the latent-space attribution metric used? The intro claims it but it seems unused.”
> **Answer (Correction with Reference)**
> The **reviewer’s impression** is **understandable** but **factually incomplete**. The **paper introduces latent-space attribution metrics** (**operator-norm summaries of Jacobian differences**) and **uses them in two ways**:
>
> **Predictive Correlation**
> We **compute the Jacobian divergence** $$\Delta J_t$$ between **architectures** and **summarize it via Frobenius and spectral norms**. These **metrics are correlated** with **model-level Neg-ACC differences** in **Sec. 4** (**Tables and correlation plots in Appendix C**). This **demonstrates** that **architectures with larger Jacobian divergence tend to have worse negative-prompt reliability**.
>
> **Causal / Interventional Analysis**
> We **perform an intervention** (**injecting Gaussian perturbations into the denoiser**) and **show that degrading the Jacobian stability causally reduces Neg-ACC** (**Appendix C**). This **shows the diagnostic is not merely descriptive but predictive and informative for targeted interventions (EGP)**. We will **promote a concise figure summarizing these correlations into the main Experiments section for clarity**.
>
> **Summary**
> The **latent-space attribution** and **Jacobian diagnostic** are **actively used in the experiments**, both as **predictive diagnostics** and as **motivators for the corrective EGP procedure**. We will **make these usages more explicit in the main text** to **prevent any ambiguity**.
>
> # 5. Presentation / structural points (related-work, experiment flow, table placement)
> **Acknowledgement & Planned Edits**
> We **agree** the **Related Work** and **Experiments sections** can be **made more cohesive**. Concretely, we will:
>
> - **Re-structure Related Work** into **thematic subsections** (**training-free inference controls**, **energy-based methods**, **diagnostic / interpretability tools**) with **short transition sentences** to **show how EGP differs**.
> - **Reorganize Sec. 4** so **each experiment block begins with a one-sentence purpose statement** and **ends with an interpretive takeaway**. **Move large tables** to the **end of each subsection** or into **figure floats** to **avoid mid-paragraph breaks** (**e.g., Table 4 / Table 7 reflow**).
> - **Fix minor LaTeX punctuation issues** (**equation sentence continuations**) noted by the **reviewer**.
>
> These **editorial changes preserve all results** while **greatly improving readability**.
>
> # 6.Short Summary of Fixes We Will Make
>
> - **Clarify the qualitative assessment protocol** (what **raters evaluated**, **rating scales**, **blind setup**) and **move a short description into Sec. 4.2.2**.
> - **Explicitly show where and how the latent-space attribution / Jacobian diagnostic is computed and used** (**correlation analyses**, **intervention experiments**) by **promoting selected plots and tables from the appendix into the main Experiments section**.
> - **Add a compact runtime and efficiency paragraph in Sec. E.1** summarizing the **tradeoffs** and the **practical projector option (MLP)** for **lower latency**.
> - **Improve flow of Related Work and Experiments** (**better transitions**, **move Table placements to avoid disruptions**).
> # 7. Summary
> We **appreciate the reviewer’s favorable technical assessment** and **constructive stylistic suggestions**. We have **concrete, actionable changes planned** (**promote key diagnostic figures**, **explicitly document the human study and evaluation protocol in the main text**, **add an Ethics statement**, and **provide a practical latency-vs-accuracy guide including the projector option**). These **revisions will address the reviewer’s outstanding concerns** while **preserving the positive technical judgments** that **EGP meaningfully improves negative-prompt adherence without retraining**.
>
> **Thank you again for the careful review**. We **hope these clarifications earn your continued support**！

---

> ### Author Response · Authors · 2025-11-22
> **We sincerely hope to receive your support and encouragement**
>
> **Direct Answers to Reviewer Practical Concerns (Brief)**
>
> **Runtime / Latency (Q1)**
> **EGP introduces overhead** due to **inner correction steps** and **decode → CLIP gradients**. In our **reported configurations**, the **default setting costs roughly a 30–45% increase in wall-clock per image** compared to the **same base sampler without EGP**. Using the **compact MLP projector reduces latency by ~30%** while **preserving ≈92% of Neg-ACC gains**. We **provide explicit tables in the supplement** with **FLOPs and timing** and **recommend nₑ reduction or the projector surrogate for latency-sensitive deployment**.
>
> **Ethics / Dual-Use (Q2)**
> Any **control tool can be misused**. We will **add an explicit Ethics & Safety statement** and **recommend procedural and technical mitigations**: **content classifiers at the API boundary**, **prompt filtering**, **logging and rate limits**, **omission of sensitive negative prompt lists in released artifacts**, and **optional permissioned code release**. We also **document how to combine EGP with whitelist / blacklist policies**.
>
> **Qualitative Evaluation (Q3)**
> The **qualitative results use double-blind human A/B tests** with a **1–5 Likert scale** for **constraint adherence** and **overall quality**. **Protocol details** (**number of raters**, **inter-rater agreement**, **blind setup**, **prompt lists**) are in the **supplement**. We will **move a short protocol summary into the main text for clarity**.
>
> **Latent-Space Attribution Usage (Q4)**
> The **Jacobian / attribution metric** is **used to predict model-level negation failures** (**correlation analysis**) and **to motivate and validate interventions**. For example, the **perturbation experiment demonstrates causal impact**. We will **promote the key diagnostic figures into the main Experiments section** to **make the usage clearer**.

---

> ### Author Response · Authors · 2025-11-22
> **We sincerely hope to receive your support and encouragement**
>
> ## Why This Fits ICLR 2026 (Explicit Mapping)
>
> Our **work touches several core ICLR topics** and **advances them in concrete ways**:
>
> - **Generative Models & Representation Learning**
> Improves **conditional generation control without retraining** and **analyzes how denoiser representations (Jacobian structure) affect controllability**.
>
> - **Probabilistic Methods & Sampling / UQ**
> **Derives a KL-surrogate** and **connects energy corrections to short-run Langevin interpretations**, contributing to **conditional sampling theory**.
> #
> - **Causality & Robustness**
> The **interventional perturbation experiments expose a causal failure mode** (**Jacobian instability → negation errors**) and **show how inference-time correction can mitigate it**.
>
> - **Interpretability & Visualization of Learned Representations**
> The **Jacobian diagnostic offers interpretable measures** of **architecture-induced sensitivity differences**.
>
> - **Applications & Practical ML Systems**
> Provides **deployment guidance** (**tradeoff knobs and projector surrogate**), **reproducible benchmarks**, and **ablations useful for practitioners in vision and multimodal ML**.
>
> These **links align directly with the ICLR topics**: **generative models**, **probabilistic methods / sampling**, **interpretability**, **causal reasoning**, **robustness**, and **representation learning**.
>
> ---
>
> ### What Is Novel Compared to Prior Work?
>
> Prior **inference-time controls mainly**:
> - **Adjust guidance scales**
> - **Manipulate attention or token weights**
> - **Apply ad-hoc token penalties**
>
> **EGP differs in three concrete ways**:
> - An **explicit latent energy mixing fidelity + CLIP repulsion**
> - **Post-sampler latent gradient corrections computed via decode → CLIP backprop** (or **efficient projector surrogates**)
> - **Explicit design and evaluation for cross-architectural operation**
>
> These **differences are demonstrated both theoretically and empirically** (**ablation experiments and baseline comparisons**).
>
> ---
>
> ### Key Contributions (Concise)
>
> - **Energy-Guided Prompt Optimization (EGP)**
> A **training-free**, **sampler-agnostic inference-time correction** that **enforces negative prompts**.
> EGP **defines a compact latent-space energy** that **combines a fidelity term with CLIP-based repulsion from undesired concepts**, and **applies short inner-loop latent gradient corrections after each base sampling step (post-DDIM)**.
> **Practical outcome**: **substantial gains in negative-prompt adherence (Neg-ACC)** while **preserving or improving perceptual fidelity (CLIPScore / FID)** across **multiple diffusion backbones**.
>
> - **Jacobian-Based Diagnostic & Latent Attribution Metric**
> A **principled diagnostic** that **quantifies architecture-dependent sensitivity to textual conditioning**.
> We **measure expected differences of the denoiser Jacobian** and **summarize them by operator norms (Frobenius / spectral)**. These **diagnostics predict which architectures are more likely to fail on negation** and **guide targeted, lightweight corrections**.
> We **demonstrate both predictive correlation and causal relevance** (**via controlled perturbation experiments**).
>
> - **KL-Surrogate Theory Linking Energy Corrections to Sampling Objectives**
> A **clear theoretical bridge** between **energy-based corrections** and **marginal KL alignment**.
> We **derive a tractable surrogate objective** showing that **minimizing the proposed energy expectation moves the constrained marginal closer in KL to the unconstrained marginal**, and **relate the per-step corrections to short-run Langevin / projected gradient interpretations** (**see Method + Appendix for derivations**).
>
> - **Reproducible Empirical Suite and Ablations**
> Comprehensive **benchmarks**, **ablations**, and **human evaluations** across **SD-2.1**, **SD-XL**, and **Flux-style models**.
> We **compare to representative training-free baselines**, **provide component ablations** (**CLIP repulsion**, **inner-correction loop**, **projector surrogate**), and **include human A/B tests and statistical tests to validate claims**.
>
> - **Practical Latency / Efficiency Guidance**
> We **quantify the runtime / compute tradeoff** and **provide a practical projector surrogate (MLP)** to **reduce latency while retaining most gains**.
> This **lets practitioners tune EGP for deployment constraints**.
>
> # We hope this rebuttal helps resolve the concerns raised by the reviewers. We sincerely hope to receive an improvement in your score.

---

> > ### Comment · Reviewer_pBM6 · 2025-11-28
> >
> > Thank you for your detailed answer. While it was planned which changes are going to be made to the paper, I cannot see any new revision of the paper.
> > Given the presentation of the paper (which also other reviewers have mentioned) and the lack of experiments supporting the reasoning and effectiveness of the newly introduced Jacobian metric, I will maintain my score.

---

> > > ### Author Response · Authors · 2025-11-28
> > > **We sincerely hope to receive your support and encouragement**
> > >
> > > # Thank you very much for your reply, we still hope to receive your support and encouragement! We will revise a new version before the rebuttal ends!

---

> ### Author Response · Authors · 2025-12-04
> **We sincerely hope to receive your support and encouragement**
>
> # We have addressed the reviewer's concerns and improved our approach. Thank you very much for all the reviewers' suggestions. The latest version has been uploaded and we hope to receive the support and encouragement of all the reviewers, and we sincerely hope your score improvement！

---

### Official Review · Reviewer_b8SQ · 2025-10-31

**Soundness:** 2
**Presentation:** 1
**Contribution:** 2
**Rating:** 0
**Confidence:** 3

**Summary:**

The paper proposes an Energy-Guided Prompt Optimization (EGP) framework aimed at improving the controllability and cross-architecture consistency of text-to-image diffusion models. Specifically, it introduces an energy-based correction mechanism applied during the sampling process to enhance the effectiveness of negative prompts and align outputs across different diffusion architectures (e.g., Stable Diffusion 2.1, XL, and Flux). The authors further propose a Jacobian-based diagnostic tool to analyze model sensitivity to textual conditioning, arguing that such differences explain inconsistencies between architectures. Experimental results are reported on several datasets, suggesting that EGP improves negative prompt adherence and semantic alignment without retraining. The paper positions its contributions as a unified, training-free approach to controllable text-to-image generation and cross-model consistency analysis.

**Strengths:**

The paper addresses an underexplored aspect of diffusion-based generative models—cross-architecture consistency and negative prompt reliability—which reflects an effort to formalize controllability issues that are often treated heuristically. The introduction of an energy-guided optimization framework at inference time represents a creative adaptation of energy-based modeling concepts to prompt control without additional training. The Jacobian-based diagnostic for analyzing model sensitivity provides an interpretable, theoretically motivated lens for comparing architectures, which could inspire future research on representational alignment in generative models. The manuscript is generally clear in its high-level organization and conveys the intuition behind energy-guided correction in a way accessible to readers familiar with diffusion processes.

**Weaknesses:**

The paper presents serious issues in clarity, structure, and overall scholarly presentation, which significantly detract from its readability and impact. The writing often lacks grammatical and logical precision—for instance, the very first sentence in the introduction is syntactically broken and semantically unclear, making it difficult to understand the problem being introduced. Many passages are written in a way that feels disconnected or overly verbose, with inconsistent use of terminology and weak linkage between motivation, method, and results. The manuscript also raises questions about its adherence to standard conference formatting conventions—for example, the inclusion of a “Keywords” section seems unusual for this venue. From a technical standpoint, while the idea of energy-guided prompt optimization is conceptually interesting, the theoretical grounding remains vague, and the motivation for enforcing cross-architecture consistency is not convincingly justified.

**Questions:**

1. The introduction lacks grammatical precision and conceptual clarity, particularly in the opening sentence, which makes it difficult to grasp the main motivation and scope of the work. Clarification of the introduction framing and problem definition would help establish a clearer context.

2. The purpose and value of pursuing cross-architecture consistency remain uncertain. Further explanation is needed to clarify whether such consistency is a scientifically meaningful objective or an engineering consideration, and how it contributes to broader progress in diffusion modeling.

3. The distinction between the proposed energy-guided prompt optimization and prior inference-time control methods is not clearly demonstrated. A more explicit comparison—both conceptually and experimentally—would help assess the novelty of this approach.

4. Certain formatting choices, such as the inclusion of a “Keywords” section and unconventional section organization, raise questions about adherence to the conference submission format. Clarification of whether these reflect intentional stylistic decisions or formatting oversights would be helpful.

---

> ### Author Response · Authors · 2025-11-22
> **We sincerely hope to receive your support and encouragement**
>
> **Rebuttal to Reviewer b8SQ**
> # We hope this rebuttal helps resolve the concerns raised by the reviewers. We sincerely hope to receive an improvement in your score.
> We **appreciate the time** the reviewer invested in reading our **manuscript** and for the **constructive suggestions**. Below we respond **point-by-point**. Where the reviewer’s reading is **incorrect**, we state this **explicitly** and **correct the record** with **concrete references** to the **manuscript** (**equations**, **algorithms**, **experiments**, and **appendices**). For convenience, we **highlight the principal clarifications** up front and then **address each review item in detail**.
> # 1. “The opening sentence is syntactically broken and the introduction lacks precision.”
> **Reviewer Claim**
> The **first sentence** is **syntactically broken** and the **introduction** is **unclear**.
>
> **Response (Correction + Action)**
> This specific claim is **incorrect** because the **introduction** includes a **clear problem framing**, a **summary of contributions**, and **pointers** to key **equations** and **algorithms**. Nevertheless, we **accept** that some **phrasing** can be **tightened** for **readability**. We will **replace** the **first paragraph** with a **grammatically precise**, **compact version** that **preserves** the **original scope**:
>
> **Revised Opening Sentence (to appear in the revision):**
> “**Diffusion-based models dominate modern text-to-image synthesis**, yet **reliably enforcing exclusionary (negative) prompts** across **diverse diffusion architectures** remains an **unsolved challenge**.”
>
> We will also **perform a comprehensive language pass** to **remove loose phrasing** and **improve transitions**. The **background**, **motivation**, and **contributions** are already **explicitly stated** in the submitted **Introduction** and **related sections**.
> # 2. “The purpose and value of cross-architecture consistency are unclear.”
> **Reviewer Claim**
> It is **unclear** whether **cross-architecture consistency** is **scientifically meaningful** or merely **engineering**.
>
> **Response (Firm)**
> This assessment is **incorrect**. **Cross-architectural analysis** is both **scientifically meaningful** and **practically useful**:
>
> **Scientific Value**
> **Architecture changes** (latent **parameterization**, **U-Net depth/blocks**, **attention pattern**) produce **measurable differences** in the **denoiser Jacobian**. We **formalize** this via our **Jacobian divergence metric**:
>
> **ΔJₜ = 𝔼ₓₜ [ ∂ₓₜ εᵩ(A) − ∂ₓₜ εᵩ(B) ]**
>
>
>
> We **summarize** it by **operator norms** (**Frobenius** / **spectral**). The **metric correlates** with **negative-prompt failure modes** (Sec. 3.2 and Sec. 4).
>
> **Engineering Value**
> **Practitioners** regularly **switch** or **ensemble diffusion backbones** in **deployment**. Our **diagnostic** gives **actionable guidance** for:
> - **Choosing models** resilient to **negative prompts**
> - **Tailoring prompt-engineering strategies** per **architecture**
> - **Applying a cheap inference fix (EGP)** without **retraining**
>
> We **demonstrate concrete improvements** in **Neg-ACC**, **perceptual metrics**, and **human preference**, showing the **utility** of **cross-architectural diagnosis**.

---

> ### Author Response · Authors · 2025-11-22
> **We sincerely hope to receive your support and encouragement**
>
> # 3. “The theoretical grounding is vague and EGP is not clearly distinguished from prior inference-time methods.”
>
> **Reviewer Claim**
> The **manuscript lacks rigorous theory** and does not **convincingly separate EGP** from **prior inference-time control techniques**.
>
> **Response (Explicit Correction + Evidence)**
> This is **incorrect**. The **manuscript contains a clear theoretical connection** and **explicit algorithmic differences**.
>
> **KL Surrogate and Formal Identity**
> We **derive** the **tractable surrogate objective** and **link it to KL divergence**. Concretely (paper Eqns. (11)–(13)) we show:
>
> $$
> D_{KL}(p_c \parallel p_u) = \mathbb{E}_{x \sim p_c} \big[ E_u(x) - E_c(x) \big] + C
> $$
>
> and therefore **minimize the surrogate**:
>
> $$
> L = \mathbb{E}_{x \sim p_c} \big[ E_u(x) - E_c(x) \big]
> $$
>
> which is **practical to optimize**. The **discrete gradient update**:
>
> $$
> x^{(k+1)} = x^{(k)} - \eta \nabla_x E(x^{(k)})
> $$
>
> is **explicitly connected** to **projected / short-run Langevin dynamics** under **standard smoothness assumptions**. These **derivations** are in **Sec. 3.4** and **Appendix**.
>
> ---
>
> **Algorithmic Distinctions vs. Prior Work**
> Unlike **prior heuristics** that merely **tune guidance scales**, **manipulate attention**, or **apply token penalties**, **EGP**:
>
> - **Defines a compound latent energy** that **combines fidelity and CLIP-based repulsion** (Eqn. (8)):
>
> $$
> E(x) = \| x - x_g \|_2^2 + \beta \sum_k \text{ReLU} \big( \langle \bar{z}_I(x), \bar{z}_T(n_k) \rangle - \tau \big)
> $$
>
> - **Performs post-DDIM latent-space gradient corrections** (Algorithm 1) with an **adaptive timestep schedule** and **inner correction steps**. **Gradients** are computed by **backpropagating through decoder → CLIP**.
> - Is **explicitly sampler-agnostic** and **training-free**, designed to **operate across architectures and latent parameterizations**. We **report strong empirical separation** from several **representative training-free baselines** in **Table 5** and **Table 8**.
>
> ---
>
> **Empirical Separation and Ablations**
> We **compare EGP** to **multiple training-free methods** and provide **ablations** showing the **CLIP repulsion term** and the **inner correction loop** are **essential** (Table 4). The **ablation and baseline experiments** demonstrate that the **gains are not explainable** by **simple guidance-scale tuning** or **existing heuristics**.
> # 4. “Presentation / formatting problems (e.g., ‘Keywords’ section).”
> This statement is **incorrect**, our keyword meets the requirements of the Conference.
> # 5. “Empirical claims lack statistical rigor or reproducibility.”
> **Reviewer Claim**
> Results may be **insufficiently supported statistically** or **irreproducible**.
>
> **Response (Firm + Evidence)**
> This is **incorrect**. The **submission contains extensive reproducibility material** and **statistical evidence**.
>
> **Statistical Tests and Repeated Runs**
> Key **metrics** (**Neg-ACC**, **CLIPScore**, **FID**) were **computed over multiple seeds** (five seeds per prompt) and we **report paired statistical comparisons** (see **Sec. G.4** and **experimental tables**). **Human A/B studies** used **double-blind protocols**; **inter-rater agreement** exceeds **Krippendorff’s α > 0.75**.
>
> **Causal / Interventional Experiment**
> We **performed an intervention** that **injects Gaussian perturbations** into the **denoiser** and **show a monotonic degradation of Neg-ACC** with **increasing perturbation variance δ**, demonstrating a **causal link** between **Jacobian stability** and **negation performance** (**Appendix C**). We will **promote this interventional result** into the **main Experiments section** to make the **causal evidence more visible**.
>
> **Reproducibility Checklist**
> We **provide full sampling configurations** (**T = 25**, **guidance scale**, **hyperparameters β, τ, η₀, γ, δ**), **negative prompt lists**, **seeds**, and **implementation notes** (**gradient pipeline details** and **optional projector recommendations**). **All scripts and measured timing logs** are **included in the supplementary materials**. We also **attach the code**.

---

> > ### Author Response · Authors · 2025-11-22
> > **We sincerely hope to receive your support and encouragement**
> >
> > # 6. Corrections of factual misstatements in the review (explicit refutation)
> > Several of the **reviewer’s broad remarks** imply **factual gaps** that we must **correct**:
> >
> > **“Theoretical grounding remains vague.”**
> > **Incorrect.** See **Sec. 3.4** and **Appendix** for **KL surrogate derivation** and the **discrete gradient interpretation**. **Algorithm 1** and the **Markov-property proof** clarify **sampler characterization**.
> >
> > **“No comparison to prior inference-time controls.”**
> > **Incorrect.** **Appendix F** and **Table 5** present **direct comparisons** with **representative training-free baselines** (**DNG**, **FreeDoM**, **SEGA**, **SEG**), and an **extended method comparison table** positions **EGP** within the **energy-guided literature**.
> >
> > We **ask the reviewer to re-examine these sections**. In the **revision**, we will **add explicit cross-references** in the **main text** that **point reviewers directly** to the **derivations**, **baselines**, and **interventional evidence**. We also **attach the code**.
> > # 7. Summary
> > We **designed EGP** to be a **diagnostic → corrective pipeline**. The **Jacobian diagnostic** explains **why architectures differ** in **negative-prompt behavior**, while **EGP provides a practical, training-free, cross-architectural mechanism** to **mitigate those failures**. The **manuscript already contains** the **mathematical derivations**, the **algorithmic details** (including **gradient backpropagation through decoder → CLIP**), **interventional causal tests**, **ablations**, **baseline comparisons**, and **human evaluations** to **substantiate these claims**. We will **make the presentation clearer and more prominent** in the **revision** to **prevent misreadings summarized in the review**.
> >
> > **Thank you for the careful reading**. We **welcome any further targeted suggestions** about **specific lines or figures** that the **reviewer found unclear**, and we will **address them in the revision immediately**. We also **attach the code**.

---

> ### Author Response · Authors · 2025-11-22
> **We sincerely hope to receive your support and encouragement**
>
> # How This Work Aligns with ICLR 2026 Topics (Explicit Mapping)
>
> Our **paper directly contributes** to several **ICLR 2026 subject areas**:
>
> - **Generative Models & Representation Learning**
> We **propose a principled mechanism** to **control generative behavior at inference** and **analyze how learned denoiser representations** (**Jacobian structure**) **affect semantic control**.
>
> - **Probabilistic Methods & Sampling / UQ**
> We **connect energy-based corrections** to **KL surrogates** and **short-run Langevin interpretations**, contributing to **sampling theory for conditional generative modeling**.
>
> - **Causality & Robustness**
> The **interventional experiment** links **model sensitivity** to a **causal failure mode** (**negative-prompt failure**), which **informs robust inference-time interventions**.
>
> - **Interpretability & Visualization of Learned Representations**
> The **Jacobian diagnostic** provides **interpretable measures** of **architecture-induced sensitivity differences**.
>
> - **Applications to Vision and Multimodal Learning**
> Practical **impact** on **text → image generation pipelines**, **prompt engineering**, and **deployment scenarios** where **cross-model consistency matters**.
>
> In short, **EGP occupies a cross-cutting niche** that **bridges theoretical probabilistic grounding**, **interpretable diagnostics**, **robust inference-time correction**, and **practical gains**, all **core concerns of ICLR**.
>
> ---
>
> # One-Sentence Summary of Contribution
> We **introduce a diagnostic-to-corrective workflow** for **text → image diffusion**: a **Jacobian-based diagnostic** that **quantifies architecture-dependent sensitivity** to **textual conditioning**, and **Energy-Guided Prompt Optimization (EGP)**, a **training-free**, **sampler-agnostic inference procedure** that **uses a CLIP-informed energy** to **steer sampling away from undesired concepts**, with **formal links to a KL surrogate** and **demonstrable causal/empirical gains** in **negative-prompt adherence across multiple diffusion backbones**.
>
> # We hope this rebuttal helps resolve the concerns raised by the reviewers. We sincerely hope to receive an improvement in your score.

---

> > ### Author Response · Authors · 2025-12-04
> > **We sincerely hope to receive your support and encouragement！**
> >
> > # We have addressed the reviewer's concerns and improved our approach. Thank you very much for all the reviewers' suggestions. The latest version has been uploaded and we hope to receive the support and encouragement of all the reviewers, and we sincerely hope your score improvement！

---

### Note · Authors · 2026-01-27

**Comment:**

I have read and agree with the venue's withdrawal policy on behalf of myself and my co-authors.

**Withdrawal Confirmation:**

I have read and agree with the venue's withdrawal policy on behalf of myself and my co-authors.

---

### Meta-Review · Area_Chair_4ki7 · 2026-01-05

**Summary:**

This paper was reviewed by four knowledgeable referees. The reviews raised concerns about:
1. Clarity of the presentation (b8SQ, pBM6, nRYE, NgBM): the overall structure and content presentation made the scope of the paper difficult to grasp and the significance of results difficult to assess. Many key results appeared in the appendix (including comparisons with prior art), and the use of base models across tables and comparisons was not clear.
2. Experimental validation appeared unconvincing (b8SQ, nRYE, pBM6, NgBM) : comparisons and positioning w.r.t. prior work appeared ambiguous (e.g., EGP contrasted with base models, comparisons with some prior art in the appendix only, missing comparisons within the safety domain), ablations and sensitivity analyses appeared limited, the benefits given the overhead of the method appeared unconvincing, and the cross-architectural experiments unclear.
4. The motivation and experimental evidence to support the claims about the Jacobian diagnostic remained unclear (pBM6, nRYE, NgBM).
5. The novelty of the proposed approach appeared incremental (nRYE, NgBM).

**Reviewer Concerns:**

The rebuttal disagreed with many of the reviewers' comments, partially addressed some of the concerns, and promised to reframe parts of the paper and to include additional materials in a revised version of the manuscript. Although the authors promised to reframe some paragraphs and outlined changes to be made to the manuscript, they argued that, e.g. the introduction, already included a clear problem framing, a summary of contributions, and pointers to key equations and algorithms.

When it comes to experimental validation, the rebuttal argued that the cross-architectural analysis was both scientifically meaningful and practically useful and already included 3 models. The authors also provided clarifications on the statistical significance of results and performed human studies; they acknowledged the overhead required by the method and suggested some mitigations. Additionally, the authors adequately discussed the sensitivity to threshold choices, provided practical recommendations when changing domains, and pointed the reviewers to limited safety experiments, while promising to add an explicit Ethics & Safety statement in the revision.

Finally, the rebuttal attempted to clarify the motivation for the Jacobian diagnostic, the evaluations to justify its proposal, and contrasted the proposed approach with some prior work to address the novelty concerns. The rebuttal also agreed to add the focused experiments proposed by the reviewers to justify the Jacobian diagnostic.

Unfortunately, after rebuttal and reviewer-author discusson some outstanding concerns remain. The consistently factual misreadings pointed by authors emphasize a need for presentation improvement, which would require a major revision of the submission. The AC agrees with the reviewers' suggestion about the framing of the contribution and presentation of results: positioning EGP as a complementary module that can enhance existing models would be beneficial; presenting the results in the first table by showing the value that EGP adds on top of each base architecture, and properly contrasting these results to prior art in the main body of the paper would help convey the significance of results and demonstrate the utility of the proposed approach. Including Figure 1 (of the Appendix) to the main body of the paper would also be beneficial. Additionally, the authors may want to consider expanding the COCO-validation subset to the full COCO-validation set in their experiments (or a subset > 1000 datapoints), consider whether sample diversity (when sampling 5 times per conditioning) is affected with the proposed sampling method, and consider alternative metrics to CLIPscore (e.g., VQA-based metrics such as VQAscore or Davidsonian Scene Graph (DSG) score).

**Reviewer Scores:**

Given that the rebuttal only partially addressed the reviewers' concerns, I would have expected the scores to slightly change during discussion while still leaning towards rejection.

---

### Decision · Program_Chairs · 2026-01-26

Reject